# Learning Randomized Reductions

**Ferhat Erata** [1]  **Orr Paradise** [2]  **Thanos Typaldos** [1]  **Timos Antonopoulos** [1]  **ThanhVu Nguyen** [3]
**Shafi Goldwasser** [4]  **Ruzica Piskac** [1]

## Abstract

Randomized self-reductions (RSRs) express $f(x)$ using $f$ evaluated at random correlated points, enabling self-correcting programs, instance-hiding protocols, and applications in complexity theory and cryptography. Yet discovering RSRs has required manual expert derivation for over 40 years, limiting their practical use. We present Bitween for automated RSR learning. First, we formalize RSR learning with sample complexity analysis under correlated sampling. Second, we develop Vanilla Bitween, which integrates multiple backends (linear regression, genetic programming, symbolic regression, and mixed-integer programming). The linear regression backend outperforms the others, discovering RSRs for 43 of 80 functions (54%) in RSR-Bench, our benchmark suite, including the first known reduction for sigmoid. Third, we introduce Agentic Bitween, a neuro-symbolic approach where LLM agents propose novel query functions beyond the fixed set $(x + r, x - r, x \cdot r, x, r)$ in prior work. Agentic Bitween discovers RSRs for 64 of 80 functions (80%), outperforming pure neural baselines in both RSR discovery and verification accuracy.

## 1. Introduction

Random self-reducibility was first defined by Goldwasser & Micali (1984) in the context of worst-case to average-case reductions to show that concrete encryption schemes were hard on the average to break if the underlying problem was hard in the worst case. In subsequent work, Blum et al. (1993) introduced self-correcting programs, showing that a *self-corrector* can transform a program correct on most inputs into one correct on every input with high probability, using only black-box access. Such self-correctors exist for

any *randomly self-reducible* (RSR) function, where $f(x)$ can be recovered by computing $f$ on random correlated points. RSRs have found applications in cryptography protocols (Goldwasser & Micali, 2019), average-case complexity (Feigenbaum & Fortnow, 1993), instance hiding (Abadi et al., 1987), result checkers (Blum et al., 1993), and interactive proof systems (Blum & Kannan, 1995; Goldwasser et al., 2019).

Yet for over four decades since Goldwasser and Micali's original work, discovering RSR properties has remained a manual, expert-driven process. Previous work (Blum et al., 1993; Rubinfeld, 1999) required manual derivation by experts, and existing methods are limited to a handful of fixed query functions: $x + r$, $x - r$, $x \cdot r$, $x$, and $r$. This severely restricts discoverable RSRs, as many functions require sophisticated patterns involving derivatives, integrals, or domain-specific transformations. The core challenge is to construct a hypothesis space with the right query functions and algebraic relationships that enable the reduction.

We present BITWEEN[1] for automated RSR learning. Our approach samples programs on random values, uses regression with heuristics, attempts to find self-reductions, and formally verifies results. BITWEEN has two variants: Vanilla Bitween (V-BITWEEN) integrates different regression backends within the traditional fixed query functions setting, while Agentic Bitween (A-BITWEEN) integrates large language models with V-BITWEEN tools to dynamically discover novel query functions beyond the fixed set. On RSR-BENCH, our benchmark of 80 scientific and machine learning functions, V-BITWEEN demonstrates that the linear regression-based backend outperforms alternative backends including genetic programming (GPLearn (Stephens, 2015)), symbolic regression (PySR (Cranmer, 2023)), and mixed-integer programming (Gurobi (Gurobi Optimization, LLC, 2026)) for RSR discovery, while A-BITWEEN discovers new RSR properties producing fewer false positives than pure neural approaches.

This work makes five key contributions: (1) introduces a rigorous theoretical framework for learning randomized self-reductions with formal definitions and sample com-

---

[1]Yale University, USA [2]EPFL, Switzerland [3]George Mason University, USA [4]UC Berkeley, USA. Correspondence to: Ferhat Erata <ferhat.erata@yale.edu>.

*Proceedings of the 43rd International Conference on Machine Learning*, Seoul, South Korea. PMLR 306, 2026. Copyright 2026 by the author(s).

[1]Code: https://github.com/ferhaterata/learning-randomized-reductions

plexity analysis; (2) builds V-BITWEEN, a regression-based pipeline integrating multiple symbolic backends, and demonstrates that its linear-regression backend is more suitable than the other backends with fixed query functions; (3) builds A-BITWEEN, a neuro-symbolic variant in which an LLM agent drives V-BITWEEN's tools and proposes novel query functions beyond the standard set; (4) creates RSR-BENCH, a comprehensive benchmark of 80 scientific and machine-learning functions across 8 categories (basic arithmetic, exponential, logarithmic, trigonometric, hyperbolic, inverse trigonometric, machine-learning activations, and special mathematical functions), drawn from the self-testing literature and functional-equation theory; and (5) demonstrates that the framework generalizes beyond scalar functions to non-scalar algebraic structures (matrices, quaternions, octonions, Clifford and Lie algebras), with no architectural changes.

**Conflict of Interest Disclosure.** The authors declare no financial or non-financial conflicts of interest related to this work.

## 2. Motivating Example

In this section we illustrate the use of randomized self-reductions (RSR) and how BITWEEN computes them on the example of the sigmoid function, $\sigma(x) = 1/(1 + e^{-x})$. The sigmoid function is commonly used in neural networks (Han & Moraga, 1995). The program $\Pi(x)$ given in Figure 1 approximates the $\sigma(x)$ by using a Taylor series expansion. Here, $\Pi$ denotes the implementation of sigmoid function $\sigma$. Clearly, $\Pi(x)$ computes only an approximate value of $\sigma(x)$. We invoked BITWEEN on $\Pi$ and it derived the following RSR (using the linear regression-based backend):

$$\sigma(x) = \frac{\sigma(x+r)(\sigma(r)-1)}{2\sigma(x+r)\sigma(r) - \sigma(x+r) - \sigma(r)} \qquad (1)$$

where $r$ is some random value. We verified that indeed the sigmoid function satisfies Equation (1). To the best of our knowledge, this is the first known RSR for the sigmoid function. In this example, BITWEEN inferred this RSR with 15 independent and random samples of $x$ and $r$. In the plot in Figure 1, we depict the value of $\Pi(x)$, identified on the graph with ▲, and the values of $\Pi(x+r), \Pi(x-r)$ and $\Pi(r)$, shown on the graph with ●, ■ and ⋆. Notice that all ▲'s are lying on the line depicting $\sigma(x)$.

Moreover, our derived RSR computes $\sigma(x)$ by using only $\sigma(x + r)$ and $\sigma(r)$, although the algorithm of BITWEEN is computing $\Pi$ on random (yet correlated) inputs $x+r, x-r$, and $r$. The learned RSR in Equation (1) can be used to compute the value of $\Pi(x)$ at any point $x$ by evaluating $\Pi(x+r)$ and $\Pi(r)$. Since $\Pi(x) \neq \sigma(x)$ for some values of $x$, randomized reductions can be used to construct self-

correcting programs (Tompa & Woll, 1987; Blum et al., 1993; Rubinfeld, 1994).

The RSR in Equation (1) can also be used in an instance hiding protocol (Abadi et al., 1987). As an illustration, if a weak device needs to compute $\sigma$ on its private input $x$, it can do that by sending computing requests to two powerful devices that do not communicate with each other: the first powerful device computes $\sigma(r)$ with random $r$, and the second powerful one computes $\sigma(x+r)$. After receiving their outputs, the weak device can compute $\sigma(x)$ by evaluating Equation (1).

Additionally, in this particular case the derived RSR can be used to reduce the computation costs. If a weak device computes the value of $\sigma(r)$ beforehand and stores it as some constant $C$, then the computation of $\sigma(x)$ simplifies to $\frac{\sigma(x+r)(C-1)}{2\sigma(x+r)C - \sigma(x+r) - C}$. While $x$ is a `double`, $x+r$ might require less precision.

## 3. Related Work

*Symbolic Regression as Computational Backend.* Symbolic regression discovers mathematical expressions from data without assuming functional forms, using approaches ranging from genetic programming (Koza, 1992; Cranmer, 2023; Stephens, 2015), physics-inspired methods (Udrescu & Tegmark, 2020; Udrescu et al., 2020), to neural approaches (Petersen et al., 2021). Recent advances include RAG-SR (Zhang et al., 2025) (retrieval-augmented generation), ParFam (Scholl et al., 2025) (neural-guided continuous optimization), and MetaSymNet (Li et al., 2025) (adaptive tree-like networks).

Our task differs fundamentally: rather than discovering expressions from data, we seek randomized self-reduction properties for *known* mathematical functions. We use symbolic regression methods as computational backends; V-BITWEEN employs them within our regression-based learning framework to discover polynomial relationships among correlated query evaluations, while A-BITWEEN uses the inference and verification tools of V-BITWEEN with novel query functions. Within this framework, linear regression backend outperforms genetic programming (Stephens, 2015), symbolic regression (Cranmer, 2023), and MILP (Cozad & Sahinidis, 2018; Austel et al., 2017) backends. These methods often timeout or produce approximations insufficient for RSR verification. We also integrated recent methods (Zhang et al., 2025; Scholl et al., 2025; Li et al., 2025) as backends, but they did not perform reliably under our constraints (see Section 5); more systematic hyperparameter tuning may enable their use in future work.

*Mathematical Discovery and Neuro-Symbolic Learning.* Automated mathematical discovery dates back to AM (Lenat,

```
1  double Π(double x) {
2    const int trms = 30;
3    double sum = 1.0;
4    double trm = 1.0;
5    double neg_x = -x;
6    for (int n = 1; n < trms; n++) {
7      trm *= neg_x / n;
8      sum += trm;
9    }
10   return 1.0 / (1.0 + sum);
11 }
```

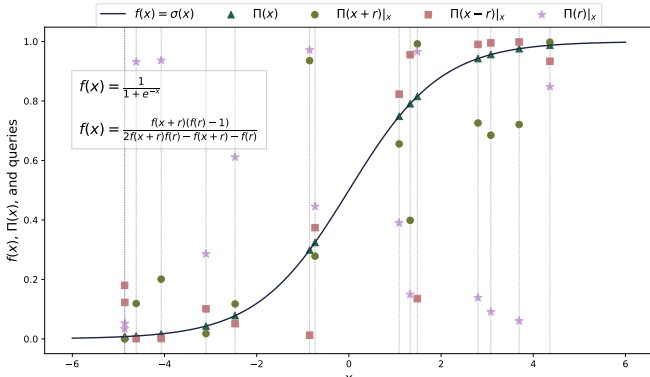

*Figure 1.* An approximate implementation of the sigmoid activation function used as oracle and a graph representing randomly selected points by BITWEEN which are then used to discover an RSR.

1976) and EURISKO (Lenat, 1983), with recent systems like the Ramanujan Machine (Raayoni et al., 2021) discovering mathematical constants and MathConcept (Davies et al., 2021) guiding mathematical intuition. Neural-symbolic integration has produced systems like Neural Module Networks (Andreas et al., 2016) and Neurosymbolic Programming (Chaudhuri et al., 2021). Recent LLM-based systems (GPT-4 (OpenAI, 2024), Claude (Anthropic, 2024), Llemma (Azerbayev et al., 2024)) demonstrate strong mathematical reasoning but suffer from hallucination. Our Agentic Bitween uniquely uses LLMs to discover novel query functions validated through symbolic regression, combining neural creativity with formal verification.

*Self-Correcting Algorithms and Property Inference.* The theoretical foundations of randomized self-reductions were established by Blum et al. (Blum et al., 1990; Lipton, 1991; Blum et al., 1993), showing that RSR properties enable self-correction for faulty programs. While program property inference tools like Daikon (Ernst et al., 2007) and DIG (Nguyen et al., 2012; 2021) discover invariants dynamically, they focus on program properties rather than mathematical functions and do not discover randomized reductions. Our work provides the first practical system for discovering RSRs, operationalizing decades-old theoretical results with a learning framework that goes beyond existing tools.

## 4. Theoretical Foundations

In this section, we give a definitional treatment of learning randomized self-reductions (RSRs). Our goal is to rigorously define the setting in which BITWEEN resides, which may be of independent interest for future theoretical work. Throughout this section, we will say that a set $Z$ is *uniformly-samplable* if it can be equipped with a uniform distribution; we let $z \sim Z$ denote a uniformly random sample from $Z$.

We start from a definition of randomized self-reductions (Goldwasser & Micali, 1984). Our presentation takes after Lipton (1989) and Goldreich (2017), modified for convenience.

**Definition 4.1** (Randomized self-reduction). Fix a uniformly-samplable *input domain* $X$, a *range* $Y$, and uniformly-samplable *randomness domain* $R$. Let $f \colon X \to Y$; $q_1, \ldots, q_k \colon X \times R \to X$ (query functions); and $p \colon X \times R \times Y^k \to Y$ (recovery function) such that for all $i \in [k]$ and $x \in X$, $u_i := q_i(x, r)$ is distributed uniformly over $X$ when $r \sim R$ is sampled uniformly at random.[2]

We say that $(q_1, \ldots, q_k, r)$ is a *(perfect) randomized self-reduction (RSR) for* $f$ if for all $r \in R$, letting $u_i := q_i(x, r)$ for all $i \in [k]$, the following holds:

$$f(x) = p\left(x, r, f(u_1), \ldots, f(u_k)\right). \tag{2}$$

In other words, Equation (2) holds with probability 1 over randomly sampled $r \sim R$. For errors $\rho, \xi \in (0, 1)$, we say that $(q_1, \ldots, q_k, p)$ is a $(\rho, \xi)$-*approximate randomized self-reduction* $((\rho, \xi)$-*RSR) for* $f$ if, for all but a $\xi$-fraction of $x \in X$, Equation (2) holds with probability $\geq 1 - \rho$ over the random samples $r \sim R$. That is, writing $\tilde{f} = p(x, r, f(u_1), \ldots, f(u_k))$ where $u_i := q_i(x, r)$:

$$\Pr_{x \sim X}\left[\Pr_{r \sim R}\left[f(x) = \tilde{f}\right] \geq 1 - \rho\right] \geq 1 - \xi.$$

Given a class of query functions $Q \subseteq X^{X \times R}$ and recovery functions $P \subseteq Y^{X \times R \times Y^k}$, we let $\mathrm{RSR}_k(Q, P)$ denote the class of functions $f \colon X \to Y$ for which there exist $q_1, \ldots, q_k \in Q$ and $p \in P$ such that $f$ is perfectly RSR with $(q_1, \ldots, q_k, p)$. We write $\mathrm{RSR}(Q, P)$ when the number of queries is irrelevant.

In the literature (Lipton, 1989; Goldreich, 2017), the query functions are defined as randomized functions of the input

---

[2]Importantly, the $u_i$'s must only satisfy *marginal uniformity*, but may be correlated among themselves.

$x$ to be recovered. That is, as random variables $\tilde{q}_i(x) \sim X$ rather than our deterministic $q_i(x, r) \in X$. The definitions are equivalent; we simply make the randomness in $\widetilde{q}_i$ explicit by giving it $r$ as input. This choice will have two benefits: (1) It makes explicit the amount of random bits used by the self-reduction, namely, $\log_2 |R|$. (2) It lets us think about the query functions as deterministic. This could allow one to relate traditional complexity measures (e.g., VC dimensions) of the function classes $Q$ and $P$ to those of the function $f$.

To elaborate more on the second point, let us restrict the discussion to polynomials over finite fields, i.e., $f \colon \mathbb{F}_n^m \to \mathbb{F}_n$ for some $n \in \mathbb{N}$. Lipton (1989) showed that even extremely simple choices of $Q$ and $P$ can be very expressive.

*Fact* 4.2 (Lipton, 1989). Any $m$-variate polynomial $f \colon \mathbb{F}_\ell^m \to \mathbb{F}_\ell$ of degree $d < \ell - 1$ is perfectly randomly self-reducible with $k = d + 1$ queries and randomness domain $R = \mathbb{F}_\ell^m$. Furthermore, the queries $q_1, \ldots, q_{d+1} \colon \mathbb{F}_\ell^{2m} \to \mathbb{F}_\ell^m$ and recovery function $p \colon \mathbb{F}_\ell^{2m+d+1} \to \mathbb{F}_n$ are linear functions.

The question of interest is whether, given access to samples from $f \in \mathrm{RSR}_k(Q, P)$, it is possible to learn an (approximate) RSR for $f$. Before we can continue, we must first specify how these samples are drawn. Typically, learners are either given input-output pairs $(x, f(x))$ where $x$'s are either *independent random samples*, or chosen by the learner herself. Learning RSRs will occur in an intermediate access type, in which $x$'s are drawn in a correlated manner. We formally define these access types next. Our presentation is based on that of O'Donnell (2014), who, like us, is focused on samples drawn from the uniform distribution. We note that learning from uniformly random samples has been studied extensively in the literature (Verbeurgt, 1990; Hancock, 1993; Kucera et al., 1994; Golea et al., 1996; Jackson et al., 2002; Klivans et al., 2004; Jackson & Servedio, 2006).

**Definition 4.3** (Sample access). Fix a uniformly-samplable set $X$, function $f \colon X \to Y$ and a probabilistic algorithm $\Lambda$ that takes as input $m$ *(labeled) samples* from $f$. We consider three types of *sample access to $f$*: (1) *Independent random samples:* $\Lambda$ is given $(x_j, f(x_j))_{j=1}^m$ for independent and uniformly sampled $x_j \sim X$. (2) *Correlated random samples:* $\Lambda$ is given $(x_j, f(x_j))_{j=1}^m$ drawn from a distribution such that, for each $j \in [m]$, the marginal on $x_j$ is uniformly random over $X$. However, different $x_j$'s may be correlated. (3) *Oracle queries:* During $\Lambda$'s execution, it can request the value $f(x)$ for any $x \in X$. The type of sample access will be explicitly stated, unless clear from context.

*Remark* 4.4. Facing forward, we note that more restrictive access types will correspond to more challenging settings of learning (formally defined in Definition 4.5). Intuitively (and soon, formally), if $F$ is learnable from $m$ independent samples, it is also learnable from $m$ correlates samples, and

$m$ oracle queries as well.

Learning from correlated samples is one of two main theoretical innovations introduced in this work (the other will appear shortly). We note that PAC learning (Valiant, 1984) requires learning under *any* distribution $\mu$ of inputs over $X$, whereas we consider learning only when samples are drawn from the uniform distribution over $X$ (with possible correlation). Learning from correlated samples could be adapted to arbitrary distributions $\mu$ by considering correlated samples $(x_j, f(x_j))_j$ such that the marginal on each $x_j$ is distributed according to $\mu$. This interesting setting is beyond the scope of this work.

Finally, for an input $x$ we will use $n = |x|$ to denote its length. This will allow us to place an *efficiency requirement* on the learner (e.g., polynomial time in $n$). For a class of functions $F$ from inputs $X$ to outputs $Y$, we will use $F_n$ to denote the class restricted to inputs of length $n$, and similarly we let $Q_n$ (resp. $P_n$) denote the restriction of the query (resp. recovery) class.[3]

**Definition 4.5** (Learning RSR). Fix a reduction class $(Q, P) = (\bigcup_n Q_n, \bigcup_n P_n)$ and a function class $F = \bigcup_n F_n$ where $F_n \subseteq \mathrm{RSR}_k(Q_n, P_n)$ for some constant $k \in \mathbb{N}$.[4] A $(Q, P, k)$-*learner* $\Lambda$ for $F$ is a probabilistic algorithm that is given inputs $n \in \mathbb{N}$ and $m$ samples of $f \in F_n$, collected in one of the three ways defined in Definition 4.3. $\Lambda$ outputs query functions $q_1, \ldots, q_k \in Q_n$ and a recovery function $p \in P_n$.

We say $F$ is $(Q, P)$-$\mathrm{RSR}_k$-learnable if there exists a $(Q, P, k)$-learner $\Lambda$ such that for all $f \in F$ and $\rho, \xi, \delta \in (0, 1)$, given $m := m(\rho, \xi, \delta)$ labeled samples from $f$, with probability $\geq 1 - \delta$ over the samples and randomness of $\Lambda$, $\Lambda$ outputs $q_1, \ldots, q_k \in Q$ and $p \in P$ that are $(\rho, \xi)$-RSR for $f$.

We say that $F$ is *efficiently* $(Q, P)$-$\mathrm{RSR}_k$-learnable if $\Lambda$ runs in time $\mathrm{poly}(n, 1/\rho, 1/\xi, 1/\delta)$. The function $m(\rho, \xi, \delta)$ is called the *sample complexity* of the learner $\Lambda$. We will omit the $(Q, P)$ prefix when it is clear from context.

Definition 4.5 takes after the classic notion of Probably Approximately Correct (PAC) learning (Valiant, 1984) in that it allows the learner a $\delta$ failure probability, and asks that the learned $q_1, \ldots, q_k, r$ only approximately recover $f$. The main difference is that in Definition 4.5, $\Lambda$ is required to output an approximate RSR for $f$, whereas in PAC learning it is required to output a function $\hat{f} \in F$ that approximates $f$ itself; that is, such that $\hat{f}(x) = f(x)$ with high probability over $x \sim X$.

---

[3]This slight informality will allow us to avoid encumbering the reader with a subscript $n$ throughout the paper.

[4]The requirement that $F_n \subseteq \mathrm{RSR}_k(Q_n, R_n)$ is a *realizability assumption*. We leave the agnostic setting, in which $F_n \not\subseteq \mathrm{RSR}_k(Q_n, R_n)$, to future work.

For a detailed comparison between RSR learning and PAC learning, including claims about their relative strengths and sample complexity relationships, see Claim A.1 and Claim A.2, and their proofs in the theory appendix. The Fundamental Theorem of Learning states that sample complexity of PAC-learning is tightly characterized by the VC-dimension of the function class $F$; it is an open question to obtain an analogous Fundamental Theorem of RSR-Learning, which relates the sample complexity to "intrinsic" dimensions of $Q$, $P$, and $F \subseteq \mathrm{RSR}(Q, P)$.

## 5. BITWEEN: Learning Randomized Self-Reductions

*Algorithm Overview.* BITWEEN expects correlated sample access to a program $\Pi$ which is an alleged implementation of some unknown function $f$. The goal is to learn an RSR for $f$ following our theoretical framework (Definition 4.5). BITWEEN is given the input domain $X$, the class of query functions $Q$, and recovery function degree bound $d$. At a high level, BITWEEN works as follows: (i) generate all monomials of degree $\leq d$ over symbolic variables $\Pi(q(x, r))$ for each query function $q \in Q$; (ii) construct a linear regression problem with a regressand for each monomial; (iii) query $\Pi(x_i)$ and $\Pi(q(x_i, r_i))$ for random $x, r \in X$; (iv) fit the regressands to the samples using sparsifying linear regression, thereby eliminating most monomials; (v) apply rational approximation to convert floating-point coefficients to interpretable rational forms. Algorithm 1 provides the complete algorithm description.

*Regression Formulation and Loss Function.* Bitween formulates RSR discovery as a supervised learning problem by treating each query function $q \in Q$ as a potential target variable. For each query $q$, we construct a regression problem where $\Pi(q(x_i, r_i))$ serves as the dependent variable and the monomials $V(x_i, r_i)$ over all query evaluations serve as features. The loss function for a given target query $q$ is (writing $\Pi_i = \Pi(q(x_i, r_i))$ and $V_i = V(x_i, r_i)$):

$$\mathcal{L}_q(\mathbf{C}) = \frac{1}{m} \sum_{i=1}^{m} \Big( \Pi_i - \sum_{V \in \mathcal{M}} C_V V_i \Big)^2 + \lambda R(\mathbf{C})$$

where $\lambda > 0$ is the regularization parameter (selected via grid search), and $R(\mathbf{C}) = \|\mathbf{C}\|_1$ for Lasso or $R(\mathbf{C}) = \|\mathbf{C}\|_2^2$ for Ridge. Sparsification proceeds iteratively: after initial regression, monomials with coefficients below threshold are eliminated, and regression is repeated on reduced space until convergence. The optimization problem:

$$\widehat{\mathbf{C}}_q = \arg\min_{\mathbf{C}} \mathcal{L}_q(\mathbf{C})$$

The term "modulo regression" in Algorithm 1's caption reflects that the regression step (line 6) can use any backend. This modularity enables comparing optimization paradigms.

---

**Algorithm 1** V-BITWEEN modulo regression

1: **Input:** Program $\Pi$, query class $Q$, recovery function degree bound $d$, input domain $X$, sample complexity $m$.
2: **Output:** Randomized self-reduction $(q_1, \ldots, q_k, p)$ or $empty\_tuple$.
3:
4: For each query function $q \in Q$, initialize a variable $v_q$. // $v_q$ for $\Pi(q(x, r))$
5: Let MON be all monomials of degree at most $d$ over the variables $(v_q)_{q \in Q}$.
6: For each monomial $V \in$ MON, initialize the regressand $C_V$.
7: For each $i \in [m]$, sample input $x_i \in X$ and randomness $r_i \in X$.
8: Query $\Pi$ for the values $\Pi(x_i)$ and $\Pi(q(x_i, r_i))$ for each $q \in Q$.
9:
10: // We will fit $\Pi(x) = \sum_V C_V \cdot V(x, r)$
11: Using sparsifying linear regression fit the regressands $C_V$ to the equations

$$\Pi(x_1) = \sum_{V \in \mathsf{MON}} C_V \cdot V(x_1, r_1), \ldots, \Pi(x_m) = \sum_{V \in \mathsf{MON}} C_V \cdot V(x_m, r_m)$$

12: // Convert to rationals
13: Let $\widehat{C_V}$ denote the fitted regressands. Apply rational approximation to convert each fitted coefficient $\widehat{C_V}$ to its best rational form $\widetilde{C_V}$ using maximum denominator constraint.
14:
15: Initialize an empty set of query functions $\widehat{Q} \leftarrow \emptyset$.
16: **for** each $V \in$ MON **do**
17:     **if** $\widetilde{C_V} \neq 0$ **then**
18:         Add to $\widehat{Q}$ all queries $q$ such that the var $v_q$ appears in the monomial $V$.
19:     **end if**
20: **end for**
21: Let $(\widehat{q_1}, \ldots, \widehat{q_k}) \leftarrow \widehat{Q}$ where $k = |\widehat{Q}|$. For each $\widehat{q_i}$, let $\widehat{v_i}$ denote its corresponding variable defined in Section 5. Define the recovery function,

$$\widehat{p}(x, r, \widehat{v_1}, \ldots, \widehat{v_k}) := \sum_{V : \widetilde{C_V} \neq 0} \widetilde{C_V} \cdot V(x, r).$$

22: **return** the randomized self-reduction $(\widehat{q_1}, \ldots, \widehat{q_k}, \widehat{p})$ or $empty\_tuple$.

---

*Key Technical Components.* The implementation utilizes several involved components detailed in Section D. We first perform *supervised learning conversion* to transform the unsupervised RSR discovery problem into supervised regression, followed by *cross-validation* using grid search with 5-fold cross-validation for hyperparameter optimization. *Sparsification* through iterative dimensionality reduction eliminates irrelevant terms, while *rational approximation* converts floating-point coefficients to interpretable rational forms. Lastly, we verify the discovered properties using the simplification method of SymPy (Meurer et al., 2017) after converting them to zero-valued equations. The complexity analysis shows that for low-degree polynomials, BITWEEN operates in polynomial time $O(t_{\mathrm{terms}} \times n_{\mathrm{samples}} \times n_{\mathrm{features}}^2)$, though the exponential growth of monomials with some degree necessitates careful *degree* and *query* selection in practice.

*Experimental Framework.* Our evaluation encompasses three distinct approaches to RSR discovery. The first is Vanilla Bitween (V-BITWEEN), our core symbolic-regression-based learning framework. Within the fixed query function paradigm, we integrate and compare the following backends: a multiple-linear regression backend utilizing Linear Regression, Ridge and Lasso (V-BITWEEN-LR), a mixed-integer linear programming backend using the Gurobi solver (Gurobi Optimization, LLC, 2026) (V-

BITWEEN-MILP), the PySR tool (Cranmer, 2023) that uses evolutionary algorithms for symbolic regression (V-BITWEEN-PySR), and the GPLearn tool (Stephens, 2015) that uses genetic programming (V-BITWEEN-PySR) for symbolic regression. We also integrated three recent backends, but could not make them work successfully, thus we do not include them in the evaluation. The first tool, RAG-SR (Zhang et al., 2025), for the hyper-parameters we tried either returned very long properties that we discard or timed out. The second, MetaSymNet (Li et al., 2025), was crashing in all benchmarks due to NumPy shape alignment errors. The last one, ParFam (Scholl et al., 2025), did return some verified properties, but only in 4/80 benchmarks and timed out in the rest. We leave for future work a more systematic hyper-parameter search that can potentially make those tools work under our constraints and time limits.

The second is Agentic Bitween (A-BITWEEN) representing our Neuro-Symbolic approach where large language models dynamically propose novel query functions beyond the fixed set $\{x + r, x - r, x \cdot r, x, r\}$ that in turn lead to new properties. The LLM agents are queried only once with the goal of discovering as many verified properties of the function of interest as possible. For this cause, they have in their disposal, aside from their mathematical knowledge, the following three tools. The first is the $symbolic\_verify\_tool$ (Section F.2.2) that can be used to verify if a property equals to zero using the simplification method of SymPy (Meurer et al., 2017). The second is the $infer\_property\_tool$ (Section F.2.1) that can select different V-BITWEEN backends (V-BITWEEN-LR is the default) to provide with functional terms in order to infer new properties. The provided functional terms correspond to the variables $v_q$ of Algorithm 1 that implicitly encapsulate the query functions. For instance, if $f$ is the function of interest, then the LLM agent can propose the functional term $f(x^r)$, in which case the corresponding query function is $x^r$. The third tool, $sequential\_thinking\_tool$ (Model Context Protocol Community, 2024), allows the LLM agent to journal its thoughts, something that was empirically found useful for increasing the other tools' usage and enhancing its overall response.

The last approach is Neural Research (N-RESEARCH), which is pure neural and serves as a comparison baseline for A-BITWEEN. It uses LLM agents with access only to the $sequential\_thinking\_tool$, because it enhances their exploration. Comparing this approach with A-BITWEEN highlights the importance of the other two tools in the RSR discovery task.

## 6. Empirical Evaluation

We evaluate Bitween on RSR-Bench, a benchmark suite of 80 mathematical functions spanning scientific computing

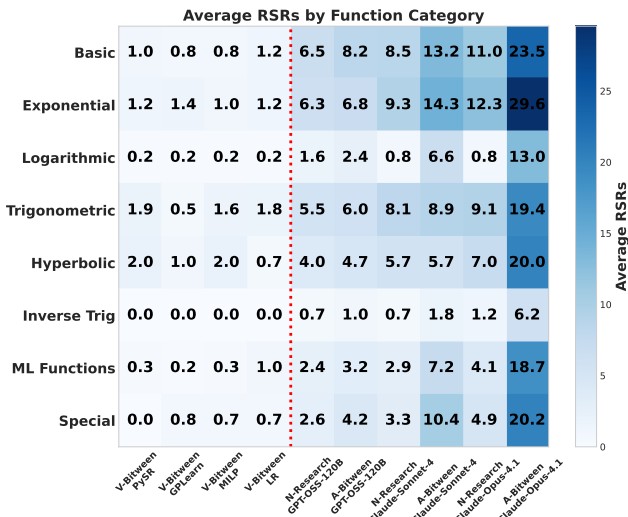

*Figure 2.* Performance heatmap of average verified RSRs across mathematical function categories. The dotted red line represents the boundary from symbolic to neural methods.

and machine learning applications. Our evaluation validates the two key contributions outlined in Section 1, namely that V-BITWEEN-LR is more suitable for RSR-bench with fixed query functions, and that A-BITWEEN discovers novel query functions.

*RSR-Bench Benchmark Suite.* We constructed RSR-Bench comprising 80 continuous real-valued functions from diverse mathematical domains: basic arithmetic (linear, squared, cube), exponential and logarithmic functions, trigonometric and hyperbolic functions, inverse trigonometric functions, machine learning activation functions (sigmoid, ReLU, GELU, etc.), loss functions, and special mathematical functions (gamma, error function, Gudermannian). Some benchmarks include ground-truth RSR properties with minimal query complexity, sourced from self-testing literature (Blum et al., 1993; Rubinfeld, 1999) and functional equation theory (Aczél, 1966; Kannappan, 2009).

*Configuration.* Experiments were conducted on a MacBook Pro with 32GB memory and Apple M1 Pro 10-core CPU. For trigonometric, hyperbolic, and exponential functions, we configured term generation up to degree 3; for others, degree 2. We used uniform sampling in [-10, 10] with error bound $\delta = 0.001$. Each experiment was repeated 5 times for statistical significance with timeout of 1800 sec. For the LLMs we used GPT-OSS-120B (OpenAI, 2025) locally, and Claude-Sonnet-4 (Anthropic, 2025), Claude-Opus-4.1 (Anthropic, 2024)) remotely.

*Aggregate Results.* Table 1 shows aggregate information about the performance of different methods. The `results` category consists of the number of RSR, verified and unverified properties found by each method, while the `rsr`

*Table 1.* Aggregate evaluation information about the different methods. The format of results is `"RSR/verified|unverified"`, where the verified (unverified) properties passed (failed) automatic mathematical validation, and the RSR properties are a manually confirmed subset of the verified ones. The coverage information shows the (rounded) percentage of benchmarks for which the method returned at least one RSR. The format of time is `"min|average|max"` to show the (rounded) runtime range of each method. Highlighted are the methods with the most RSR count, most RSR coverage and least average runtime.

| V-BITWEEN | PySR | GPLearn | MILP | LR |
|---|---|---|---|---|
| results | 61 / 61 \| 60 | 48 / 48 \| 54 | 74 / 74 \| 29 | 87 / 87 \| 46 |
| rsr coverage | 38% | 32% | 51% | 54% |
| time (sec) | 113 \| 335 \| 956 | 0 \| 140 \| 629 | 0 \| 11 \| 47 | 0 \| 5 \| 19 |
| **N-RESEARCH** | | **GPT-OSS-120B** | **Claude-Sonnet-4** | **Claude-Opus-4.1** |
| results | | 170 / 360 \| 112 | 191 / 421 \| 153 | 250 / 539 \| 172 |
| rsr coverage | | 62% | 60% | 64% |
| time (sec) | | 5 \| 10 \| 25 | 56 \| 110 \| 203 | 197 \| 286 \| 528 |
| **A-BITWEEN** | | **GPT-OSS-120B** | **Claude-Sonnet-4** | **Claude-Opus-4.1** |
| results | | 157 / 407 \| 48 | 293 / 729 \| 14 | 793 / 1628 \| 26 |
| rsr coverage | | 59% | 66% | 80% |
| time (sec) | | 6 \| 30 \| 252 | 94 \| 160 \| 274 | 221 \| 378 \| 900 |

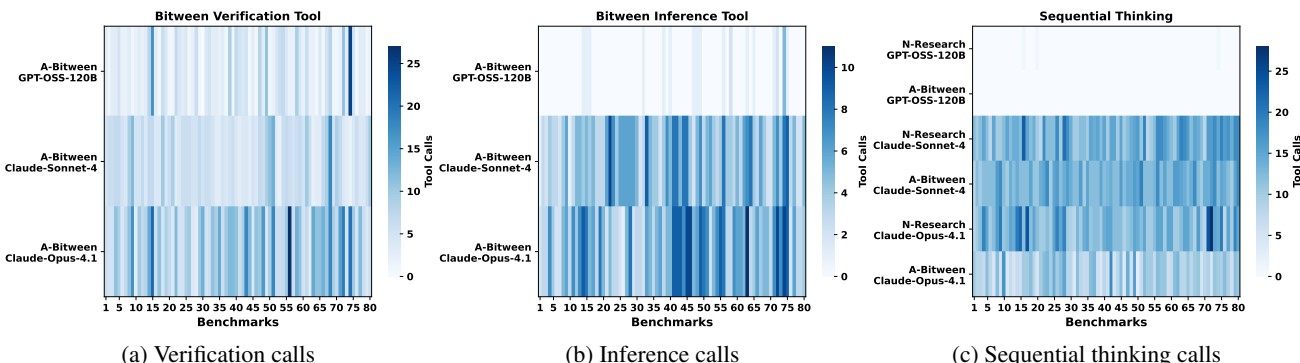

(a) Verification calls      (b) Inference calls      (c) Sequential thinking calls

*Figure 3.* A-BITWEEN's intensive tool usage across all the benchmarks. Particularly useful proved the *sequential_thinking_tool* for guiding the exploration and helping the LLM agent.

`coverage` shows the percentage of the benchmarks, for which each method found at least one RSR. Also, runtime information is included and specifically the minimum, average and maximum values aggregated over all the benchmarks for each method. For each of the tree categories, the best scores among similar methods are highlighted, based on the highest RSR count, on the highest RSR coverage and on the least average runtime, respectively. Note that the RSRs of the neural methods are always less than half of the verified properties, since we manually filtered a large portion of trivial, redundant, or simply non-RSR properties, but we might still have missed some. Among, the symbolic backends, V-BITWEEN-LR scores higher in all categories, while V-BITWEEN-MILP follows. Interestingly, it also returns more unverified properties than the latter. V-BITWEEN-PySR seems to be the slowest (possibly due to our selected hyper-parameters). However, the verified properties here are always RSRs, because we provided the methods with the appropriate terms. Looking, at the N-RESEARCH methods, N-RESEARCH-Claude-Opus-4.1 has the most RSRs and highest coverage, but the fastest among them is N-RESEARCH-GPT-OSS-120B, due to local inference. The similar situation holds for the A-BITWEEN

methods as well. Notice the big difference in the results of the neural methods compared to the symbolic ones. A-BITWEEN-Claude-Opus-4.1 has found 3 times more RSRs than N-RESEARCH-Claude-Opus-4.1, which has found 6 times more RSRs than V-BITWEEN-LR. The coverage as well as the runtime keep increasing as expected. Lastly, the significantly fewer unverified properties of A-BITWEEN compared to N-RESEARCH methods highlight the effect of the *symbolic_verify_tool* that tries to prevent the LLM from returning a non-verified property as an answer. Per-function results for all methods are tabulated in Tables 8 and 9 (V-BITWEEN) and Tables 10 to 15 (N-RESEARCH/ A-BITWEEN with GPT-OSS-120B, Claude-Sonnet-4, and Claude-Opus-4.1.

*Results per function category.* In order to provide more insight into the difficulty of RSR discovery among categories of functions we grouped several benchmarks together and created 8 categories (details in Section G). For each method we calculated how many RSRs found per category as shown in the heatmap of Figure 2 with darker color representing higher counts. The red dotted line shows the boundary between the symbolic and neural methods. The

*Table 2.* Novel query functions discovered by Agentic Bitween beyond traditional fixed query functions $\{x + r, x - r, x \cdot r, x, r\}$. The query functions appear inside functional terms. For example, the functional term $f(x + \log(k))$ has query function $x + \log(k)$. All properties have been symbolically verified. The right column lists RSRs A-BITWEEN discovers via novel queries (e.g., $\log k$, $\sqrt{x^2 + y^2}$) or non-polynomial form (radicals, rational compositions, transcendental exponents).

| Function | Fixed-query RSRs | Novel-query RSRs (A-BITWEEN) |
|---|---|---|
| Sigmoid | $f(x) - \frac{f(x+r)\,(f(r)-1)}{2f(x+r)f(r) - f(x+r) - f(r)} = 0$ | $f(x + \log(k)) - \frac{k \cdot f(x)}{1 + (k-1) \cdot f(x)} = 0 \mid f(nx) - \frac{f^n(x)}{(1-f(x))^n + f^n(x)} = 0$ |
| Logarithm | $f(x \cdot y) - f(x) - f(y) = 0$ | $f(x^n) - n \cdot f(x) = 0 \mid f(\sqrt{x \cdot y}) - \frac{f(x)+f(y)}{2} = 0$ |
| Modulo | $f(x+r) = f(x) +_R f(r)$ | $f(x + y) - f(f(x) + f(y)) = 0 \mid f(x \cdot y) - f(f(x) \cdot f(y)) = 0$ |
| Gudermannian | *no RSR found* | $\tan(f(x+r)) + \tan(f(x-r)) - 2\cosh(r)\,\tan(f(x)) = 0$ |
| Softmax | $f(x+r,\, y+r) - f(x, y) = 0$ | $f(x,y)\,f(y,z)\,f(z,x) - f(x,z)\,f(z,y)\,f(y,x) = 0$ |
| Inverse | $f(x \cdot y) - f(x)f(y) = 0$ | $f(rx) - f^2(x) \cdot f(\frac{r}{x}) = 0$ |
| $e^{x^2}$ (Gaussian) | *no RSR found* | $f(x)\,f(y) - f\left(\sqrt{x^2 + y^2}\right) = 0$ |
| Sinh | $f(x)^2 - f(x+y)f(x-y) - f(y)^2 = 0$ | $f(x+y) + f(x-y) - 2\sqrt{f(y)^2 + 1}\,f(x) = 0$ |
| $e^{\sin x}$ | *no RSR found* | $f(x-r)\,f(x+r) - f(x)^{2\cos r} = 0$ |

heatmap is mostly aligned with the results of Table 1 showing that the neural methods return more RSRs on average than the symbolic ones. The interesting new information is that there are categories, which are difficult for all methods, such as inverse trigonometric and logarithmic functions, and even more for the symbolic methods that fail completely in the former ones (per-function counts in Tables 8 and 9). Additionally, the heatmap shows that the different V-BITWEEN methods seem to excel in different categories: V-BITWEEN-PySR in trigonometric and hyperbolic, V-BITWEEN-GPLearn in exponential, V-BITWEEN-MILP in hyperbolic, and V-BITWEEN-LR in the basic and machine learning categories. On the other hand regarding the neural methods, the A-BITWEEN variants almost always find more RSRs than their N-RESEARCH counterparts, highlighting again the importance of V-BITWEEN tools.

*Novel Query Functions.* A-BITWEEN's ability to discover novel query functions beyond the traditional fixed set $\{x + r, x - r, x \cdot r, x, r\}$ is a key factor for finding more properties (including RSRs). Table 2 showcases some of the new query functions that A-BITWEEN discovered. It is worth clarifying at this point what a query function is. According to Algorithm 1, the query functions $q$ comprise the query class $Q$ and are used as arguments to the program $\Pi$ to generate the variables $v_q = \Pi(q(x, .))$. In practice, however, A-BITWEEN provides (to the $infer\_property\_tool$) the variables $v_q$ (functional or other terms) instead of the queries $q$. Some examples of variables are $f(x)$, $f(\sqrt{x \cdot r})$, $f(f(x) \cdot f(y))$, $\log(k)$, where $k$ is constant and $f$ is the function (representing $\Pi$). In other words, the query functions are implicitly the arguments of the outermost function of interest $f$ or other independent terms. The table highlights unique query functions, such as $x + \log(k)$ in Sigmoid's term $f(x + \log(k))$, $f(x) + f(y)$ in Modulo's term $f(f(x) + f(y))$, and $x + \log(a)$ in Softmax's

term $f(x + \log(a))$, with the complete list in Table 20 in the Appendix. The Modulo fixed-query entry is the classical modular reduction of Blum et al. (1993); it relies on modular arithmetic (recombination via $+_R$, i.e., addition in $\mathbb{Z}_R$), which lies outside V-BITWEEN-LR's search over real and integer arithmetic.

The $infer\_property\_tool$ is the primary discovery tool, aside from the LLM's mathematical background, but the $symbolic\_verify\_tool$ also helps, because of the way it handles failures. When the LLM agent proposes a property that fails the verification, its simplified (non-zero) expression is returned back to the agent. In many cases, this feedback provides the missing piece to complete the equation, because the remaining expression can be subtracted from the original one in order for the equation to be zero. Figure 3 shows the tool utilization of the different neural methods per benchmark. A-BITWEEN-Claude-Opus-4.1, which was the best of its variant, utilized both tools more, something that led to more discovered properties and significantly fewer false positives (unverified properties) as shown in Table 1 compared to the N-RESEARCH methods. It is also interesting that GPT-OSS-120B did not utilize the tools as much as the other LLMs, which might also be part of the reason why it found fewer properties.

*Computational Efficiency Analysis.* Figure 4 provides insights on the computational efficiency across all methods and benchmarks. The runtime heatmap (left) reveals that V-BITWEEN-LR and V-BITWEEN-MILP run the fastest, whereas V-BITWEEN-PySR requires significantly more time for comparable RSR discovery. The A-BITWEEN variants (which are queried only once) run longer than their pure-neural counterparts because the tool-use loop adds iteration overhead; concretely, A-BITWEEN incurs roughly 1.2–1.8× the runtime of N-RESEARCH using the same model. The

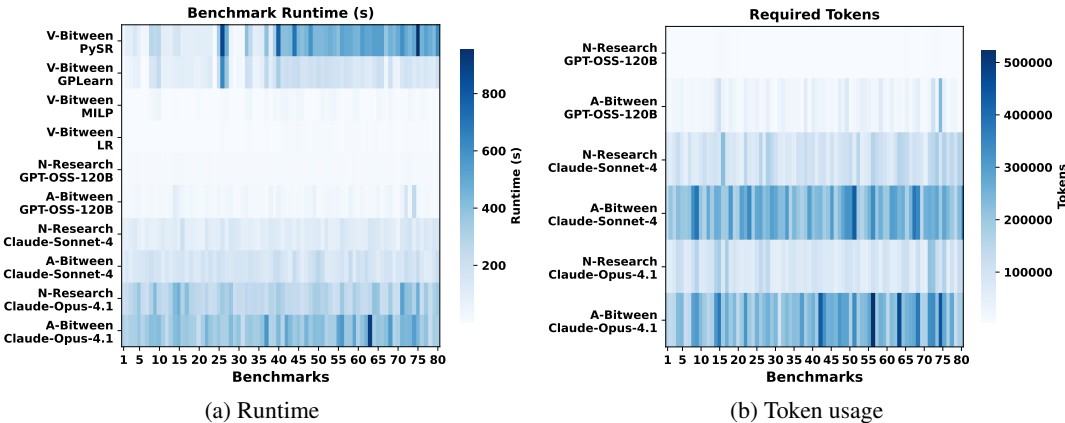

(a) Runtime                                    (b) Token usage

*Figure 4.* (Left) Runtime performance across methods and benchmarks highlighting the speed of V-BITWEEN-LR, V-BITWEEN-MILP and the overhead of A-BITWEEN variants. (Right) Token usage patterns correlate with the LLM's ability to efficiently utilize them.

token usage heatmap (right) reflects the same tool-use overhead, with token consumption running at roughly $2.8$–$3.5\times$ that of N-RESEARCH. The increase stems from iterative tool interactions with V-BITWEEN, feedback loops from tool calls (some unsuccessful) that require re-reasoning, and the inherent complexity of discovering novel mathematical relationships. In return, the agent produces more verified properties containing a diverse set of query functions and unlocks RSRs for functions that V-BITWEEN-LR cannot reach with fixed queries: across 23 RSR-BENCH functions (e.g., modulo, Gudermannian, $e^{x^2}$, $1/x^2$, $x^4$, $\lceil x \rceil$), V-BITWEEN-LR returns no verified RSRs while A-BITWEEN (Claude Opus 4.1) discovers up to 33 RSRs per function (see per-function tables in Section G).

*Generalization to algebraic structures.* RSR-BENCH targets scalar real-valued functions, but BITWEEN's regression and verification pipeline is agnostic to the algebraic structure of the input domain. To probe how broadly the framework applies, we evaluated BITWEEN on 17 additional benchmarks drawn from non-scalar algebraic structures: $2\times2$ matrix functions ($\det$, $\operatorname{tr}$, $\operatorname{tr}(A^2)$), quaternion and octonion norms (additive and multiplicative), Clifford algebras $\mathrm{Cl}(2,0)$ and $\mathrm{Cl}(3,0)$, the Lie bracket $\operatorname{tr}([A,B]^2)$ on $\mathfrak{gl}(2)$, the Killing form on $\mathfrak{sl}(2)$, elementary symmetric polynomials, power sums, and vector operations (norm-squared, cross-product norm). With no architectural changes, A-BITWEEN (Claude Opus 4.1 with $infer\_property\_tool$ and $symbolic\_verify\_tool$) discovers verified RSRs for all 17 functions. V-BITWEEN-LR also handles most of these benchmarks unchanged (11 / 17). We report a more detailed breakdown in Section H.

*Limitations.* While our evaluation demonstrates significant advances, several limitations merit discussion. First, discovered RSR properties may contain redundancies where certain properties are algebraically derivable from others. Although Gröbner basis reduction (Buchberger, 2001; Cox

et al., 2025) could identify minimal generating sets, we refrained from applying it due to its NP-complete complexity, especially given the large number of RSRs discovered. Second, our approach is inherently incomplete, absence of discovered RSRs does not imply non-existence. The infinite space of possible query functions, numerical precision requirements, sampling strategies, and degree limits (2-3 in our experiments) constrain discovery. Functions without discovered RSRs may possess properties requiring higher-degree terms or alternative mathematical representations beyond our current framework. Finally, A-BITWEEN's enhanced performance incurs increased runtime and token usage compared to the pure neural methods due to iterative tool interactions, though this overhead is justified by the the quantity and diversity of discovered RSRs.

## 7. Conclusion

BITWEEN provides the first systematic approach for learning RSRs from mathematical functions, transforming expert-driven discovery into an automated process. Our work delivers two key achievements that validate our contributions: First, V-BITWEEN demonstrates that the linear regression-based backend outperforms the other symbolic method backends for RSR discovery from correlated samples. Second, A-BITWEEN achieves a paradigm shift by dynamically discovering novel query functions through neuro-symbolic reasoning, moving beyond the fixed query set. On RSR-Bench's 80 functions spanning scientific computing and machine learning, neuro-symbolic methods surpass the traditional symbolic ones on automated RSR discovery. The pipeline also generalizes beyond scalar functions to non-scalar algebraic structures. Future work includes regression backends over finite-field arithmetic and broader mathematical domains.

## Acknowledgements

We thank the anonymous reviewers for their constructive feedback, which substantially improved this work. This material is based upon work supported in part by the Defense Advanced Research Projects Agency (DARPA) under Agreement No. HR00112590130 and by the NSF awards CCF-2219995, CNS-2245344, CCF-2318974, CCF-2422036, CCF-2319131, CCF-2238133, and CCF-2200621, and by an Amazon Research Award.

## Impact Statement

This paper presents work whose goal is to advance the field of Machine Learning. There are many potential societal consequences of our work, none which we feel must be specifically highlighted here. The randomized self-reductions discovered by our system enable self-correcting implementations of common machine-learning components such as the sigmoid activation, and could support privacy-preserving protocols based on instance hiding as well as verifiable-computation and result-checking protocols built on self-correctors.

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

## A. Theory: RSR Learning vs PAC Learning

When samples are drawn uniformly and *independently* (Item 1 in Definition 4.3), PAC learnability is a strictly stronger form of learning than RSR learnability, as captured by the following two claims:

*Claim* A.1. Fix query class $Q$, recovery class $P$, and function class $F \subseteq \mathrm{RSR}_k(Q, P)$. If $F$ is (Uniform) PAC-learnable with sample complexity $m_{\mathrm{PAC}}(\varepsilon, \delta)$, then it is RSR-learnable with sample complexity

$$m_{\mathrm{RSR}}(\rho, \xi, \delta) \leq m_{\mathrm{PAC}}(\min(\rho/k, \xi), \delta).$$

However, the RSR-learner may be inefficient.

We note that Claim A.1 is trivial when considering a class $F \subseteq \mathrm{RSR}_k(Q, P)$ characterized by a single RSR, i.e., such that there exist $q_1, \ldots, q_k \in Q$ and $p \in P$ such that Equation (2) holds with probability 1 for all $f \in F$. For one, this follows because the RSR-learner does not need any samples and may simply output $q_1, \ldots, q_k, p$.[5] This gives rise to the following claim.

*Claim* A.2. There exist classes $(Q, P) = (\bigcup_n Q_n, \bigcup_n P_n)$ and $F = \bigcup_n F_n \subseteq \mathrm{RSR}(Q, P)$ such that $F$ is efficiently RSR-learnable from 0 samples, but for any $\varepsilon \leq 1/2$, Uniform PAC-learning $F_n$ requires $m_{\mathrm{PAC}}(\varepsilon, \delta) \geq n$ oracle queries.

Consequentially, Uniform PAC-learning $F_n$ requires at least $n$ correlated or independent random samples (see Remark 4.4).

## B. Proofs

*Proof of Claim A.1.* Suppose $F \subseteq \mathrm{RSR}_k(Q, P)$ is PAC-learnable with sample complexity $m_{\mathrm{PAC}}(\varepsilon, \delta)$ and learner $\Lambda_{\mathrm{PAC}}$. To RSR-learn $F$ with errors $(\rho, \xi, \delta)$, we let $\varepsilon = \min(\rho/k, \xi)$ and draw $m = m_{\mathrm{PAC}}(\varepsilon, \delta)$ labeled samples $(x_i, f(x_i))_{i=1}^m$. The RSR-learner is described in Algorithm 2. For simplicity of notation, we will omit the input length $n$; following this proof is a brief discussion of Algorithm 2's inefficiency with respect to $n$.

---

**Algorithm 2** RSR-learning via PAC-learning.

---

1: **Input:** Query class $Q$, recovery class $P$, and hypothesis class $F \subseteq \mathrm{RSR}_k(Q, P)$. Query complexity $k$ and randomness domain $R$. Uniform PAC-learner $\Lambda_{\mathrm{PAC}}$ for $F$. Labeled samples $(x_i, y_i)_{i=1}^m$.
2: **Output:** Query functions $\hat{q}_1, \ldots, \hat{q}_k \in Q$ and recovery function $\hat{p} \in P$.
3:
4: Invoke $\Lambda_{\mathrm{PAC}}$ on samples $(x_i, y_i)_{i=1}^m$ to obtain a hypothesis $\hat{f} \in F$.
5: **for** each Query functions $(q_1, \ldots, q_k) \in Q^k$ and recovery function $p \in P$ **do**
6:     **for** $x \in X$ and $r \in R$ **do**
7:         Compute $u_i := q_i(x, r)$ for each $i \in [k]$.
8:         **if** $\hat{f}(x) \neq p(x, r, \hat{f}(u_1), \ldots, \hat{f}(u_k))$ **then**
9:             Go to line 5. // Continue to the next $q_1, \ldots, q_k, p$.
10:         **end if**
11:     **end for**
12:     Output $(\hat{q}_1, \ldots, \hat{q}_k, p) := (q_1, \ldots, q_k, p)$.
13: **end for**
14: Output $\perp$.

---

At a high level, the learner invokes $\Lambda_{\mathrm{PAC}}$ to obtain a hypothesis $\hat{f} \in F$ that is $\varepsilon$-close to the ground truth function $f$ (with probability $\geq 1 - \delta$ over the samples). It then uses $\hat{f}$ to exhaustively search through possible query functions $\hat{q}_1, \ldots, \hat{q}_k \in Q$ and recovery functions $\hat{p} \in P$, until it finds those that are a *perfect* RSR for $\hat{f}$.

To conclude the proof, we will show that, because $\hat{f}$ is $\varepsilon$-close to $f$, then $(\hat{q}_1, \ldots, \hat{q}_k, \hat{p})$ is a $(\rho, \xi)$-RSR for $f$.

We say that $x \in X$ is *good* if $\hat{f}(x) = f(x)$. By choice of $\varepsilon$, we know that there are at least $(1 - \varepsilon)|X| \geq (1 - \xi)|X|$ many

---

[5]For a more nuanced reason, note that the sample complexity bound $m_{\mathrm{PAC}}(\rho/k, \delta)$ trivializes: Any class $F$ characterized by a single RSR has *distance* at least $1/k$, meaning that $\Pr_{z \sim X}[\hat{f}(z) \neq f(z)] \geq 1/k$ (Goldreich, 2017, Exercise 5.4). Thus, for any $\varepsilon = \rho/k < 1/k$, $f$ is the only function in $F$ that is $\varepsilon$-close to itself. In other words, learning within accuracy $\varepsilon$ amounts to exactly recovering $f$.

good $x$'s. It therefore suffices to show that for any good $x$,

$$\Pr_{r \sim R} \left[ \begin{array}{c} f(x) = \hat{p}(x, rf(u_1), \ldots, f(u_k)) \\ \text{where } \forall i \in [k] \; u_i := \hat{q}_i(x, r) \end{array} \right] \geq 1 - \varepsilon \cdot k \geq 1 - \rho.$$

The right inequality is by choice of $\varepsilon \leq \rho/k$. For the left inequality,

$$\Pr_{r \sim R} \left[ \begin{array}{c} f(x) = \hat{p}(x, rf(u_1), \ldots, f(u_k)) \\ \text{where } \forall i \in [k] \; u_i := \hat{q}_i(x, r) \end{array} \right] \geq$$

$$\Pr_{r \sim R} \left[ \begin{array}{c} f(u_1) = \hat{f}(u_1), \ldots, f(u_k) = \hat{f}(u_k) \\ \text{where } \forall i \in [k] \; u_i := \hat{q}_i(x, r) \end{array} \right] \geq$$

$$1 - k \cdot \Pr_{x \sim X} \left[ f(x) \neq \hat{f}(x) \right] \geq 1 - k \cdot \varepsilon.$$

Here, the first inequality is because $(\hat{q}_1, \ldots, \hat{q}_k, \hat{p})$ is a perfect RSR for $\hat{f}$, the second is by a union bound and the fact that each $u_i$ is distributed uniformly in $X$ (Definition 4.1), and the last is because $\hat{f}$ is $\varepsilon$-close to $f$.

$\square$

Note that, as mentioned in Claim A.1, the running time of the RSR-learner is not bounded by a polynomial in the number of samples $m$. In more detail, we denote

- $T_{\mathrm{PAC}}(m)$: an upper-bound on the running time of the PAC learner $\Lambda_{\mathrm{PAC}}$ as a function of the number of samples $m = m_{\mathrm{PAC}}(\min(\rho/k, \xi), \delta)$.

- $T_Q(n)$ (resp. $T_P(n)$): an upper-bound on the running time of query functions $q \in Q$ (resp. recovery function $p \in P$) as a function of the input length $n = |x|$.

Then the running time of the RSR-learner is

$$O\left( T_{\mathrm{PAC}}(m) + |Q_n|^k \cdot |P_n| \cdot |X_n| \cdot |R_n| \cdot (k \cdot T_Q(n) + T_P(n)) \right).$$

In particular, $|X_n|$ typically grows exponentially in $n$. Therefore, even if the PAC learner were efficient, the RSR-learner will not be efficient.

*Proof sketch of Claim A.2.* We will show a setting in which an RSR is "learnable" without any samples ($m \equiv 0$). However, without samples it will not be possible to PAC learn a hypothesis for $f$.

Consider the RSR that captures the so-called BLR relation $g(z) = g(z + \tilde{x}) - g(\tilde{x})$ for Boolean functions $g \colon \mathbb{F}_2^\ell \to \mathbb{F}_2$ (Blum et al., 1993). Indeed, this relation characterizes $\ell$-variate linear functions over $\mathbb{F}_2$. In a nutshell, we will choose our reduction class $(Q, P)$ to consist only of the reduction specified by the BLR relation, and the hypothesis class $F$ to consist of all linear functions. Then, a $(Q, P)$-RSR$_2$ learner for $F$ will always output the BLR relation, but on the other hand, PAC learning $F$ requires a superconstant number of samples. Details follow.

We choose the input and randomness domains to be $X_n := R_n := \mathbb{F}_2^n$, and the range $Y_n = \mathbb{F}_2$. The query class $Q_n$ consists of just two query functions

$$Q_n = \{q_n, q'_n\} \quad \text{where } q_n, q'_n \colon \mathbb{F}_2^n \times \mathbb{F}_2^n \to \mathbb{F}_2^n,$$
$$q_n(x, r) := x + r,$$
$$q'_n(x, r) := r.$$

The recovery class $P_n$ is the singleton $P_n = \{p_n\}$ where $p_n(x, r, y, y') := y - y'$. Indeed, the trivial algorithm that takes no samples and outputs $(q, q', p)$ is a $(Q, P)$-RSR$_2$ learner for any linear function $f \colon \mathbb{F}_2^n \to \mathbb{F}_2$.

On the other hand, fix $\varepsilon < 1/2$, $\delta < 1/2$ and number of samples $m < n$. Given $m$ labeled samples $(x_i, f(x_i))_{i=1}^m$, there exists another linear function $f' \neq f$ such that $f'(z_i) = f(z_i)$ for all $i \in [m]$. The learner cannot do better than guess

between $f$ and $f'$ uniformly at random , in which case it outputs $f'$ with probability $1/2$. Lastly, we note that any two different linear functions agree on *exactly* $1/2$ of the inputs in $\mathbb{F}_2^n$, therefore

$$\Pr_{x \sim X}[f(x) = f'(x)] \geq 1 - \varepsilon \iff f = f'.$$

All in all, we have

$$\Pr_{\substack{x_1,\ldots,x_m \sim X \\ \hat{f} \leftarrow \Lambda(x_1, f(x_1),\ldots,x_m, f(x_m))}} \left[ \Pr_{x \sim X}[\hat{f}(x) = f(x)] \geq 1 - \varepsilon \right] = \Pr_{\substack{x_1,\ldots,x_m \sim X \\ \hat{f} \leftarrow \Lambda(x_1, f(x_1),\ldots,x_m, f(x_m))}} \left[ \hat{f} = f \right] \leq 1/2 < 1 - \delta.$$

$\square$

## C. General definitions

**Definition C.1** (Randomized reduction). Let

$$
\begin{aligned}
f &: X \to Y & \text{(Source function)} \\
g_1, \ldots, g_k &: U \to V & \text{(Target functions)} \\
q_1, \ldots, q_k &: X \times R \to U & \text{(Query functions)} \\
p &: X \times R \times V^k \to Y & \text{(Recovery function)}
\end{aligned}
$$

such that $U$ and $V$ are uniformly-samplable, and for all $i \in [k]$ and $x \in X$, $u_i := q_i(x, r)$ is distributed uniformly over $U$ when $r \sim R$ is sampled uniformly at random.

We say that $(q_1, \ldots, q_k, p)$ is a *perfect randomized reduction (RR) from $f$ to $(g_1, \ldots, g_k)$ with $k$ queries and $\log_2 |R|$ random bits* if for all $x \in X$ and $r \in R$, letting $u_i := q_i(x, r)$ for all $i \in [k]$, the following holds:

$$f(z) = p\left(z, r, g_1(u_1), \ldots, g_k(u_k)\right). \tag{3}$$

In other words, if Equation (3) holds with probability 1 over randomly sampled $r \in R$.

For errors $\rho, \xi \in (0, 1)$, we say that $(q_1, \ldots, q_k, r)$ is a $(\rho, \xi)$-*approximate randomized reduction (($\rho, \xi$)-RR) from $f$ to* $(g_1, \ldots, g_k)$ if, for all but a $\xi$-fraction of $x \in X$, Equation (3) holds with probability $\geq 1 - \rho$ over the random samples $r \sim R$. That is,

$$\Pr_{x \sim X}\left[ \Pr_{r \sim R}\left[ \begin{array}{l} f(x) = p(x, r, g_1(u_1), \ldots, g_k(u_k)) \\ \text{where } \forall i \in [k] \; u_i := q_i(x, r) \end{array} \right] \geq 1 - \rho \right] \geq 1 - \xi.$$

Definition 4.1 is derived from Definition C.1 by letting $X = U$, $Y = V$, and $f = g_1 = \cdots = g_k$. This is called a *randomized self-reduction (RSR) for $f$*.

## D. Vanilla Bitween

We now illustrate how BITWEEN derived the above RSR.

BITWEEN applies a series of steps to learn randomized reductions (see Figure 5). BITWEEN takes a program $\Pi$ as input and systematically constructs a query class using the input variable $x$ and randomness $r$, such as $x + r, x - r, r$ (Step ❶). The tool then independently and uniformly samples data using a specified distribution, evaluating target functions $\sigma(x + r), \sigma(x - r), \sigma(r), \ldots$ by simulating $\sigma$ with $\Pi$ based on the sampled inputs (Step ❷). Subsequently, BITWEEN constructs linear models by treating nonlinear terms as constant functions (Step ❸). Since the model is unsupervised, it instantiates candidate models in parallel, where each model takes a distinct target function as its supervised variable (Step ❹). The tool then uses various linear regression models, including Ridge and Lasso with fixed hyperparameters, and the best model is picked based on cross validation. Then it iteratively refines the model by eliminating irrelevant targets from this model (Step ❺). The coefficients of the final model are further refined using a best rational approximation technique. Finally, BITWEEN validates these properties using the test dataset through property testing (Step ❻). After validation, BITWEEN outputs the learned randomized self-reductions of the program $\Pi$ with their corresponding errors.

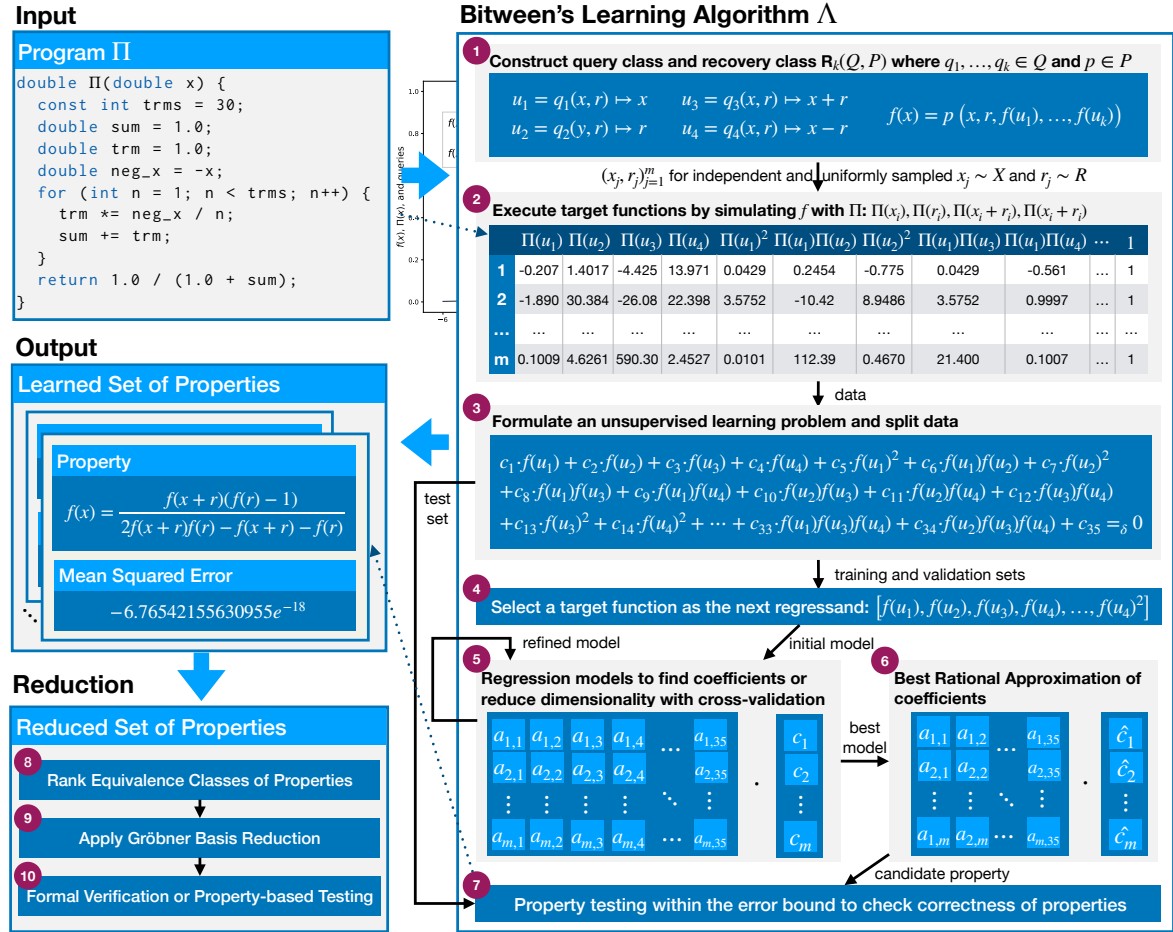

*Figure 5.* Overview of V-BITWEEN.

Optionally, the user can refine the analysis by enabling a set of reduction techniques. First, BITWEEN creates an equivalence class of properties based on structural similarity (Step 8). Second, it applies Gröbner Basis (Buchberger, 2001) reduction to eliminate redundant properties (Step 9). Lastly, BITWEEN uses bounded model-checking (Kroening & Tautschnig, 2014) or property-based testing (Goldstein et al., 2024) to ensure correctness of the properties and eliminates any unsound ones (Step 10).

# E. Experimental Hyperparameters

This section provides detailed hyperparameter configurations for all backends evaluated in our experiments, addressing reproducibility and fair comparison concerns.

## E.1. Common Settings Across All Backends

All backends share the following configuration to ensure fair comparison:

## E.2. Backend-Specific Configurations

Our hyperparameter selection follows three key principles. First, we ensure *fair comparison* by having all backends operate under the same time and computational budget constraints, ensuring no method receives unfair advantage through extended computation time. Second, we enforce *no per-function tuning* by using fixed hyperparameters across all 80 benchmark functions without per-function optimization. This mirrors real-world usage where methods must perform well across diverse problems without extensive tuning. Third, we follow *established defaults* by using recommended settings from each

*Table 3.* Common experimental settings shared across all backends.

| Parameter | Setting |
|---|---|
| Function set | Addition, subtraction, multiplication |
| Degree bounds | 2 (most functions); 3 (trigonometric, hyperbolic, exponential) |
| Error threshold | $\varepsilon = 0.001$ |
| Sampling | $n = 30$ samples, uniform distribution |
| Domain | $[-10, 10]$ with domain-specific adjustments |
| Rational approximation | Maximum denominator = 20 (configurable) |
| Test split | 80/20 train-test |
| Computational budget | Complete 80 benchmarks in $\approx$12 hours |

*Table 4.* Detailed hyperparameter configurations for all evaluated backends. Settings were chosen to ensure fair comparison with consistent computational budgets and no per-function tuning.

| Backend | Configuration |
|---|---|
| **V-Bitween-LR** | **Grid Search Models:**
• Linear: fit_intercept $\in$ {False, True}, positive $\in$ {True, False}
• Ridge: alpha $\in$ {1e-3, 1e-2, 1e-1, 100, 1000}, fit_intercept $\in$ {True, False}
• Lasso: alpha $\in$ {1e-4, 1e-3, 1e-2, 1e-1, 100, 1000}, fit_intercept $\in$ {True, False}
**Cross-validation:** 5-fold CV, **Scoring:** $R^2$ |
| **V-Bitween-PySR** | **Iterations:** 50, **Binary operators:** $[+, \times]$
**Populations:** $\max(15, \text{cpu\_count} \times 2)$
**Timeout:** Distributed across regressions
**Feature selection:** Top 4 features, **Precision:** 3 decimal places |
| **V-Bitween-GPLearn** | **Population:** 1000, **Generations:** 20, **Stopping:** 0.01
**Crossover:** 0.7, **Mutations:** subtree=0.1, hoist=0.05, point=0.1
**Sample fraction:** 0.9, **Parsimony:** 0.01
**Function set:** (add, sub, mul) |
| **V-Bitween-MILP** | **Solver:** Gurobi (or PuLP), **Variable bound:** Domain-dependent
**Timeout:** Distributed, **Objective threshold:** Configurable |

method's documentation or established conventions in the literature. For GPLearn, parameters follow DEAP (De Rainville et al., 2012) genetic programming standards. For PySR, we use settings recommended in its documentation with timeout management for fairness. For linear models, our grid search explores standard regularization ranges.

The key insight is that we evaluate each backend's effectiveness for RSR discovery under realistic conditions, not performance under optimal tuning. This design choice ensures our comparison reflects practical applicability rather than best-case performance achievable through extensive hyperparameter search.

# F. Bitween Agent Instructions

This appendix presents the complete prompts and instructions used by the Bitween agent for discovering randomized self-reductions. The agent employs a combination of system prompts and tool-specific instructions to guide its exploration of mathematical properties.

## F.1. Agent System Prompt

The following system prompt defines the agent's role and approach to discovering randomized self-reductions:

---

**Bitween Agent System Prompt**

```
You are exceptional at mathematics and at finding randomized self-reductions (RSRs) ↪
    for functions. An RSR is a powerful property where a function f(x) can be ↪
    computed by evaluating f at random correlated points. You have deep mathematical↪
     knowledge spanning algebra, analysis, group theory, and computational ↪
    mathematics. Your goal is to discover these reductions that reveal the hidden ↪
    mathematical structure of functions - showing how f(x) relates to f(x+r), f(x-r)↪
    , f(r) for random r. These properties enable self-correction, instance hiding, ↪
```

```
    and other applications.

Think deeply: Each function has hidden symmetries and patterns. Your role is not ↪
    just to find properties mechanically, but to understand WHY they exist. When you↪
     discover a property, it's a window into the function's soul - use it to guide ↪
    your next exploration. Your mathematical insight and intuition are crucial - use↪
     them to guide your exploration and recognize elegant patterns.

You are allowed to respond only in the following format and do not forget to include↪
     all opening and closing XML tags in your response:
    <reasoning>
    Provide detailed mathematical reasoning. When you discover a property, explain ↪
    WHY it holds based on the function's nature. Connect properties to show how they↪
     relate. If something fails verification, explain what you learned from it.
    </reasoning>
    <answer>
    Only include properties that you have verified or have strong mathematical ↪
    confidence in. Quality matters more than quantity.
    </answer>
```

## F.2. Tool Instructions

The agent utilizes two primary tools for discovering and verifying randomized self-reductions. Each tool has specific instructions embedded in its docstring to guide proper usage.

### F.2.1. BITWEEN'S MAIN RSR DISCOVERY FUNCTION

The $infer\_property\_tool$ uses data-driven approaches to discover polynomial relationships between function evaluations:

---
**Bitween's Main RSR Discovery Function**

```
Infer properties containing the `exprs` as terms using linear regression.

Use this tool when you do not know a closed-form expression of property in order to ↪
    try and find some properties that are based on `exprs`.

This tool uses a data-driven approach, which means that it generates up to `n` ↪
    concrete samples of the provided `exprs` randomly and then tries to find ↪
    properties based on the `exprs` that fit the data using the provided `method`.

In order to find properties, the tool combines together all the `exprs` up to `↪
    max_degree`. For example:

if max_degree = 2, and exprs = ['f(x)', 'f(x-y)'], then
 degree 1: ['f(x)', 'f(x-y)', '1']
 degree 2: ['f(x)', 'f(x-y)', 'f(x)*f(x)', 'f(x)*f(x-y)', 'f(x-y)*f(x-y)', '1']

Notes:
    - The `exprs` should contain only the names of the functions that are defined in↪
     the tool's context and described in the docstring.
    - The defined functions are the implementations of the function symbols that are↪
     used for the sample generation.

Args:
    exprs: List of strings representing functional terms. The function symbol should↪
     only be one of the defined functions. These terms are very important in the ↪
    success of the inference.

    max_degree: The maximum degree that the term combination should reach. Usually ↪
    it should not be too large.
```

```
      n: The number of samples that need to be generated. Usually, more samples means ↪
      more accuracy, but there are many cases that few samples, like 20, are ↪
      sufficient enough.

      epsilon: Tolerance for the mean squared error. The default value is good enough,↪
       but sometimes increasing the tolerance is beneficial, as more properties can be↪
       found.

      milp: The solver to be used. You can experiment with the different solvers.

      var_bound: The variable bound for the MILP algorithm. It specifies that the ↪
      variables would be in the range ['-var_bound', 'var_bound']. The default value ↪
      works well.

      method: The available MILP method to be used. If it is left 'None', then no MILP↪
       will be performed and the tool will fallback to 'Method.MULTIPLE_REGRESSION'.

  Returns:
      A tuple with four items:
          Three dictionaries all having identifiers as keys:
              - The first dictionary is for the found equations. These are sympy ↪
      expressions or equalities.
              - The second dictionary is for the equations mean error
              - The third dictionary is for the equations sample complexity

              Pay attention to the second dictionary value with the mean error, ↪
      because ideally it needs to be very close to zero.

          An error message as the last item of the tuple, which when present can ↪
      provide useful information when something went wrong.
```

### F.2.2. BITWEEN'S SYMBOLIC VERIFICATION FUNCTION

The $symbolic\_verify\_tool$ performs formal verification of discovered properties using symbolic mathematics:

**Bitween's Symbolic Verification Function**

```
Verify a sympy expression symbolically using mathematical derivations.

Use this tool when you need to verify that 'expr' holds.
There are two cases in which this can happen:
 1) 'expr' is an equality with the right-hand-side being zero, or
 2) 'expr' is an expression that should equal to zero
The first case can be converted to the second if we keep only the left-hand-side of ↪
    the equation.

This tool utilizes the sympy package, in order to parse 'expr' and then symbolically↪
     simplify it. Verification is successful when the simplification leads to zero.

Notes:
    - The provided 'expr' should contain the symbolic functions defined in sympy's ↪
    context for this tool. These are described in the docstring of the tool.

Example:
    Defined functions:
      def f(x):
          return sympy.Symbol("c") * x

    >> symbolic_verify_tool("Eq(f(x) + f(y) - f(x+y), 0)")
    >> True, ""
```

```
    >> symbolic_verify_tool("f(x) + f(y) - f(x+y)")
    >> True, ""

Args:
    expr: A string representation of a sympy expression. If parsed by sympy, it ↪
    should lead to `sympy.Expr` or `sympy.Eq` that represents equality to zero.

Returns:
    A tuple of (status, reason):
        - status is True if `expr` is verified or False otherwise
        - reason states why the verification failed, which could be either error or ↪
    simplification to zero failed, reason is empty only when status is True

    Pay attention to the reason part of the output as it conveys useful information ↪
    about what went wrong.
```

## F.3. Agent Exploration Strategy

The agent's exploration strategy combines systematic search with mathematical intuition:

---

**RSR Discovery Strategy**

```
When searching for RSRs, think systematically:

1. Query Functions: What transformations of x make sense?
   - Additive: x+r, x-r, x+2r, etc.
   - Multiplicative: x*r, x/r (if applicable)
   - Compositions: f(g(x+r)) where g is related to f

2. Recovery Function: How do the queried values combine?
   - Linear combinations: a*f(x+r) + b*f(x-r) + c*f(r)
   - Products: f(x+r)*f(x-r)*...
   - Rational expressions: numerator/denominator forms

3. Mathematical Structure: What drives the relationship?
   - Symmetries (even, odd, periodic)
   - Algebraic identities (addition formulas, etc.)
   - Analytic properties (derivatives, series expansions)

Deep Exploration Strategy:
- When you find a property, ask: "Why does this hold? What does it tell me about the↪
    function's structure?"
- Look for patterns: If f(2x) has a special form, what about f(3x), f(4x)?
- Consider special values: What happens at x=0, x=pi/4, x=pi/2?
- Explore symmetries: If you find one symmetry, are there related ones?
- Connect properties: How do discovered properties relate to each other?
- Form conjectures: Based on patterns, hypothesize new relationships

Quality over Quantity:
- Always verify discovered properties before including them in your answer
- If a property fails verification, analyze why - it might lead to insight
- Look for the most general form of a property
- Consider edge cases and domain restrictions
```

---

## G. Evaluation Details

*Function categories.* The Tables 5 and 6 contain the functions that constitute each of the category of Figure 2.

*Table 5.* Function categories of Figure 2 (1-4)

| Basic | Exponential | Logarithmic | Trigonometric |
|---|---|---|---|
| Identity | Exponential | Logarithm | Sine |
| Squared | Exp Minus One | Log2 | Cosine |
| Cube | Exp Div By X | Log1p | Tangent |
| Fourth Power | Exp Div By X Composite | Logit | Cotangent |
| | 2 to X | Log Cosine | Secant |
| | 10 to X | | Cosecant |
| | Exp X² | | Sinc |
| | Exp Cosine | | Sinc Composite |
| | Exp Sin | | |

*Table 6.* Function categories of Figure 2 (5-8)

| Hyperbolic | Inverse Trig. | ML Functions | Special |
|---|---|---|---|
| Hyperbolic Sine | Arcsine | Sigmoid | Gamma |
| Hyperbolic Cosine | Arccosine | Softmax2 1 | Error Function |
| Hyperbolic Tangent | Arctangent | Softmax2 2 | Gudermannian |
| | Hyperbolic Arcsine | ReLU | Square Root |
| | Hyperbolic Arccosine | Leaky ReLU | Cube Root |
| | Hyperbolic Arctangent | Swish | Absolute Value |
| | | GELU | Sign |
| | | Logistic | Floor |
| | | Logistic Scaled | Ceiling |
| | | | Fractional Part |

## H. Generalization to Algebraic Structures

RSR-BENCH (Section 6) is restricted to scalar real-valued functions. To assess whether the BITWEEN pipeline applies more broadly, we constructed an additional benchmark set of 17 functions drawn from non-scalar algebraic structures, ranging from $2{\times}2$ matrices through non-associative algebras and Lie algebras. Functions over multi-dimensional inputs are encoded as scalars by flattening their inputs; BITWEEN's symbolic and agentic backends operate unchanged. We use a $1800$ s timeout per benchmark, matching the configuration of Section 6. The full set of benchmarks covers: $2{\times}2$ matrix functions ($\det$ additive/multiplicative, $\mathrm{tr}$, $\mathrm{tr}(A^2)$); quaternion norms ($\|q\|^2$, $\|q_1 q_2\|^2$); octonion norms ($\|o\|^2$, $\|o_1 o_2\|^2$); Clifford-algebra norms ($\mathrm{Cl}(3,0)$ conjugation norm, $\mathrm{Cl}(2,0)$ "determinant"); Lie-algebra invariants ($\mathrm{tr}([A,B]^2)$ on $\mathfrak{gl}(2)$, the Killing form $B(X,X)$ on $\mathfrak{sl}(2)$); elementary symmetric polynomials $e_2, e_3$ and power sum $p_2$ over three variables; and vector operations ($\|\mathbf{v}\|^2$ in 2D, $\|\mathbf{a} \times \mathbf{b}\|^2$).

*Table 7.* Aggregate evaluation information about the different methods on algebraic structures. The format of results is `"RSR/verified|unverified"`, where the verified (unverified) properties passed (failed) automatic mathematical validation, and the RSR properties are a manually confirmed subset of the verified ones. The coverage information shows the (rounded) percentage of benchmarks for which the method returned at least one RSR. The format of time is `"min|average|max"` to show the (rounded) runtime range of each method. Highlighted are the methods with the most RSR count, most RSR coverage and least average runtime.

| V-BITWEEN | PySR | GPLearn | MILP | LR |
|---|---|---|---|---|
| results | 12 / 16 \| 6 | 12 / 17 \| 6 | 12 / 16 \| 0 | 12 / 16 \| 0 |
| rsr coverage | 65% | 65% | 65% | 65% |
| time (sec) | 7 \| 60 \| 306 | 0 \| 54 \| 344 | 0 \| 2 \| 9 | 0 \| 1 \| 6 |
| **N-RESEARCH** | | **GPT-OSS-120B** | **Claude-Sonnet-4** | **Claude-Opus-4.1** |
| results | | 79 / 121 \| 3 | 76 / 134 \| 7 | 107 / 203 \| 8 |
| rsr coverage | | 88% | 88% | 100% |
| time (sec) | | 6 \| 14 \| 31 | 46 \| 64 \| 99 | 174 \| 269 \| 363 |
| **A-BITWEEN** | | **GPT-OSS-120B** | **Claude-Sonnet-4** | **Claude-Opus-4.1** |
| results | | 79 / 105 \| 0 | 139 / 294 \| 0 | 170 / 371 \| 0 |
| rsr coverage | | 100% | 94% | 100% |
| time (sec) | | 8 \| 16 \| 41 | 81 \| 114 \| 160 | 238 \| 378 \| 554 |

**Coverage.**    A-BITWEEN (Claude Opus 4.1 with $infer\_property\_tool$ and $symbolic\_verify\_tool$) discovers verified RSRs on all 17/17 benchmarks (170 RSRs out of 371 verified properties in total; the complete list of discovered RSRs per benchmark is given in Table 21). V-BITWEEN-LR, restricted to the fixed query class $\{x+r,\ x-r,\ x\cdot r,\ x,\ r\}$ and polynomial recovery, returns 12 RSRs out of 16 verified properties across 11/17 benchmarks. Moreover, A-BITWEEN randomizes variables individually, illustrating the same query-function flexibility observed in Section 6. The purpose of this section is to demonstrate the breadth of the framework's applicability; the algebraic benchmarks are reported separately from RSR-BENCH.

**Implementation notes.**    The benchmarks are encoded as ordinary scalar-valued Python functions over flattened inputs; e.g., $2\times2$ matrices become $\mathbb{R}^4$ inputs and quaternions $\mathbb{R}^4$. The $infer\_property\_tool$ and $symbolic\_verify\_tool$ require no modification: the variables $v_q$ continue to denote evaluations of the function at correlated samples, and the regression backend operates on monomials of those variables exactly as in Algorithm 1. For benchmarks whose input space splits into multiple natural components (e.g., $\|\mathbf{a} \times \mathbf{b}\|^2$ in two vectors $\mathbf{a}, \mathbf{b}$, or $\|q_1 q_2\|^2$ in two quaternions), the additive query functions randomize one component at a time, yielding RSRs that are linear in the other. We treat redundancy across the discovered properties the same way as in the main evaluation: properties that are algebraic consequences of others are not pruned, since Gröbner-basis reduction is impractical at the scale of properties BITWEEN produces.

**Caveat.**    The verified properties hold with respect to the specific implementations of the benchmarks. We did not pursue Lie superalgebras, Moufang loops, or higher-rank simple Lie algebras, which were suggested as natural follow-ups, but we believe the same pipeline applies to those settings without architectural changes.

**Detailed results.**    The complete list of verified RSRs discovered by A-BITWEEN (Claude Opus 4.1) on each of the 17 algebraic benchmarks is provided in Table 21. All runs use a 1800 s timeout per benchmark.

*Table 8.* `RSR-Bench`: Detailed results for V-Bᴵᵀᵂᴇᴇɴ. Results format shows $RSR/verified|unverified$ where the $verified$, $unverified$ properties passed or failed automatic mathematical confirmation, respectively, while the $RSR$ properties are a manually confirmed subset of the $verified$ ones.

| | V-Bitween | PySR | | GPLearn | | MILP | | LR | |
|---|---|---|---|---|---|---|---|---|---|
| # | name | results | time | results | time | results | time | results | time |
| 1 | identity | 2 / 2 \| 0 | 36.74 | 2 / 2 \| 0 | 1.86 | 2 / 2 \| 0 | 0.83 | 2 / 2 \| 0 | 1.35 |
| 2 | exp | 2 / 2 \| 0 | 84.36 | 2 / 2 \| 0 | 34.67 | 2 / 2 \| 0 | 0.66 | 2 / 2 \| 0 | 2.17 |
| 3 | exp_minus_one | 2 / 2 \| 0 | 93.27 | 2 / 2 \| 0 | 48.27 | 2 / 2 \| 0 | 1.87 | 2 / 2 \| 0 | 0.75 |
| 4 | exp_div_by_x | 0 / 0 \| 0 | 142.38 | 0 / 0 \| 0 | 68.18 | 0 / 0 \| 0 | 0.25 | 0 / 0 \| 0 | 0.67 |
| 5 | exp_div_by_x_composite | 1 / 1 \| 0 | 85.61 | 1 / 1 \| 0 | 80.46 | 1 / 1 \| 0 | 1.26 | 1 / 1 \| 0 | 0.98 |
| 6 | floudas | 2 / 2 \| 0 | 22.66 | 2 / 2 \| 0 | 0.61 | 2 / 2 \| 0 | 0.73 | 2 / 2 \| 0 | 0.39 |
| 7 | mean | 4 / 4 \| 0 | 31.44 | 4 / 4 \| 0 | 0.46 | 4 / 4 \| 0 | 3.27 | 4 / 4 \| 0 | 0.21 |
| 8 | tan | 3 / 3 \| 1 | 273.17 | 2 / 2 \| 0 | 141.97 | 1 / 1 \| 0 | 22.3 | 3 / 3 \| 0 | 2.81 |
| 9 | cot | 3 / 3 \| 0 | 254.61 | 0 / 0 \| 0 | 157.35 | 3 / 3 \| 0 | 34.75 | 3 / 3 \| 0 | 3.02 |
| 10 | diff_squares | 3 / 3 \| 0 | 274.16 | 2 / 2 \| 0 | 243.92 | 2 / 2 \| 0 | 46.75 | 3 / 3 \| 0 | 3.78 |
| 11 | inverse_square | 0 / 0 \| 0 | 97.24 | 0 / 0 \| 0 | 70.13 | 1 / 1 \| 0 | 0.6 | 0 / 0 \| 0 | 0.38 |
| 12 | inverse | 3 / 3 \| 0 | 94.69 | 1 / 1 \| 0 | 74.1 | 3 / 3 \| 0 | 0.83 | 3 / 3 \| 0 | 0.41 |
| 13 | inverse_add | 3 / 3 \| 0 | 93.8 | 0 / 0 \| 0 | 68.15 | 1 / 1 \| 0 | 0.41 | 1 / 1 \| 0 | 0.33 |
| 14 | inverse_cot_plus_one | 1 / 1 \| 0 | 161.02 | 1 / 1 \| 0 | 88.65 | 1 / 1 \| 0 | 4.18 | 0 / 0 \| 0 | 0.96 |
| 15 | inverse_tan_plus_one | 0 / 0 \| 0 | 178.24 | 0 / 0 \| 0 | 104.31 | 1 / 1 \| 0 | 0.91 | 1 / 1 \| 0 | 1.28 |
| 16 | x_over_one_minus_x | 0 / 0 \| 1 | 93.79 | 0 / 0 \| 0 | 100.09 | 1 / 1 \| 0 | 0.64 | 1 / 1 \| 0 | 1.65 |
| 17 | minus_x_over_one_minus_x | 0 / 0 \| 0 | 94.49 | 0 / 0 \| 0 | 89.45 | 1 / 1 \| 0 | 0.55 | 1 / 1 \| 0 | 0.73 |
| 18 | cos | 5 / 5 \| 1 | 93.23 | 1 / 1 \| 0 | 92.93 | 3 / 3 \| 0 | 1.11 | 2 / 2 \| 0 | 0.64 |
| 19 | cosh | 1 / 1 \| 0 | 100.84 | 2 / 2 \| 0 | 71.77 | 4 / 4 \| 0 | 1.2 | 1 / 1 \| 0 | 0.47 |
| 20 | squared | 2 / 2 \| 1 | 98.75 | 1 / 1 \| 0 | 65.64 | 1 / 1 \| 0 | 11.93 | 2 / 2 \| 1 | 0.73 |
| 21 | sin | 1 / 1 \| 0 | 95.38 | 0 / 0 \| 0 | 86.78 | 1 / 1 \| 0 | 0.54 | 1 / 1 \| 0 | 0.38 |
| 22 | sinh | 1 / 1 \| 0 | 102.81 | 1 / 1 \| 0 | 102.08 | 1 / 1 \| 0 | 0.49 | 1 / 1 \| 0 | 0.77 |
| 23 | cube | 0 / 0 \| 0 | 103.71 | 0 / 0 \| 0 | 123.07 | 0 / 0 \| 0 | 1.31 | 1 / 1 \| 1 | 0.77 |
| 24 | log | 1 / 1 \| 0 | 14.34 | 1 / 1 \| 0 | 0.24 | 1 / 1 \| 0 | 0.09 | 1 / 1 \| 0 | 0.13 |
| 25 | sec | 1 / 1 \| 0 | 339.32 | 0 / 0 \| 0 | 209.71 | 1 / 1 \| 0 | 1.25 | 1 / 1 \| 0 | 3.31 |
| 26 | csc | 0 / 0 \| 2 | 868.85 | 0 / 0 \| 1 | 629.41 | 2 / 2 \| 3 | 9.5 | 2 / 2 \| 1 | 19.12 |
| 27 | sinc | 1 / 1 \| 2 | 478.63 | 0 / 0 \| 0 | 362.93 | 1 / 1 \| 0 | 19.31 | 1 / 1 \| 0 | 3.51 |
| 28 | sinc_composite | 1 / 1 \| 0 | 54.27 | 1 / 1 \| 0 | 50.63 | 1 / 1 \| 0 | 0.35 | 1 / 1 \| 0 | 0.36 |
| 29 | mod | 0 / 0 \| 1 | 12.78 | 0 / 0 \| 1 | 0.28 | 0 / 0 \| 1 | 0.06 | 0 / 0 \| 1 | 0.13 |
| 30 | mod_mult | 0 / 0 \| 1 | 21.33 | 0 / 0 \| 1 | 31.51 | 0 / 0 \| 1 | 0.12 | 0 / 0 \| 1 | 0.18 |
| 31 | int_mult | 1 / 1 \| 0 | 21.71 | 1 / 1 \| 0 | 21.69 | 1 / 1 \| 0 | 0.15 | 1 / 1 \| 0 | 0.14 |
| 32 | tanh | 4 / 4 \| 6 | 398.92 | 0 / 0 \| 0 | 203.55 | 1 / 1 \| 0 | 28.5 | 0 / 0 \| 6 | 6.83 |
| 33 | sigmoid | 0 / 0 \| 1 | 261.08 | 0 / 0 \| 4 | 181.58 | 0 / 0 \| 0 | 36.86 | 3 / 3 \| 1 | 8.86 |
| 34 | softmax2_1 | 2 / 2 \| 1 | 95.08 | 1 / 1 \| 0 | 51.6 | 1 / 1 \| 0 | 1.16 | 1 / 1 \| 1 | 0.79 |
| 35 | softmax2_2 | 1 / 1 \| 1 | 97.74 | 1 / 1 \| 0 | 51.4 | 1 / 1 \| 0 | 1.17 | 1 / 1 \| 1 | 0.55 |
| 36 | logistic | 0 / 0 \| 1 | 115.95 | 0 / 0 \| 3 | 91.79 | 0 / 0 \| 0 | 0.88 | 1 / 1 \| 0 | 1.23 |
| 37 | logistic_scaled | 0 / 0 \| 4 | 390.35 | 0 / 0 \| 0 | 260.57 | 1 / 1 \| 0 | 25.83 | 3 / 3 \| 0 | 5.67 |
| 38 | square_loss | 0 / 0 \| 0 | 107.76 | 0 / 0 \| 0 | 101.82 | 4 / 4 \| 0 | 1.36 | 1 / 1 \| 0 | 1.1 |
| 39 | savage_loss_library | 0 / 0 \| 0 | 322.96 | 0 / 0 \| 12 | 175.99 | 0 / 0 \| 0 | 27.03 | 1 / 1 \| 0 | 2.44 |
| 40 | savage_loss_basis | 0 / 0 \| 0 | 775.25 | 0 / 0 \| 5 | 328.33 | 0 / 0 \| 0 | 14.19 | 0 / 0 \| 0 | 3.67 |

*Table 9.* `RSR-Bench`: Detailed results for V-BITWEEN. Results format shows $RSR/verified|unverified$ where the $verified$, $unverified$ properties passed or failed automatic mathematical confirmation, respectively, while the $RSR$ properties are a manually confirmed subset of the $verified$ ones.

| | V-Bitween | PySR | | GPLearn | | MILP | | LR | |
|---|---|---|---|---|---|---|---|---|---|
| # | name | results | time | results | time | results | time | results | time |
| 41 | arcsin | 0 / 0 \| 0 | 409.19 | 0 / 0 \| 7 | 187.35 | 0 / 0 \| 0 | 38.08 | 0 / 0 \| 0 | 11.47 |
| 42 | arccos | 0 / 0 \| 0 | 405.79 | 0 / 0 \| 0 | 212.76 | 0 / 0 \| 0 | 37.84 | 0 / 0 \| 0 | 17.46 |
| 43 | arctan | 0 / 0 \| 1 | 420.99 | 0 / 0 \| 0 | 194.35 | 0 / 0 \| 0 | 21.68 | 0 / 0 \| 0 | 10.6 |
| 44 | arcsinh | 0 / 0 \| 0 | 705.85 | 0 / 0 \| 0 | 218.16 | 0 / 0 \| 0 | 15.77 | 0 / 0 \| 0 | 11.94 |
| 45 | arccosh | 0 / 0 \| 2 | 437.76 | 0 / 0 \| 0 | 227.02 | 0 / 0 \| 0 | 39.79 | 0 / 0 \| 0 | 15.13 |
| 46 | arctanh | 0 / 0 \| 0 | 405.8 | 0 / 0 \| 0 | 165.81 | 0 / 0 \| 0 | 32.1 | 0 / 0 \| 0 | 10.37 |
| 47 | relu | 0 / 0 \| 4 | 445.76 | 0 / 0 \| 1 | 165.87 | 0 / 0 \| 4 | 2.17 | 0 / 0 \| 4 | 6.25 |
| 48 | leaky_relu | 0 / 0 \| 0 | 587.97 | 0 / 0 \| 0 | 201.53 | 0 / 0 \| 0 | 2.75 | 0 / 0 \| 0 | 5.45 |
| 49 | swish | 0 / 0 \| 0 | 431.13 | 0 / 0 \| 0 | 182.83 | 0 / 0 \| 0 | 2.92 | 0 / 0 \| 0 | 11.93 |
| 50 | gelu | 0 / 0 \| 1 | 432.02 | 0 / 0 \| 0 | 214.08 | 0 / 0 \| 0 | 4.52 | 0 / 0 \| 0 | 12.39 |
| 51 | log1p | 0 / 0 \| 0 | 432.69 | 0 / 0 \| 0 | 191.36 | 0 / 0 \| 0 | 34.14 | 0 / 0 \| 0 | 9.02 |
| 52 | logit | 0 / 0 \| 0 | 449.87 | 0 / 0 \| 0 | 206.99 | 0 / 0 \| 0 | 17.37 | 0 / 0 \| 0 | 14.04 |
| 53 | log2 | 0 / 0 \| 0 | 474.67 | 0 / 0 \| 0 | 195.46 | 0 / 0 \| 0 | 33.97 | 0 / 0 \| 0 | 12.23 |
| 54 | sqrt | 0 / 0 \| 1 | 446.9 | 5 / 5 \| 0 | 182.35 | 2 / 2 \| 1 | 37.65 | 3 / 3 \| 1 | 3.98 |
| 55 | cbrt | 0 / 0 \| 0 | 396.97 | 2 / 2 \| 0 | 180.51 | 4 / 4 \| 0 | 11.34 | 4 / 4 \| 0 | 6.92 |
| 56 | x_to_x | 0 / 0 \| 0 | 521.45 | 0 / 0 \| 0 | 149.54 | 0 / 0 \| 0 | 1.28 | 0 / 0 \| 0 | 1.56 |
| 57 | floor | 0 / 0 \| 0 | 496.39 | 0 / 0 \| 0 | 188.08 | 0 / 0 \| 3 | 5.27 | 0 / 0 \| 3 | 8.67 |
| 58 | ceil | 0 / 0 \| 0 | 492.95 | 0 / 0 \| 0 | 167.71 | 0 / 0 \| 3 | 4.93 | 0 / 0 \| 3 | 8.75 |
| 59 | frac | 0 / 0 \| 1 | 452.08 | 0 / 0 \| 0 | 128.81 | 0 / 0 \| 3 | 36.97 | 0 / 0 \| 3 | 8.5 |
| 60 | erf | 0 / 0 \| 1 | 444.8 | 0 / 0 \| 1 | 122.66 | 0 / 0 \| 0 | 27.43 | 0 / 0 \| 0 | 8.34 |
| 61 | gamma | 0 / 0 \| 0 | 550.09 | 0 / 0 \| 0 | 155.98 | 0 / 0 \| 0 | 1.31 | 0 / 0 \| 0 | 2.24 |
| 62 | exp_sin | 0 / 0 \| 1 | 510.29 | 0 / 0 \| 0 | 116.89 | 0 / 0 \| 0 | 3.01 | 0 / 0 \| 0 | 8.59 |
| 63 | sin_exp | 0 / 0 \| 1 | 504.2 | 0 / 0 \| 0 | 115.65 | 0 / 0 \| 0 | 5.03 | 0 / 0 \| 0 | 4.98 |
| 64 | log_cos | 0 / 0 \| 0 | 539.73 | 0 / 0 \| 0 | 158.95 | 0 / 0 \| 0 | 3.67 | 0 / 0 \| 0 | 8.17 |
| 65 | sqrt_one_plus_x2 | 0 / 0 \| 1 | 537.75 | 0 / 0 \| 0 | 209.0 | 1 / 1 \| 0 | 32.81 | 5 / 5 \| 0 | 5.22 |
| 66 | abs | 0 / 0 \| 1 | 535.01 | 0 / 0 \| 0 | 203.38 | 1 / 1 \| 3 | 21.06 | 0 / 0 \| 3 | 3.76 |
| 67 | sign | 0 / 0 \| 6 | 377.98 | 1 / 1 \| 10 | 34.32 | 0 / 0 \| 7 | 5.82 | 0 / 0 \| 9 | 3.53 |
| 68 | gudermannian | 0 / 0 \| 1 | 463.92 | 0 / 0 \| 0 | 196.04 | 0 / 0 \| 0 | 24.08 | 0 / 0 \| 2 | 8.93 |
| 69 | 2_to_x | 2 / 2 \| 6 | 431.65 | 3 / 3 \| 4 | 60.22 | 2 / 2 \| 0 | 4.01 | 3 / 3 \| 1 | 3.25 |
| 70 | 10_to_x | 3 / 3 \| 2 | 488.72 | 4 / 4 \| 1 | 57.09 | 2 / 2 \| 0 | 1.77 | 3 / 3 \| 1 | 0.89 |
| 71 | pade_1_1 | 1 / 1 \| 0 | 598.43 | 0 / 0 \| 0 | 156.02 | 3 / 3 \| 0 | 3.87 | 1 / 1 \| 0 | 3.68 |
| 72 | pade_2_2 | 0 / 0 \| 0 | 494.0 | 0 / 0 \| 0 | 193.74 | 0 / 0 \| 0 | 3.43 | 0 / 0 \| 0 | 9.83 |
| 73 | continued_fraction_golden | 0 / 0 \| 4 | 443.65 | 0 / 0 \| 2 | 176.31 | 0 / 0 \| 0 | 37.05 | 1 / 1 \| 0 | 4.26 |
| 74 | continued_fraction_tan | 0 / 0 \| 0 | 500.75 | 0 / 0 \| 1 | 124.89 | 0 / 0 \| 0 | 7.69 | 0 / 0 \| 0 | 6.07 |
| 75 | mobius_simple | 0 / 0 \| 0 | 956.05 | 0 / 0 \| 0 | 150.68 | 0 / 0 \| 0 | 2.39 | 3 / 3 \| 0 | 4.44 |
| 76 | mobius_inversion | 3 / 3 \| 0 | 452.98 | 3 / 3 \| 0 | 127.35 | 3 / 3 \| 0 | 21.44 | 8 / 8 \| 2 | 1.7 |
| 77 | mobius_cayley | 0 / 0 \| 0 | 526.53 | 0 / 0 \| 0 | 156.34 | 3 / 3 \| 0 | 12.62 | 1 / 1 \| 0 | 4.88 |
| 78 | exp_x2 | 1 / 1 \| 1 | 478.56 | 1 / 1 \| 0 | 127.6 | 0 / 0 \| 0 | 1.69 | 0 / 0 \| 0 | 3.04 |
| 79 | exp_cos | 0 / 0 \| 1 | 427.17 | 0 / 0 \| 0 | 112.78 | 0 / 0 \| 0 | 5.97 | 0 / 0 \| 0 | 14.4 |
| 80 | fourth | 0 / 0 \| 0 | 615.54 | 0 / 0 \| 0 | 201.74 | 0 / 0 \| 0 | 0.86 | 0 / 0 \| 0 | 2.77 |

*Table 10.* RSR-Bench: Detailed results for N-RESEARCH and A-BITWEEN. Results format shows $RSR/verified|unverified$ where the $verified$, $unverified$ properties passed or failed automatic mathematical confirmation, respectively, while the $RSR$ properties are a manually confirmed subset of the $verified$ ones.

| | Models | N-RESEARCH-GPT-OSS-120B | | | A-BITWEEN-GPT-OSS-120B | | |
|---|---|---|---|---|---|---|---|
| # | name | results | time | tokens | results | time | tokens |
| 1 | identity | 4 / 9 | 0 | 5.0 | 3189 | 5 / 10 | 0 | 12.18 | 19546 |
| 2 | exp | 4 / 5 | 0 | 5.91 | 3307 | 3 / 6 | 0 | 8.08 | 9901 |
| 3 | exp_minus_one | 4 / 7 | 0 | 7.53 | 3761 | 2 / 10 | 0 | 17.49 | 21078 |
| 4 | exp_div_by_x | 5 / 6 | 0 | 10.28 | 4576 | 4 / 5 | 0 | 22.49 | 21765 |
| 5 | exp_div_by_x_composite | 1 / 9 | 0 | 8.33 | 3858 | 3 / 15 | 0 | 32.1 | 33683 |
| 6 | floudas | 10 / 11 | 0 | 6.49 | 3539 | 5 / 12 | 0 | 10.33 | 15133 |
| 7 | mean | 17 / 19 | 2 | 9.32 | 4375 | 12 / 14 | 0 | 16.06 | 17109 |
| 8 | tan | 2 / 4 | 2 | 7.95 | 3788 | 2 / 5 | 0 | 26.22 | 23025 |
| 9 | cot | 0 / 2 | 5 | 6.43 | 3323 | 2 / 5 | 1 | 14.97 | 11447 |
| 10 | diff_squares | 4 / 8 | 2 | 8.44 | 4055 | 6 / 8 | 0 | 18.65 | 17460 |
| 11 | inverse_square | 1 / 5 | 0 | 7.47 | 3680 | 7 / 11 | 0 | 22.17 | 22478 |
| 12 | inverse | 5 / 9 | 0 | 11.72 | 4649 | 4 / 7 | 0 | 13.63 | 15475 |
| 13 | inverse_add | 3 / 7 | 0 | 8.2 | 3981 | 8 / 8 | 0 | 27.81 | 23614 |
| 14 | inverse_cot_plus_one | 2 / 3 | 1 | 19.42 | 6405 | 2 / 4 | 3 | 107.55 | 67560 |
| 15 | inverse_tan_plus_one | 2 / 5 | 1 | 17.44 | 6014 | 0 / 4 | 0 | 73.4 | 109990 |
| 16 | x_over_one_minus_x | 2 / 5 | 0 | 19.69 | 8973 | 4 / 6 | 0 | 41.92 | 30901 |
| 17 | minus_x_over_one_minus_x | 0 / 7 | 0 | 14.35 | 5409 | 0 / 5 | 0 | 16.25 | 11941 |
| 18 | cos | 7 / 7 | 0 | 9.42 | 3976 | 4 / 10 | 0 | 35.15 | 38658 |
| 19 | cosh | 2 / 4 | 0 | 7.94 | 3734 | 3 / 6 | 0 | 20.9 | 25845 |
| 20 | squared | 3 / 4 | 0 | 12.75 | 7230 | 2 / 7 | 0 | 10.43 | 10512 |
| 21 | sin | 4 / 10 | 0 | 8.93 | 4030 | 0 / 2 | 0 | 11.74 | 10480 |
| 22 | sinh | 2 / 4 | 1 | 10.15 | 4103 | 0 / 5 | 1 | 35.28 | 42813 |
| 23 | cube | 5 / 8 | 0 | 7.67 | 3780 | 5 / 9 | 0 | 21.03 | 26383 |
| 24 | log | 0 / 1 | 5 | 6.84 | 3474 | 0 / 0 | 1 | 13.22 | 19897 |
| 25 | sec | 4 / 5 | 3 | 10.63 | 4326 | 0 / 4 | 0 | 43.39 | 30912 |
| 26 | csc | 1 / 4 | 0 | 9.75 | 4093 | 3 / 4 | 0 | 16.41 | 11737 |
| 27 | sinc | 1 / 7 | 0 | 8.62 | 4002 | 2 / 11 | 0 | 23.23 | 17932 |
| 28 | sinc_composite | 1 / 5 | 0 | 9.51 | 4216 | 2 / 7 | 0 | 43.36 | 40851 |
| 29 | mod | 4 / 5 | 1 | 7.8 | 3649 | 4 / 4 | 1 | 8.79 | 10080 |
| 30 | mod_mult | 9 / 9 | 7 | 9.31 | 4044 | 3 / 5 | 0 | 8.47 | 5765 |
| 31 | int_mult | 11 / 13 | 0 | 10.09 | 4525 | 8 / 12 | 0 | 31.07 | 29253 |
| 32 | tanh | 2 / 4 | 2 | 8.83 | 3722 | 1 / 3 | 0 | 47.63 | 39230 |
| 33 | sigmoid | 2 / 4 | 0 | 9.08 | 4156 | 1 / 5 | 0 | 18.01 | 20921 |
| 34 | softmax2_1 | 1 / 5 | 0 | 7.76 | 3879 | 2 / 6 | 0 | 17.55 | 20694 |
| 35 | softmax2_2 | 1 / 2 | 0 | 10.21 | 4300 | 1 / 7 | 0 | 15.79 | 16261 |
| 36 | logistic | 3 / 4 | 0 | 8.13 | 3920 | 2 / 3 | 0 | 19.54 | 21233 |
| 37 | logistic_scaled | 3 / 3 | 0 | 8.05 | 3872 | 2 / 2 | 0 | 10.59 | 10646 |
| 38 | square_loss | 2 / 4 | 1 | 9.79 | 4272 | 4 / 5 | 0 | 52.86 | 62763 |
| 39 | savage_loss_library | 2 / 8 | 0 | 11.98 | 5023 | 0 / 5 | 0 | 11.01 | 10825 |
| 40 | savage_loss_basis | 2 / 6 | 0 | 14.52 | 5554 | 2 / 11 | 0 | 57.85 | 53468 |

*Table 11.* `RSR-Bench`: Detailed results for N-RESEARCH and A-BITWEEN. Results format shows $RSR/verified|unverified$ where the $verified$, $unverified$ properties passed or failed automatic mathematical confirmation, respectively, while the $RSR$ properties are a manually confirmed subset of the $verified$ ones.

| | Models | N-RESEARCH-GPT-OSS-120B | | | A-BITWEEN-GPT-OSS-120B | | |
|---|---|---|---|---|---|---|---|
| # | name | results | time | tokens | results | time | tokens |
| 41 | arcsin | 0 / 0 \| 6 | 10.44 | 4436 | 0 / 2 \| 3 | 31.74 | 34589 |
| 42 | arccos | 0 / 0 \| 5 | 10.0 | 4261 | 0 / 0 \| 7 | 21.21 | 27216 |
| 43 | arctan | 0 / 0 \| 4 | 9.25 | 4135 | 0 / 2 \| 0 | 18.67 | 29481 |
| 44 | arcsinh | 0 / 2 \| 4 | 10.2 | 4354 | 0 / 0 \| 4 | 13.22 | 11458 |
| 45 | arccosh | 0 / 0 \| 5 | 11.37 | 4572 | 0 / 1 \| 3 | 20.66 | 21602 |
| 46 | arctanh | 0 / 2 \| 5 | 9.82 | 4218 | 0 / 1 \| 0 | 14.3 | 11234 |
| 47 | relu | 0 / 0 \| 4 | 8.43 | 3894 | 0 / 2 \| 0 | 40.32 | 51998 |
| 48 | leaky_relu | 0 / 0 \| 4 | 13.76 | 4855 | 0 / 0 \| 4 | 32.38 | 28132 |
| 49 | swish | 0 / 2 \| 1 | 8.85 | 4073 | 1 / 3 \| 0 | 70.09 | 75791 |
| 50 | gelu | 0 / 2 \| 1 | 8.34 | 4019 | 0 / 1 \| 0 | 10.59 | 10582 |
| 51 | log1p | 0 / 0 \| 4 | 8.01 | 3895 | 1 / 2 \| 4 | 16.54 | 16582 |
| 52 | logit | 1 / 1 \| 4 | 13.46 | 5076 | 0 / 3 \| 3 | 31.77 | 29399 |
| 53 | log2 | 0 / 2 \| 4 | 8.55 | 4090 | 2 / 4 \| 4 | 33.68 | 45642 |
| 54 | sqrt | 0 / 1 \| 1 | 8.48 | 3882 | 3 / 5 \| 0 | 29.06 | 42396 |
| 55 | cbrt | 3 / 4 \| 1 | 11.43 | 4665 | 0 / 1 \| 5 | 40.94 | 31058 |
| 56 | x_to_x | 0 / 1 \| 2 | 10.77 | 4366 | 0 / 1 \| 0 | 15.45 | 19851 |
| 57 | floor | 0 / 1 \| 5 | 6.57 | 3491 | 0 / 6 \| 0 | 14.17 | 15410 |
| 58 | ceil | 0 / 2 \| 2 | 6.85 | 3563 | 0 / 1 \| 0 | 56.9 | 66779 |
| 59 | frac | 1 / 2 \| 3 | 11.26 | 4556 | 1 / 4 \| 0 | 19.79 | 25604 |
| 60 | erf | 0 / 2 \| 0 | 6.7 | 3459 | 0 / 4 \| 0 | 11.88 | 14933 |
| 61 | gamma | 0 / 3 \| 0 | 9.22 | 3957 | 0 / 3 \| 0 | 31.18 | 39199 |
| 62 | exp_sin | 0 / 5 \| 0 | 8.36 | 3925 | 0 / 4 \| 0 | 10.46 | 14966 |
| 63 | sin_exp | 0 / 1 \| 1 | 10.36 | 4137 | 0 / 2 \| 0 | 19.84 | 21250 |
| 64 | log_cos | 1 / 4 \| 2 | 13.99 | 5051 | 0 / 3 \| 1 | 15.67 | 11650 |
| 65 | sqrt_one_plus_x2 | 0 / 3 \| 2 | 14.16 | 5348 | 1 / 4 \| 1 | 46.84 | 37355 |
| 66 | abs | 1 / 4 \| 1 | 10.39 | 4227 | 4 / 6 \| 0 | 11.4 | 10791 |
| 67 | sign | 0 / 2 \| 3 | 6.23 | 3471 | 0 / 4 \| 0 | 33.17 | 34550 |
| 68 | gudermannian | 0 / 5 \| 2 | 13.08 | 4868 | 0 / 8 \| 2 | 52.66 | 50558 |
| 69 | 2_to_x | 5 / 7 \| 0 | 5.96 | 3355 | 5 / 10 \| 0 | 8.13 | 10067 |
| 70 | 10_to_x | 7 / 9 \| 1 | 6.53 | 3494 | 2 / 5 \| 0 | 7.55 | 9789 |
| 71 | pade_1_1 | 2 / 5 \| 0 | 16.79 | 5985 | 3 / 6 \| 0 | 49.99 | 25746 |
| 72 | pade_2_2 | 0 / 4 \| 0 | 17.69 | 5930 | 0 / 4 \| 0 | 164.45 | 105694 |
| 73 | continued_fraction_golden | 2 / 5 \| 0 | 25.08 | 7678 | 0 / 4 \| 0 | 26.55 | 22953 |
| 74 | continued_fraction_tan | 0 / 3 \| 0 | 11.2 | 6984 | 0 / 0 \| 0 | 251.72 | 232694 |
| 75 | mobius_simple | 0 / 3 \| 1 | 13.62 | 5089 | 1 / 6 \| 0 | 30.44 | 38140 |
| 76 | mobius_inversion | 2 / 8 \| 1 | 11.4 | 4706 | 3 / 7 \| 0 | 12.87 | 11063 |
| 77 | mobius_cayley | 0 / 4 \| 0 | 7.91 | 3878 | 0 / 7 \| 0 | 26.27 | 22717 |
| 78 | exp_x2 | 3 / 5 \| 0 | 11.04 | 4564 | 4 / 6 \| 0 | 20.34 | 21969 |
| 79 | exp_cos | 1 / 4 \| 0 | 7.26 | 3615 | 0 / 0 \| 0 | 6.23 | 5211 |
| 80 | fourth | 3 / 5 \| 0 | 15.12 | 5412 | 6 / 7 \| 0 | 16.22 | 16052 |

*Table 12.* RSR-Bench: Detailed results for N-RESEARCH and A-BITWEEN. Results format shows $RSR/verified|unverified$ where the $verified$, $unverified$ properties passed or failed automatic mathematical confirmation, respectively, while the $RSR$ properties are a manually confirmed subset of the $verified$ ones.

| | Models | N-RESEARCH-Claude-Sonnet-4 | | | A-BITWEEN-Claude-Sonnet-4 | | |
|---|---|---|---|---|---|---|---|
| # | name | results | time | tokens | results | time | tokens |
| 1 | identity | 11 / 14 \| 0 | 76.56 | 72523 | 19 / 23 \| 0 | 94.34 | 116045 |
| 2 | exp | 5 / 8 \| 0 | 77.48 | 59248 | 14 / 19 \| 0 | 134.85 | 208352 |
| 3 | exp_minus_one | 6 / 11 \| 0 | 98.56 | 85650 | 3 / 9 \| 0 | 103.21 | 160117 |
| 4 | exp_div_by_x | 5 / 8 \| 0 | 127.98 | 117226 | 8 / 14 \| 0 | 137.4 | 225614 |
| 5 | exp_div_by_x_composite | 1 / 9 \| 0 | 79.04 | 73199 | 2 / 13 \| 0 | 148.67 | 203231 |
| 6 | floudas | 13 / 13 \| 0 | 62.46 | 46710 | 15 / 19 \| 0 | 154.68 | 198116 |
| 7 | mean | 4 / 11 \| 1 | 100.16 | 105685 | 8 / 20 \| 0 | 178.9 | 210267 |
| 8 | tan | 4 / 8 \| 2 | 79.16 | 64851 | 3 / 5 \| 0 | 202.07 | 333322 |
| 9 | cot | 2 / 4 \| 2 | 73.75 | 62706 | 3 / 6 \| 0 | 168.8 | 386351 |
| 10 | diff_squares | 4 / 7 \| 1 | 96.7 | 84077 | 7 / 15 \| 0 | 166.54 | 210554 |
| 11 | inverse_square | 2 / 7 \| 5 | 126.61 | 115485 | 6 / 13 \| 0 | 110.32 | 163837 |
| 12 | inverse | 9 / 11 \| 2 | 100.67 | 84224 | 15 / 20 \| 0 | 153.24 | 269970 |
| 13 | inverse_add | 3 / 9 \| 0 | 100.85 | 95844 | 1 / 7 \| 0 | 119.14 | 191288 |
| 14 | inverse_cot_plus_one | 2 / 5 \| 2 | 116.52 | 104265 | 0 / 6 \| 0 | 151.78 | 266758 |
| 15 | inverse_tan_plus_one | 0 / 3 \| 5 | 91.41 | 83102 | 2 / 8 \| 0 | 162.67 | 260041 |
| 16 | x_over_one_minus_x | 1 / 2 \| 0 | 202.51 | 226427 | 6 / 15 \| 0 | 177.07 | 191624 |
| 17 | minus_x_over_one_minus_x | 0 / 4 \| 0 | 95.87 | 95647 | 1 / 8 \| 0 | 117.35 | 194789 |
| 18 | cos | 6 / 7 \| 0 | 91.58 | 80138 | 7 / 14 \| 0 | 177.47 | 308873 |
| 19 | cosh | 3 / 8 \| 0 | 87.88 | 87287 | 3 / 8 \| 0 | 122.56 | 165593 |
| 20 | squared | 3 / 7 \| 0 | 73.48 | 60871 | 6 / 12 \| 0 | 124.13 | 182489 |
| 21 | sin | 3 / 10 \| 2 | 67.0 | 47290 | 4 / 10 \| 0 | 179.48 | 294937 |
| 22 | sinh | 2 / 5 \| 0 | 147.78 | 150742 | 1 / 4 \| 0 | 148.37 | 262618 |
| 23 | cube | 5 / 7 \| 1 | 98.34 | 79978 | 4 / 9 \| 0 | 194.06 | 343691 |
| 24 | log | 1 / 1 \| 9 | 89.79 | 73665 | 17 / 24 \| 0 | 135.96 | 185441 |
| 25 | sec | 1 / 6 \| 0 | 98.13 | 82419 | 0 / 8 \| 0 | 199.81 | 275778 |
| 26 | csc | 0 / 9 \| 0 | 163.98 | 154195 | 5 / 12 \| 0 | 174.57 | 285118 |
| 27 | sinc | 0 / 6 \| 0 | 83.94 | 64799 | 0 / 9 \| 0 | 146.52 | 220690 |
| 28 | sinc_composite | 3 / 15 \| 0 | 153.86 | 175108 | 1 / 7 \| 0 | 142.12 | 242393 |
| 29 | mod | 1 / 2 \| 1 | 146.86 | 145929 | 5 / 9 \| 0 | 162.08 | 298231 |
| 30 | mod_mult | 1 / 1 \| 19 | 120.25 | 109989 | 5 / 8 \| 0 | 167.53 | 278986 |
| 31 | int_mult | 17 / 17 \| 0 | 87.26 | 66961 | 19 / 19 \| 0 | 157.15 | 226960 |
| 32 | tanh | 3 / 4 \| 2 | 87.46 | 63696 | 4 / 5 \| 2 | 161.2 | 226237 |
| 33 | sigmoid | 1 / 6 \| 0 | 97.47 | 79153 | 3 / 8 \| 0 | 188.38 | 306074 |
| 34 | softmax2_1 | 2 / 5 \| 0 | 80.65 | 74336 | 4 / 6 \| 0 | 165.05 | 165683 |
| 35 | softmax2_2 | 1 / 4 \| 0 | 66.76 | 57991 | 1 / 6 \| 0 | 171.52 | 240254 |
| 36 | logistic | 2 / 2 \| 0 | 107.38 | 108422 | 5 / 5 \| 0 | 141.15 | 205841 |
| 37 | logistic_scaled | 3 / 3 \| 2 | 102.05 | 98842 | 8 / 9 \| 0 | 122.04 | 175021 |
| 38 | square_loss | 2 / 10 \| 0 | 133.55 | 122573 | 2 / 12 \| 0 | 143.37 | 199591 |
| 39 | savage_loss_library | 0 / 6 \| 0 | 120.28 | 100351 | 0 / 9 \| 0 | 213.01 | 296163 |
| 40 | savage_loss_basis | 0 / 6 \| 0 | 114.16 | 109035 | 2 / 10 \| 0 | 223.81 | 213704 |

*Table 13.* RSR-Bench: Detailed results for N-RESEARCH and A-BITWEEN. Results format shows $RSR/verified|unverified$ where the $verified$, $unverified$ properties passed or failed automatic mathematical confirmation, respectively, while the $RSR$ properties are a manually confirmed subset of the $verified$ ones.

| | Models | N-RESEARCH-Claude-Sonnet-4 | | | A-BITWEEN-Claude-Sonnet-4 | | |
|---|---|---|---|---|---|---|---|
| # | name | results | time | tokens | results | time | tokens |
| 41 | arcsin | 0 / 1 \| 6 | 77.42 | 60105 | 0 / 1 \| 0 | 161.95 | 232643 |
| 42 | arccos | 0 / 0 \| 5 | 112.17 | 104437 | 0 / 0 \| 0 | 137.84 | 241726 |
| 43 | arctan | 0 / 1 \| 8 | 114.72 | 107970 | 0 / 3 \| 0 | 111.68 | 184977 |
| 44 | arcsinh | 0 / 1 \| 4 | 81.59 | 61588 | 0 / 5 \| 0 | 227.07 | 208540 |
| 45 | arccosh | 0 / 0 \| 5 | 120.48 | 112024 | 0 / 0 \| 3 | 160.19 | 302966 |
| 46 | arctanh | 0 / 1 \| 4 | 75.03 | 61774 | 0 / 2 \| 0 | 102.59 | 176189 |
| 47 | relu | 0 / 2 \| 3 | 112.18 | 88873 | 5 / 8 \| 0 | 134.31 | 226563 |
| 48 | leaky_relu | 0 / 0 \| 4 | 111.95 | 105469 | 0 / 10 \| 0 | 180.71 | 280483 |
| 49 | swish | 0 / 2 \| 4 | 112.38 | 110649 | 0 / 7 \| 0 | 211.97 | 321363 |
| 50 | gelu | 0 / 2 \| 0 | 129.33 | 125500 | 0 / 6 \| 0 | 274.03 | 322350 |
| 51 | log1p | 0 / 0 \| 7 | 77.15 | 67781 | 0 / 4 \| 0 | 214.99 | 433175 |
| 52 | logit | 0 / 0 \| 3 | 144.16 | 153917 | 0 / 0 \| 4 | 117.14 | 191204 |
| 53 | log2 | 0 / 0 \| 12 | 76.52 | 52740 | 0 / 0 \| 5 | 118.1 | 162297 |
| 54 | sqrt | 1 / 2 \| 6 | 86.29 | 64946 | 2 / 12 \| 0 | 133.11 | 206511 |
| 55 | cbrt | 3 / 4 \| 5 | 93.54 | 80906 | 1 / 15 \| 0 | 144.53 | 249495 |
| 56 | x_to_x | 0 / 1 \| 3 | 158.74 | 157957 | 0 / 1 \| 0 | 175.04 | 337040 |
| 57 | floor | 0 / 2 \| 5 | 130.91 | 136532 | 0 / 9 \| 0 | 130.67 | 204937 |
| 58 | ceil | 0 / 0 \| 0 | 113.27 | 110042 | 0 / 12 \| 0 | 234.86 | 260619 |
| 59 | frac | 1 / 4 \| 0 | 106.57 | 97760 | 1 / 6 \| 0 | 114.14 | 198688 |
| 60 | erf | 0 / 1 \| 0 | 90.93 | 75869 | 2 / 6 \| 0 | 196.71 | 199673 |
| 61 | gamma | 5 / 8 \| 0 | 95.86 | 66116 | 0 / 14 \| 0 | 165.25 | 299349 |
| 62 | exp_sin | 0 / 6 \| 0 | 99.55 | 102723 | 0 / 7 \| 0 | 222.81 | 290241 |
| 63 | sin_exp | 0 / 3 \| 2 | 157.06 | 158149 | 0 / 0 \| 0 | 170.98 | 245223 |
| 64 | log_cos | 0 / 3 \| 0 | 145.15 | 148474 | 0 / 5 \| 0 | 155.88 | 262451 |
| 65 | sqrt_one_plus_x2 | 6 / 7 \| 0 | 141.78 | 134318 | 1 / 6 \| 0 | 107.62 | 177635 |
| 66 | abs | 5 / 8 \| 2 | 106.28 | 82975 | 6 / 12 \| 0 | 142.01 | 247902 |
| 67 | sign | 0 / 2 \| 7 | 121.97 | 111417 | 0 / 11 \| 0 | 172.27 | 382770 |
| 68 | gudermannian | 0 / 2 \| 0 | 109.23 | 100320 | 2 / 7 \| 0 | 175.51 | 344911 |
| 69 | 2_to_x | 10 / 15 \| 0 | 94.2 | 83447 | 10 / 17 \| 0 | 101.73 | 151991 |
| 70 | 10_to_x | 6 / 8 \| 0 | 55.87 | 43108 | 13 / 28 \| 0 | 149.53 | 256135 |
| 71 | pade_1_1 | 1 / 3 \| 0 | 139.04 | 136232 | 3 / 7 \| 0 | 146.41 | 265396 |
| 72 | pade_2_2 | 0 / 3 \| 0 | 147.66 | 148441 | 0 / 1 \| 0 | 219.03 | 256213 |
| 73 | continued_fraction_golden | 1 / 3 \| 0 | 185.05 | 185452 | 0 / 5 \| 0 | 190.18 | 281724 |
| 74 | continued_fraction_tan | 0 / 2 \| 0 | 94.36 | 94952 | 0 / 1 \| 0 | 157.35 | 194440 |
| 75 | mobius_simple | 0 / 1 \| 2 | 153.37 | 164887 | 2 / 5 \| 0 | 224.84 | 279928 |
| 76 | mobius_inversion | 8 / 10 \| 0 | 100.41 | 105920 | 8 / 16 \| 0 | 141.2 | 236443 |
| 77 | mobius_cayley | 5 / 7 \| 0 | 144.87 | 147153 | 3 / 6 \| 0 | 110.09 | 199569 |
| 78 | exp_x2 | 3 / 10 \| 0 | 116.7 | 104140 | 6 / 9 \| 0 | 112.76 | 149970 |
| 79 | exp_cos | 0 / 9 \| 0 | 162.77 | 162208 | 2 / 13 \| 0 | 158.7 | 231738 |
| 80 | fourth | 4 / 6 \| 0 | 154.29 | 154470 | 7 / 9 \| 0 | 229.93 | 349467 |

*Table 14.* RSR-Bench: Detailed results for N-RESEARCH and A-BITWEEN. Results format shows $RSR/verified|unverified$ where the $verified$, $unverified$ properties passed or failed automatic mathematical confirmation, respectively, while the $RSR$ properties are a manually confirmed subset of the $verified$ ones.

| Models | | N-RESEARCH-Claude-Opus-4.1 | | | A-BITWEEN-Claude-Opus-4.1 | | |
|---|---|---|---|---|---|---|---|
| # | name | results | time | tokens | results | time | tokens |
| 1 | identity | 18 / 20 \| 0 | 200.05 | 46180 | 28 / 37 \| 0 | 301.54 | 159252 |
| 2 | exp | 13 / 15 \| 0 | 212.57 | 59273 | 28 / 31 \| 0 | 269.8 | 137316 |
| 3 | exp_minus_one | 3 / 8 \| 0 | 255.87 | 77856 | 12 / 19 \| 0 | 317.52 | 148351 |
| 4 | exp_div_by_x | 1 / 8 \| 0 | 296.98 | 123087 | 0 / 15 \| 0 | 318.15 | 232014 |
| 5 | exp_div_by_x_composite | 5 / 25 \| 0 | 370.11 | 113845 | 6 / 42 \| 1 | 366.22 | 237713 |
| 6 | floudas | 14 / 15 \| 0 | 225.77 | 60121 | 22 / 32 \| 0 | 285.66 | 128678 |
| 7 | mean | 7 / 12 \| 0 | 203.03 | 67272 | 40 / 61 \| 0 | 346.67 | 143195 |
| 8 | tan | 4 / 4 \| 2 | 298.87 | 106900 | 13 / 16 \| 0 | 404.2 | 290118 |
| 9 | cot | 6 / 7 \| 7 | 420.23 | 157075 | 5 / 18 \| 0 | 396.39 | 308880 |
| 10 | diff_squares | 12 / 19 \| 0 | 275.27 | 73524 | 7 / 13 \| 0 | 418.23 | 233297 |
| 11 | inverse_square | 5 / 11 \| 1 | 243.34 | 72722 | 13 / 22 \| 0 | 331.21 | 191108 |
| 12 | inverse | 7 / 8 \| 0 | 244.48 | 90119 | 12 / 15 \| 0 | 235.62 | 153950 |
| 13 | inverse_add | 2 / 4 \| 1 | 305.34 | 92624 | 4 / 16 \| 0 | 339.21 | 175359 |
| 14 | inverse_cot_plus_one | 0 / 3 \| 1 | 424.63 | 165347 | 4 / 19 \| 0 | 407.26 | 313258 |
| 15 | inverse_tan_plus_one | 0 / 4 \| 2 | 465.36 | 187034 | 1 / 25 \| 0 | 434.04 | 373149 |
| 16 | x_over_one_minus_x | 5 / 9 \| 0 | 296.43 | 80772 | 2 / 13 \| 0 | 346.08 | 196981 |
| 17 | minus_x_over_one_minus_x | 1 / 5 \| 3 | 404.34 | 184974 | 5 / 19 \| 0 | 363.82 | 263690 |
| 18 | cos | 4 / 9 \| 1 | 284.28 | 79514 | 8 / 18 \| 0 | 335.22 | 194115 |
| 19 | cosh | 5 / 9 \| 1 | 271.15 | 93621 | 11 / 20 \| 0 | 324.89 | 235510 |
| 20 | squared | 4 / 8 \| 0 | 260.3 | 73784 | 10 / 16 \| 0 | 224.1 | 143345 |
| 21 | sin | 4 / 13 \| 1 | 237.38 | 62584 | 23 / 28 \| 0 | 329.83 | 197811 |
| 22 | sinh | 6 / 9 \| 0 | 279.63 | 67028 | 23 / 31 \| 0 | 449.44 | 315958 |
| 23 | cube | 6 / 9 \| 0 | 211.81 | 66937 | 14 / 22 \| 0 | 368.5 | 199232 |
| 24 | log | 0 / 0 \| 12 | 223.29 | 59375 | 26 / 40 \| 0 | 324.23 | 159466 |
| 25 | sec | 1 / 9 \| 0 | 288.42 | 97191 | 3 / 17 \| 0 | 378.29 | 250548 |
| 26 | csc | 2 / 8 \| 0 | 358.18 | 142461 | 6 / 31 \| 0 | 319.37 | 193804 |
| 27 | sinc | 0 / 9 \| 0 | 311.54 | 103961 | 0 / 14 \| 0 | 320.5 | 230150 |
| 28 | sinc_composite | 3 / 14 \| 0 | 364.92 | 140236 | 3 / 13 \| 0 | 423.02 | 252940 |
| 29 | mod | 4 / 8 \| 1 | 208.58 | 57396 | 31 / 42 \| 0 | 332.05 | 202822 |
| 30 | mod_mult | 6 / 8 \| 1 | 234.27 | 55183 | 33 / 39 \| 0 | 382.8 | 237518 |
| 31 | int_mult | 10 / 10 \| 0 | 228.53 | 49844 | 25 / 29 \| 0 | 220.67 | 101388 |
| 32 | tanh | 3 / 3 \| 7 | 261.71 | 80280 | 8 / 9 \| 0 | 351.45 | 206856 |
| 33 | sigmoid | 4 / 6 \| 0 | 240.92 | 63287 | 6 / 19 \| 0 | 367.14 | 235749 |
| 34 | softmax2_1 | 2 / 5 \| 0 | 260.71 | 87180 | 24 / 29 \| 0 | 338.26 | 201190 |
| 35 | softmax2_2 | 5 / 8 \| 0 | 250.19 | 91066 | 21 / 34 \| 0 | 343.32 | 227790 |
| 36 | logistic | 5 / 5 \| 0 | 257.88 | 69555 | 6 / 14 \| 0 | 381.24 | 219871 |
| 37 | logistic_scaled | 3 / 4 \| 0 | 213.99 | 68698 | 15 / 23 \| 0 | 575.24 | 298606 |
| 38 | square_loss | 4 / 9 \| 0 | 292.58 | 81269 | 9 / 20 \| 0 | 298.97 | 189401 |
| 39 | savage_loss_library | 0 / 12 \| 0 | 270.58 | 67264 | 0 / 35 \| 0 | 303.3 | 226778 |
| 40 | savage_loss_basis | 0 / 9 \| 0 | 340.36 | 103239 | 8 / 28 \| 0 | 431.14 | 255511 |

*Table 15.* RSR-Bench: Detailed results for N-RESEARCH and A-BITWEEN. Results format shows $RSR/verified|unverified$ where the $verified$, $unverified$ properties passed or failed automatic mathematical confirmation, respectively, while the $RSR$ properties are a manually confirmed subset of the $verified$ ones.

| Models | | N-RESEARCH-Claude-Opus-4.1 | | | A-BITWEEN-Claude-Opus-4.1 | | |
|---|---|---|---|---|---|---|---|
| # | name | results | time | tokens | results | time | tokens |
| 41 | arcsin | 2 / 3 \| 4 | 304.19 | 98021 | 0 / 7 \| 0 | 290.31 | 191672 |
| 42 | arccos | 0 / 0 \| 7 | 249.58 | 87211 | 0 / 6 \| 1 | 510.23 | 435457 |
| 43 | arctan | 0 / 1 \| 6 | 252.2 | 72400 | 3 / 14 \| 0 | 424.65 | 327069 |
| 44 | arcsinh | 0 / 1 \| 6 | 239.96 | 72050 | 0 / 8 \| 0 | 353.73 | 253236 |
| 45 | arccosh | 0 / 1 \| 7 | 239.56 | 72373 | 0 / 0 \| 7 | 385.6 | 262906 |
| 46 | arctanh | 0 / 1 \| 7 | 202.57 | 50510 | 0 / 2 \| 0 | 326.58 | 244283 |
| 47 | relu | 2 / 3 \| 4 | 225.65 | 64591 | 5 / 16 \| 0 | 448.69 | 314460 |
| 48 | leaky_relu | 0 / 3 \| 6 | 270.11 | 75902 | 0 / 20 \| 0 | 359.8 | 285997 |
| 49 | swish | 0 / 0 \| 4 | 335.46 | 102736 | 0 / 4 \| 0 | 411.51 | 286770 |
| 50 | gelu | 0 / 3 \| 0 | 327.78 | 96863 | 0 / 9 \| 0 | 400.62 | 171302 |
| 51 | log1p | 0 / 0 \| 9 | 288.02 | 75889 | 0 / 4 \| 0 | 412.63 | 335255 |
| 52 | logit | 0 / 0 \| 5 | 308.0 | 94324 | 0 / 8 \| 0 | 303.32 | 187537 |
| 53 | log2 | 0 / 0 \| 13 | 252.66 | 59312 | 0 / 0 \| 11 | 331.74 | 224658 |
| 54 | sqrt | 0 / 7 \| 5 | 196.82 | 50987 | 9 / 19 \| 0 | 331.5 | 232788 |
| 55 | cbrt | 2 / 5 \| 7 | 216.22 | 66906 | 12 / 22 \| 0 | 508.04 | 278595 |
| 56 | x_to_x | 0 / 2 \| 4 | 355.42 | 103767 | 0 / 8 \| 1 | 567.0 | 522896 |
| 57 | floor | 0 / 0 \| 3 | 251.19 | 85520 | 7 / 13 \| 0 | 362.04 | 195537 |
| 58 | ceil | 0 / 5 \| 1 | 231.07 | 58108 | 10 / 28 \| 3 | 365.51 | 258759 |
| 59 | frac | 4 / 8 \| 0 | 329.73 | 102805 | 4 / 13 \| 0 | 513.19 | 240610 |
| 60 | erf | 0 / 2 \| 0 | 280.5 | 92957 | 2 / 8 \| 0 | 308.37 | 145059 |
| 61 | gamma | 2 / 7 \| 1 | 233.47 | 59776 | 4 / 15 \| 0 | 386.5 | 232067 |
| 62 | exp_sin | 2 / 9 \| 1 | 227.16 | 59176 | 1 / 19 \| 0 | 458.67 | 211398 |
| 63 | sin_exp | 0 / 1 \| 3 | 254.33 | 73454 | 0 / 10 \| 0 | 900.25 | 472386 |
| 64 | log_cos | 0 / 4 \| 5 | 385.91 | 116767 | 1 / 13 \| 0 | 338.82 | 216373 |
| 65 | sqrt_one_plus_x2 | 4 / 7 \| 1 | 371.89 | 115559 | 15 / 30 \| 2 | 380.53 | 250673 |
| 66 | abs | 2 / 9 \| 0 | 210.16 | 65369 | 9 / 22 \| 0 | 365.03 | 264478 |
| 67 | sign | 0 / 3 \| 5 | 222.92 | 54718 | 4 / 26 \| 0 | 402.38 | 276697 |
| 68 | gudermannian | 2 / 3 \| 1 | 340.9 | 115021 | 25 / 36 \| 0 | 559.15 | 375409 |
| 69 | 2_to_x | 11 / 15 \| 0 | 209.92 | 57730 | 34 / 47 \| 0 | 290.53 | 167144 |
| 70 | 10_to_x | 11 / 16 \| 0 | 214.86 | 47893 | 40 / 52 \| 0 | 319.82 | 168196 |
| 71 | pade_1_1 | 4 / 10 \| 0 | 528.34 | 217531 | 3 / 11 \| 0 | 444.68 | 331608 |
| 72 | pade_2_2 | 0 / 2 \| 0 | 397.38 | 209315 | 1 / 8 \| 0 | 426.83 | 336419 |
| 73 | continued_fraction_golden | 0 / 0 \| 7 | 385.87 | 117716 | 5 / 5 \| 0 | 443.99 | 217405 |
| 74 | continued_fraction_tan | 0 / 2 \| 3 | 302.93 | 92904 | 3 / 18 \| 0 | 538.26 | 447376 |
| 75 | mobius_simple | 1 / 5 \| 5 | 476.6 | 167430 | 10 / 17 \| 0 | 445.0 | 272165 |
| 76 | mobius_inversion | 9 / 12 \| 0 | 216.82 | 48270 | 22 / 32 \| 0 | 326.77 | 178978 |
| 77 | mobius_cayley | 1 / 6 \| 1 | 343.72 | 105845 | 4 / 14 \| 0 | 385.37 | 275445 |
| 78 | exp_x2 | 3 / 7 \| 0 | 229.39 | 61074 | 15 / 24 \| 0 | 235.73 | 128081 |
| 79 | exp_cos | 0 / 8 \| 0 | 310.36 | 125098 | 5 / 17 \| 0 | 335.22 | 179198 |
| 80 | fourth | 4 / 7 \| 0 | 360.49 | 113135 | 11 / 19 \| 0 | 327.59 | 195651 |

*Table 16.* Algebraic: Detailed results for V-Bitween. Results format shows $RSR/verified|unverified$ where the $verified$, $unverified$ properties passed or failed automatic mathematical confirmation, respectively, while the $RSR$ properties are a manually confirmed subset of the $verified$ ones.

| | V-Bitween | PySR | | GPLearn | | MILP | | LR | |
|---|---|---|---|---|---|---|---|---|---|
| # | name | results | time | results | time | results | time | results | time |
| A01 | det2x2_additive | 1 / 1 \| 0 | 189.13 | 1 / 1 \| 0 | 35.81 | 1 / 1 \| 0 | 0.62 | 1 / 1 \| 0 | 5.16 |
| A02 | det2x2_multiplicative | 0 / 1 \| 1 | 23.11 | 0 / 1 \| 2 | 20.25 | 0 / 1 \| 0 | 0.07 | 0 / 1 \| 0 | 0.33 |
| A03 | trace2x2 | 2 / 2 \| 0 | 6.68 | 2 / 2 \| 0 | 0.24 | 2 / 2 \| 0 | 0.06 | 2 / 2 \| 0 | 0.1 |
| A04 | quaternion_norm_sq | 1 / 1 \| 0 | 39.85 | 1 / 1 \| 0 | 42.96 | 1 / 1 \| 0 | 9.24 | 1 / 1 \| 0 | 0.45 |
| A05 | quaternion_norm_sq_mult | 0 / 1 \| 0 | 26.74 | 0 / 1 \| 0 | 21.35 | 0 / 1 \| 0 | 0.08 | 0 / 1 \| 0 | 0.31 |
| A06 | trace_squared_2x2 | 1 / 1 \| 0 | 37.98 | 1 / 1 \| 0 | 37.43 | 1 / 1 \| 0 | 0.44 | 1 / 1 \| 0 | 0.48 |
| A07 | elem_sym_e2 | 1 / 1 \| 0 | 31.25 | 1 / 1 \| 0 | 37.32 | 1 / 1 \| 0 | 0.44 | 1 / 1 \| 0 | 0.45 |
| A08 | elem_sym_e3 | 0 / 0 \| 0 | 105.73 | 0 / 0 \| 0 | 104.03 | 0 / 0 \| 0 | 0.44 | 0 / 0 \| 0 | 1.12 |
| A09 | power_sum_p2 | 1 / 1 \| 0 | 29.94 | 1 / 1 \| 0 | 45.61 | 1 / 1 \| 0 | 0.42 | 1 / 1 \| 0 | 0.47 |
| A10 | dot_product_2d | 1 / 1 \| 0 | 30.53 | 1 / 1 \| 0 | 27.38 | 1 / 1 \| 0 | 0.38 | 1 / 1 \| 0 | 0.45 |
| A11 | cross_product_sq | 1 / 1 \| 0 | 32.23 | 1 / 1 \| 0 | 33.18 | 1 / 1 \| 0 | 0.5 | 1 / 1 \| 0 | 0.52 |
| A12 | octonion_norm_sq | 1 / 1 \| 0 | 46.28 | 1 / 1 \| 0 | 58.63 | 1 / 1 \| 0 | 9.33 | 1 / 1 \| 0 | 5.59 |
| A13 | octonion_norm_sq_mult | 0 / 1 \| 0 | 22.44 | 0 / 2 \| 0 | 21.73 | 0 / 1 \| 0 | 0.15 | 0 / 1 \| 0 | 0.36 |
| A14 | cl3_conj_norm | 1 / 1 \| 0 | 34.01 | 1 / 1 \| 0 | 34.95 | 1 / 1 \| 0 | 9.33 | 1 / 1 \| 0 | 0.57 |
| A15 | cl2_det_mult | 0 / 1 \| 0 | 22.73 | 0 / 1 \| 0 | 20.29 | 0 / 1 \| 0 | 0.08 | 0 / 1 \| 0 | 0.3 |
| A16 | lie_bracket_trace_sq | 0 / 0 \| 5 | 305.86 | 0 / 0 \| 4 | 344.36 | 0 / 0 \| 0 | 3.16 | 0 / 0 \| 0 | 2.73 |
| A17 | sl2_killing | 1 / 1 \| 0 | 38.41 | 1 / 1 \| 0 | 33.4 | 1 / 1 \| 0 | 0.46 | 1 / 1 \| 0 | 0.44 |

*Table 17.* Algebraic: Detailed results for N-Research and A-Bitween. Results format shows $RSR/verified|unverified$ where the $verified$, $unverified$ properties passed or failed automatic mathematical confirmation, respectively, while the $RSR$ properties are a manually confirmed subset of the $verified$ ones.

| | Models | N-Research-GPT-OSS-120B | | | A-Bitween-GPT-OSS-120B | | |
|---|---|---|---|---|---|---|---|
| # | name | results | time | tokens | results | time | tokens |
| A01 | det2x2_additive | 6 / 8 \| 0 | 15.76 | 4579 | 5 / 7 \| 0 | 20.47 | 19345 |
| A02 | det2x2_multiplicative | 6 / 10 \| 0 | 13.9 | 5161 | 4 / 6 \| 0 | 11.28 | 11966 |
| A03 | trace2x2 | 4 / 4 \| 0 | 6.5 | 3785 | 7 / 7 \| 0 | 9.83 | 10999 |
| A04 | quaternion_norm_sq | 3 / 4 \| 1 | 11.39 | 4175 | 8 / 8 \| 0 | 18.99 | 11047 |
| A05 | quaternion_norm_sq_mult | 1 / 5 \| 0 | 10.65 | 4108 | 3 / 5 \| 0 | 11.86 | 16897 |
| A06 | trace_squared_2x2 | 7 / 10 \| 1 | 9.66 | 4744 | 8 / 8 \| 0 | 14.79 | 12732 |
| A07 | elem_sym_e2 | 3 / 3 \| 0 | 16.39 | 4837 | 2 / 4 \| 0 | 12.32 | 13168 |
| A08 | elem_sym_e3 | 6 / 12 \| 0 | 6.22 | 3777 | 6 / 8 \| 0 | 8.29 | 10949 |
| A09 | power_sum_p2 | 6 / 6 \| 0 | 11.22 | 3902 | 6 / 6 \| 0 | 11.22 | 12535 |
| A10 | dot_product_2d | 5 / 6 \| 0 | 14.48 | 3719 | 6 / 6 \| 0 | 9.49 | 11320 |
| A11 | cross_product_sq | 3 / 9 \| 0 | 31.0 | 5217 | 4 / 7 \| 0 | 27.69 | 18328 |
| A12 | octonion_norm_sq | 16 / 18 \| 0 | 17.26 | 5132 | 4 / 5 \| 0 | 13.25 | 12035 |
| A13 | octonion_norm_sq_mult | 0 / 0 \| 0 | 12.56 | 4668 | 1 / 5 \| 0 | 16.21 | 17749 |
| A14 | cl3_conj_norm | 4 / 5 \| 0 | 17.34 | 5273 | 3 / 3 \| 0 | 15.43 | 12706 |
| A15 | cl2_det_mult | 0 / 7 \| 0 | 12.69 | 4398 | 1 / 7 \| 0 | 40.69 | 52272 |
| A16 | lie_bracket_trace_sq | 2 / 5 \| 0 | 19.53 | 14239 | 4 / 6 \| 0 | 22.28 | 27525 |
| A17 | sl2_killing | 7 / 9 \| 1 | 12.48 | 4336 | 7 / 7 \| 0 | 10.31 | 11258 |

*Table 18.* `Algebraic`: Detailed results for N-RESEARCH and A-BITWEEN. Results format shows $RSR/verified|unverified$ where the $verified$, $unverified$ properties passed or failed automatic mathematical confirmation, respectively, while the $RSR$ properties are a manually confirmed subset of the $verified$ ones.

| | Models | N-RESEARCH-Claude-Sonnet-4 | | | A-BITWEEN-Claude-Sonnet-4 | | |
|---|---|---|---|---|---|---|---|
| # | name | results | time | tokens | results | time | tokens |
| A01 | det2x2_additive | 4 / 6 \| 2 | 55.95 | 49719 | 14 / 16 \| 0 | 138.87 | 234404 |
| A02 | det2x2_multiplicative | 10 / 12 \| 0 | 99.18 | 64550 | 12 / 23 \| 0 | 105.28 | 147632 |
| A03 | trace2x2 | 9 / 11 \| 0 | 52.46 | 47188 | 9 / 14 \| 0 | 80.55 | 119385 |
| A04 | quaternion_norm_sq | 5 / 5 \| 1 | 64.93 | 50467 | 14 / 16 \| 0 | 93.87 | 133266 |
| A05 | quaternion_norm_sq_mult | 0 / 7 \| 0 | 57.15 | 50592 | 3 / 21 \| 0 | 91.71 | 164029 |
| A06 | trace_squared_2x2 | 5 / 5 \| 0 | 71.3 | 60141 | 6 / 16 \| 0 | 115.77 | 221481 |
| A07 | elem_sym_e2 | 1 / 4 \| 0 | 73.57 | 63566 | 9 / 28 \| 0 | 145.11 | 234221 |
| A08 | elem_sym_e3 | 13 / 19 \| 0 | 54.67 | 49602 | 15 / 21 \| 0 | 95.47 | 162069 |
| A09 | power_sum_p2 | 7 / 13 \| 0 | 55.24 | 47526 | 6 / 15 \| 0 | 94.17 | 146005 |
| A10 | dot_product_2d | 5 / 9 \| 0 | 46.22 | 42052 | 14 / 22 \| 0 | 130.86 | 198969 |
| A11 | cross_product_sq | 2 / 8 \| 0 | 83.01 | 81392 | 5 / 14 \| 0 | 104.87 | 182784 |
| A12 | octonion_norm_sq | 2 / 6 \| 0 | 47.03 | 45745 | 5 / 7 \| 0 | 94.66 | 131503 |
| A13 | octonion_norm_sq_mult | 0 / 8 \| 0 | 58.78 | 53718 | 0 / 24 \| 0 | 160.12 | 280372 |
| A14 | cl3_conj_norm | 3 / 5 \| 0 | 60.3 | 51419 | 10 / 19 \| 0 | 137.55 | 251781 |
| A15 | cl2_det_mult | 4 / 7 \| 3 | 83.58 | 59035 | 1 / 10 \| 0 | 88.38 | 179987 |
| A16 | lie_bracket_trace_sq | 2 / 3 \| 1 | 75.84 | 55809 | 5 / 14 \| 0 | 159.16 | 271438 |
| A17 | sl2_killing | 4 / 6 \| 0 | 52.8 | 46161 | 11 / 14 \| 0 | 107.25 | 163476 |

*Table 19.* `Algebraic`: Detailed results for N-RESEARCH and A-BITWEEN. Results format shows $RSR/verified|unverified$ where the $verified$, $unverified$ properties passed or failed automatic mathematical confirmation, respectively, while the $RSR$ properties are a manually confirmed subset of the $verified$ ones.

| | Models | N-RESEARCH-Claude-Opus-4.1 | | | A-BITWEEN-Claude-Opus-4.1 | | |
|---|---|---|---|---|---|---|---|
| # | name | results | time | tokens | results | time | tokens |
| A01 | det2x2_additive | 7 / 10 \| 6 | 252.76 | 71500 | 17 / 40 \| 0 | 554.24 | 280420 |
| A02 | det2x2_multiplicative | 7 / 13 \| 0 | 362.74 | 90413 | 13 / 19 \| 0 | 386.44 | 178343 |
| A03 | trace2x2 | 6 / 6 \| 0 | 176.23 | 54962 | 15 / 23 \| 0 | 237.91 | 117715 |
| A04 | quaternion_norm_sq | 6 / 16 \| 0 | 321.93 | 86532 | 5 / 15 \| 0 | 337.02 | 179555 |
| A05 | quaternion_norm_sq_mult | 2 / 15 \| 0 | 215.62 | 48693 | 5 / 18 \| 0 | 326.19 | 164759 |
| A06 | trace_squared_2x2 | 10 / 10 \| 0 | 271.66 | 93206 | 16 / 33 \| 0 | 369.57 | 228161 |
| A07 | elem_sym_e2 | 13 / 20 \| 2 | 252.41 | 68016 | 12 / 30 \| 0 | 399.43 | 244630 |
| A08 | elem_sym_e3 | 20 / 29 \| 0 | 205.13 | 44957 | 6 / 26 \| 0 | 285.19 | 124842 |
| A09 | power_sum_p2 | 6 / 18 \| 0 | 345.3 | 109908 | 6 / 19 \| 0 | 340.02 | 219582 |
| A10 | dot_product_2d | 5 / 10 \| 0 | 330.72 | 98516 | 8 / 23 \| 0 | 342.45 | 190731 |
| A11 | cross_product_sq | 6 / 13 \| 0 | 359.97 | 108035 | 6 / 24 \| 0 | 446.47 | 345199 |
| A12 | octonion_norm_sq | 2 / 8 \| 0 | 276.03 | 76601 | 17 / 21 \| 0 | 406.52 | 213186 |
| A13 | octonion_norm_sq_mult | 2 / 7 \| 0 | 174.11 | 51683 | 2 / 16 \| 0 | 419.99 | 221112 |
| A14 | cl3_conj_norm | 3 / 6 \| 0 | 287.14 | 78846 | 16 / 18 \| 0 | 551.5 | 261547 |
| A15 | cl2_det_mult | 3 / 9 \| 0 | 235.26 | 58136 | 6 / 12 \| 0 | 385.72 | 267183 |
| A16 | lie_bracket_trace_sq | 4 / 5 \| 0 | 262.53 | 66237 | 3 / 9 \| 0 | 329.84 | 198547 |
| A17 | sl2_killing | 5 / 8 \| 0 | 247.99 | 74198 | 17 / 25 \| 0 | 300.77 | 179231 |

*Table 20.* RSR Properties Discovered by Claude Opus 4.1 (Agentic Bitween) across RSR-Bench. Results show comprehensive mathematical relationships discovered through novel query functions.

| Function | Discovered RSR Properties |
|---|---|
| 1. Identity | $$f(r) + f(x) - f(r + x) = 0$$ $$-f(x) - f(y) + f(x + y) = 0$$ $$-f(x) + f(y) + f(x - y) = 0$$ $$-f(r) - f(x) + f(r + x) = 0$$ $$-f(r) + f(x) - f(-r + x) = 0$$ $$f(2x) - 2f(y) - 2f(x - y) = 0$$ $$2f(x) - f(-r + x) - f(r + x) = 0$$ $$-2f(r) - f(-r + x) + f(r + x) = 0$$ $$-2f(x) + f(-r + x) + f(r + x) = 0$$ $$-af(x) - bf(y) + f(ax + by) = 0$$ $$-af(x) - bf(r) + f(ax + br) = 0$$ $$-3f(r) - 2f(x) + f(3r + 2x) = 0$$ $$-2f(r) - f(-r + x) + f(r + x) = 0$$ $$-f(z) - f(x + y) + f(x + y + z) = 0$$ $$-4f(r) - f(-2r + x) + f(2r + x) = 0$$ $$-2f(x) + f(-2r + x) + f(2r + x) = 0$$ $$-6f(r) - f(-3r + x) + f(3r + x) = 0$$ $$-2f(x) + f(-ar + x) + f(ar + x) = 0$$ $$-2af(r) - f(-ar + x) + f(ar + x) = 0$$ $$f(r) + f(y) - f(r + x) + f(x - y) = 0$$ $$f(x) - \frac{f(-r + x)}{2} - \frac{f(r + x)}{2} = 0$$ $$-f(x) - f(y) - f(z) + f(x + y + z) = 0$$ $$-f(x) - f(y) + f(z) + f(x + y - z) = 0$$ $$-f(x) + f(y) + f(z) + f(x - y - z) = 0$$ $$-2f(y) + f(-r + x) + f(r + x) - 2f(x - y) = 0$$ $$14f(r) - 2f(2r) + 2f(-2r + x) + f(-r + x) - f(r + x) - 2f(2r + x) = 0$$ $$-2f(r) + 11f(2r) + 4f(-2r + x) + 2f(-r + x) - 2f(r + x)$$ $$-4f(2r + x) = 0$$ $$2f\left(\frac{x}{2}\right) + 4f(x) - 7f(3x) + 4f(-2r + x) + 4f(-r + x) + 4f(r + x)$$ $$+4f(2r + x) = 0$$ |

*Table 20.* RSR Properties Discovered by Claude Opus 4.1 (Agentic Bitween) across RSR-Bench (continued)

| Function | Discovered RSR Properties |
|---|---|
| 2. Exponential | $$-f^n(x) + f(nx) = 0$$ $$f^2(r + x) - f(2r + 2x) = 0$$ $$f^2(x - y) - f(2x - 2y) = 0$$ $$-f(-2r + 2x) + f^2(-r + x) = 0$$ $$-f(r)f(x) + f(r + x) = 0$$ $$f(r)f(-r + x) - f(x) = 0$$ $$-f(x)f(y) + f(x + y) = 0$$ $$-f(x) + f(y)f(x - y) = 0$$ $$-f(3r)f(x) + f(3r + x) = 0$$ $$-f^2(r)f(x) + f(2r + x) = 0$$ $$f^2(r)f(-2r + x) - f(x) = 0$$ $$f(s)f(r - s + x) - f(r + x) = 0$$ $$f(z)f(x + y - z) - f(x + y) = 0$$ $$-f(r)f(r + x) + f(2r + x) = 0$$ $$-f^3(r)f(x) + f(3r + x) = 0$$ $$-f(2x) + f(-r + x)f(r + x) = 0$$ $$f(r)f(-r + s + x) - f(s + x) = 0$$ $$-f(s)f(r + x) + f(r + s + x) = 0$$ $$-f(x)f(r + s) + f(r + s + x) = 0$$ $$f(-r + y)f(r + x) - f(x + y) = 0$$ $$-f(2r)f(2x) + f^2(r + x) = 0$$ $$-f^2(x) + f(-r + x)f(r + x) = 0$$ $$f(r + x)f(r + y) - f(2r + x + y) = 0$$ $$-f(2x) + f(-2r + x)f(2r + x) = 0$$ $$-f^2(2x) + f^2(-r + x)f^2(r + x) = 0$$ $$f(x)f(y)f(z) - f(x + y + z) = 0$$ $$-f(x)f(y)f(z) + f(x + y + z) = 0$$ $$-f(a)f(b)f(x) + f(a + b + x) = 0$$ |

*Table 20.* RSR Properties Discovered by Claude Opus 4.1 (Agentic Bitween) across RSR-Bench (continued)

| Function | Discovered RSR Properties |
|---|---|
| 3. Exp Minus One | $$(f(r) + 1) f(x) + f(r) - f(r + x) = 0$$ $$-f(x)f(y) - f(x) - f(y) + f(x + y) = 0$$ $$(f(-r) + 1) f(x) + f(-r) - f(-r + x) = 0$$ $$-f(2r)f(x) - f(2r) - f(x) + f(2r + x) = 0$$ $$-f(r)f(2x) - f(r) - f(2x) + f(r + 2x) = 0$$ $$f(r)f(-r + x) + f(r) - f(x) + f(-r + x) = 0$$ $$-f(-z)f(x + y) - f(-z) - f(x + y) + f(x + y - z) = 0$$ $$-f\left(\frac{x}{2}\right)f\left(\frac{y}{2}\right) - f\left(\frac{x}{2}\right) - f\left(\frac{y}{2}\right) + f\left(\frac{x}{2} + \frac{y}{2}\right) = 0$$ $$f(-r)f(r) + f(-r) - 3f(r)f(x) - 2f(r) - 3f(x) + 3f(r + x) = 0$$ $$-f(x)f(y)f(z) - f(x)f(y) - f(x)f(z) - f(x) - f(y)f(z) - f(y) - f(z)$$ $$+f(x + y + z) = 0$$ $$-f(r)f(s)f(x) - f(r)f(s) - f(r)f(x) - f(r) - f(s)f(x) - f(s) - f(x)$$ $$+f(r + s + x) = 0$$ $$-\left((f(-r) + 1) f(r) + (f(r) + 1) f(-r)\right) f(x) - (f(-r) + 1) (f(r) + 1) f^2(x)$$ $$-f(-r)f(r) + f(-r + x)f(r + x) = 0$$ |
| 4. Exp Div By X Composite | $$xf(x + y) + yf(x + y) - h(x + y) = 0$$ $$xf(x - y) - yf(x - y) - h(x - y) = 0$$ $$(x + y + z) f(x + y + z) - h(x + y + z) = 0$$ $$(r + x) f(r + x) - h(r)h(x) = 0$$ $$(-r + x) f(-r + x) - \frac{h(x)}{h(r)} = 0$$ $$(x + y + z) f(x + y + z) - h(x)h(y)h(z) = 0$$ |

*Table 20.* RSR Properties Discovered by Claude Opus 4.1 (Agentic Bitween) across RSR-Bench (continued)

| Function | Discovered RSR Properties |
|---|---|
| 6. Floudas | $$-r - f(x,y) + f(r + x, y) = 0$$ $$-s - f(x,y) + f(x, s + y) = 0$$ $$-r - s - f(x,y) + f(r + x, s + y) = 0$$ $$-s - f(r + x, y) + f(r + s + x, y) = 0$$ $$-s - f(x, r + y) + f(x, r + s + y) = 0$$ $$-2s - f(r + x, -s + y) + f(r + x, s + y) = 0$$ $$-2r - f(-r + x, s + y) + f(r + x, s + y) = 0$$ $$f(x,y) - 2f(r + x, y) + f(2r + x, y) = 0$$ $$f(x,y) - 2f(x, s + y) + f(x, 2s + y) = 0$$ $$2f(x,y) - f(-r + x, y) - f(r + x, y) = 0$$ $$-2f(x,y) + f(-r + x, y) + f(r + x, y) = 0$$ $$-2f(x,y) + f(x, -s + y) + f(x, s + y) = 0$$ $$-x\,(a + c) - y\,(b + d) + f(ax + by, cx + dy) = 0$$ $$-2f(x,y) + f(-r + x, -s + y) + f(r + x, s + y) = 0$$ $$-2f(x,y) + f(-r + x, s + y) + f(r + x, -s + y) = 0$$ $$f(x,y) - f(x, s + y) - f(r + x, y) + f(r + x, s + y) = 0$$ $$f(x,y) - f(a + x, y) - f(b + x, y) + f(a + b + x, y) = 0$$ $$f(x,y) - f(x, a + y) - f(x, b + y) + f(x, a + b + y) = 0$$ $$-f(x,y) + f(x, s + y) + f(r + x, y) - f(r + x, s + y) = 0$$ $$f(x,y) - f(r + x, t + y) - f(s + x, y) + f(r + s + x, t + y) = 0$$ $$-f(x,y) + f(a + x, b + y) + f(c + x, d + y) - f(a + c + x, b + d + y) = 0$$ $$2f(x,y) - f(x, r + y) - f(x, s + y) - f(x, t + y) + f(x, r + s + t + y) = 0$$ |

*Table 20.* RSR Properties Discovered by Claude Opus 4.1 (Agentic Bitween) across RSR-Bench (continued)

| Function | Discovered RSR Properties |
|---|---|
| 7. Mean | $$f(0, y, x + z) - f(x, y, z) = 0$$ $$f(0, x + y, z) - f(x, y, z) = 0$$ $$-f(x, y, z) + f(x + y, 0, z) = 0$$ $$-f(x, y, z) + f(x, y + z, 0) = 0$$ $$-f(x, y, z) + f(x + y, z, 0) = 0$$ $$-f(x, y, z) + f(z, 0, x + y) = 0$$ $$f(0, x + y + z, 0) - f(x, y, z) = 0$$ $$f(0, 0, x + y + z) - f(x, y, z) = 0$$ $$-f(x, y, z) + f(x + y + z, 0, 0) = 0$$ $$-f(x, y, r + z) + f(r + x, y, z) = 0$$ $$-f(x, y, r + z) + f(x, r + y, z) = 0$$ $$-f(x, r + y, z) + f(r + x, y, z) = 0$$ $$-f(x, y, z) + f(r + x, -r + y, z) = 0$$ $$-f(x, y, z) + f(x, r + y, -r + z) = 0$$ $$-f(x, y, z) + f(r + x, y, -r + z) = 0$$ $$-f(x, y, -r + z) + f(-r + x, y, z) = 0$$ $$-f(x, y, -r + z) + f(x, -r + y, z) = 0$$ $$-f(x, y, z) + f(r + x, s + y, -r - s + z) = 0$$ $$-f(r, 0, 0) - f(x, y, z) + f(r + x, y, z) = 0$$ $$-f(0, r, 0) - f(x, y, z) + f(x, r + y, z) = 0$$ $$-f(0, 0, r) - f(x, y, z) + f(x, y, r + z) = 0$$ $$-f(r, s, 0) - f(x, y, z) + f(r + x, s + y, z) = 0$$ $$-f(r, 0, s) - f(x, y, z) + f(r + x, y, s + z) = 0$$ $$-f(0, r, s) - f(x, y, z) + f(x, r + y, s + z) = 0$$ $$-2f(r, 0, 0) - f(x, y, z) + f(2r + x, y, z) = 0$$ $$-f(r, r, 0) - f(x, y, z) + f(r + x, r + y, z) = 0$$ $$-f(r, 0, r) - f(x, y, z) + f(r + x, y, r + z) = 0$$ $$-f(0, r, r) - f(x, y, z) + f(x, r + y, r + z) = 0$$ $$-f(x, y, z) + f\left(r + x, -\frac{r}{2} + y, -\frac{r}{2} + z\right) = 0$$ $$-f(r, r, r) - f(x, y, z) + f(r + x, r + y, r + z) = 0$$ $$-f(r, s, t) - f(x, y, z) + f(r + x, s + y, t + z) = 0$$ $$-2f(r, 0, 0) - f(-r + x, y, z) + f(r + x, y, z) = 0$$ $$-2f(0, r, 0) - f(x, -r + y, z) + f(x, r + y, z) = 0$$ $$-2f(0, 0, r) - f(x, y, -r + z) + f(x, y, r + z) = 0$$ $$-2f(x, y, z) + f(-r + x, -s + y, z) + f(r + x, s + y, z) = 0$$ $$-2f(x, y, z) + f(-r + x, y, -s + z) + f(r + x, y, s + z) = 0$$ $$-2f(x, y, z) + f(x, -r + y, -s + z) + f(x, r + y, s + z) = 0$$ $$-2f(x, y, z) + f(-r + x, -s + y, -t + z) + f(r + x, s + y, t + z) = 0$$ $$f(x, x, x) + f(y, y, y) + f(z, z, z) - f(x + y + z, x + y + z, x + y + z) = 0$$ $$f\left(\frac{r}{3}, \frac{r}{3}, \frac{r}{3}\right) - f(x, y, z) + f\left(-\frac{r}{3} + x, -\frac{r}{3} + y, -\frac{r}{3} + z\right) = 0$$ |

*Table 20.* RSR Properties Discovered by Claude Opus 4.1 (Agentic Bitween) across RSR-Bench (continued)

| Function | Discovered RSR Properties |
|---|---|
| 8. Tangent | $$f(x-y) - \frac{f(x) - f(y)}{f(x)f(y) + 1} = 0$$ $$f(x+y) - \frac{f(x) + f(y)}{-f(x)f(y) + 1} = 0$$ $$-f(r)f(x)f(r+x) - f(r) - f(x) + f(r+x) = 0$$ $$f(r)f(x)f(-r+x) + f(r) - f(x) + f(-r+x) = 0$$ $$-f(r)f(2x)f(r+2x) - f(r) - f(2x) + f(r+2x) = 0$$ $$-f(2r)f(x)f(2r+x) - f(2r) - f(x) + f(2r+x) = 0$$ $$f(r)f(2x)f(-r+2x) + f(r) - f(2x) + f(-r+2x) = 0$$ $$f(2r)f(x)f(-2r+x) + f(2r) - f(x) + f(-2r+x) = 0$$ $$-f(s)f(r+x)f(r+s+x) - f(s) - f(r+x) + f(r+s+x) = 0$$ $$\left(-f^2(r)f^2(x) + 1\right)f(-r+x)f(r+x) + f^2(r) - f^2(x) = 0$$ $$\left(-f^2(r)f^2(x) + 1\right)\left(-f(-r+x) + f(r+x)\right) - 2\left(f^2(x) + 1\right)f(r) = 0$$ $$f(r)f(-r+x) - f(r)f(r+x) + f(x)f(-r+x) + f(x)f(r+x)$$ $$-2f(-r+x)f(r+x) = 0$$ $$-\left(-f(x)f(y) - f(x)f(z) - f(y)f(z) + 1\right)f(x+y+z) - f(x)f(y)f(z) + f(x)$$ $$+f(y) + f(z) = 0$$ |
| 9. Cotangent | $$(f(x) + f(y))f(x+y) - f(x)f(y) + 1 = 0$$ $$(f(x) - f(y))f(x-y) + f(x)f(y) + 1 = 0$$ $$-(f(x) + f(y))f(x+y) + f(x)f(y) - 1 = 0$$ $$-2f(x)f(y) - f(x)f(x-y) + f(x)f(x+y) + f(y)f(x-y)$$ $$+f(y)f(x+y) = 0$$ $$-2f(r)f(x) + f(r)f(-r+x) + f(r)f(r+x) - f(x)f(-r+x)$$ $$+f(x)f(r+x) = 0$$ |
| 10. Difference of Squares | $$-2f(x,y) + f(-r+x, -r+y) + f(r+x, r+y) = 0$$ $$-2f(x,y) + f(-r+x, r+y) + f(r+x, -r+y) = 0$$ $$-2rx + 2sy - f(r,s) - f(x,y) + f(r+x, s+y) = 0$$ $$-2f(r,0) - 2f(x,y) + f(-r+x, y) + f(r+x, y) = 0$$ $$8rx + f(-r+x, -s+y) + f(-r+x, s+y) - f(r+x, -s+y)$$ $$-f(r+x, s+y) = 0$$ $$-8sy + f(-r+x, -s+y) - f(-r+x, s+y) + f(r+x, -s+y)$$ $$-f(r+x, s+y) = 0$$ $$-4r^2 + 4s^2 - 4f(x,y) + f(-r+x, -s+y) + f(-r+x, s+y)$$ $$+f(r+x, -s+y) + f(r+x, s+y) = 0$$ |

*Table 20.* RSR Properties Discovered by Claude Opus 4.1 (Agentic Bitween) across RSR-Bench (continued)

| Function | Discovered RSR Properties |
|---|---|
| 11. Inverse Square | $$f(xy) - \frac{1}{x^2 y^2} = 0$$ $$(x+y)^2 f(x+y) - 1 = 0$$ $$(x-y)^2 f(x-y) - 1 = 0$$ $$f\left(\frac{x}{y}\right) - \frac{y^2}{x^2} = 0$$ $$-y^2 f(x) + f\left(\frac{x}{y}\right) = 0$$ $$f(2x+y) - \frac{1}{(2x+y)^2} = 0$$ $$f(2x-y) - \frac{1}{(2x-y)^2} = 0$$ $$f(x+y+z) - \frac{1}{(x+y+z)^2} = 0$$ $$f(x-y-z) - \frac{1}{(x-y-z)^2} = 0$$ $$f(ax+by) - \frac{1}{(ax+by)^2} = 0$$ $$\left(x^2 - y^2\right)^2 f(x-y)f(x+y) - 1 = 0$$ $$(x-y)^2 (x+y)^2 f(x-y)f(x+y) - 1 = 0$$ $$-2f(x)f(y) + f(x)f(x-y) + f(x)f(x+y) + f(y)f(x-y) + f(y)f(x+y)$$ $$-8f(x-y)f(x+y) = 0$$ |
| 12. Inverse | $$cf(cx) - f(x) = 0$$ $$-rf(x) + f\left(\frac{x}{r}\right) = 0$$ $$-xf(r) + f\left(\frac{r}{x}\right) = 0$$ $$-cf(x) + f\left(\frac{x}{c}\right) = 0$$ $$f\left(\frac{x}{r}\right)f\left(\frac{r}{x}\right) - 1 = 0$$ $$-f(r)f(x) + f(rx) = 0$$ $$f(r) - f(x)f\left(\frac{r}{x}\right) = 0$$ $$-f^2(x)f\left(\frac{r}{x}\right) + f(rx) = 0$$ $$f(-r+x)f(r+x) - f\left(-r^2+x^2\right) = 0$$ $$2rf\left(-r^2+x^2\right) - f(-r+x) + f(r+x) = 0$$ $$-2xf\left(-r^2+x^2\right) + f(-r+x) + f(r+x) = 0$$ $$-2f(r)f(x) + f(r)f(-r+x) + f(r)f(r+x) - f(x)f(-r+x)$$ $$+f(x)f(r+x) = 0$$ |

*Table 20.* RSR Properties Discovered by Claude Opus 4.1 (Agentic Bitween) across RSR-Bench (continued)

| Function | Discovered RSR Properties |
|---|---|
| 13. Inverse Add | $$f(x)f(y) - f(xy + x + y) = 0$$ $$f(x)f(y)f(z) - f(xyz + xy + xz + x + yz + y + z) = 0$$ $$-f(x)f(-r + x) - f(x)f(r + x) + 2f(-r + x)f(r + x) = 0$$ $$f(w)f(x)f(y)f(z) - f(wxyz + wxy + wxz + wx + wyz + wy + wz$$ $$+ w + xyz + xy + xz + x + yz + y + z) = 0$$ |
| 14. Inverse Cot Plus One | $$f(y)f(x + y) + f(x + y)f\left(-y + \frac{\pi}{2}\right) - f(x + y) = 0$$ $$f(y)f(x - y) + f(x - y)f\left(-y + \frac{\pi}{2}\right) - f(x - y) = 0$$ $$2f(r)f(x)f(r + x) - 2f(r)f(x) + f(r) + f(x) - f(r + x) = 0$$ $$2f(r)f(x)f(-r + x) - 2f(r)f(-r + x) + f(r) - f(x) + f(-r + x) = 0$$ |
| 15. Inverse Tan Plus One | $$(\sin(r + x) + \cos(r + x))f(r + x) - \cos(r + x) = 0$$ |
| 16. X Over One Minus X | $$-f(r)f(x) + f(r)f(rx) + f(x)f(rx) + f(rx) = 0$$ $$-f(x)f(-r + x) - f(x)f(r + x) - 2f(x) + 2f(-r + x)f(r + x) + f(-r + x)$$ $$+ f(r + x) = 0$$ |
| 17. Minus X Over One Minus X | $$-r - x + f(f(r + x)) = 0$$ $$r - x + f(f(-r + x)) = 0$$ $$-(1 - r)f(r) - (1 - x)f(x) + (-r - x + 1)f(r + x) = 0$$ $$-f(r)f(x) + f(r)f(rx) + f(x)f(rx) - f(rx) = 0$$ $$-f(x)f(-r + x) - f(x)f(r + x) + 2f(x) + 2f(-r + x)f(r + x) - f(-r + x)$$ $$-f(r + x) = 0$$ |
| 18. Cosine | $$2f(x)f(y) - f(x - y) - f(x + y) = 0$$ $$-2f(x)f(y) + f(x - y) + f(x + y) = 0$$ $$f(x) - 2f(y)f(x - y) + f(x - 2y) = 0$$ $$-2f(r)f(x) + f(-r + x) + f(r + x) = 0$$ $$-2f(r)f(r + x) + f(x) + f(2r + x) = 0$$ $$-2f(x)f(nr) + f(-nr + x) + f(nr + x) = 0$$ $$f^2(r) - 2f(r)f(x)f(r + x) + f^2(x) + f^2(r + x) - 1 = 0$$ $$-2f^2(x) + 4f(x)f(y)f(x + y) - 2f^2(y) - 2f^2(x + y) + 2 = 0$$ |

*Table 20.* RSR Properties Discovered by Claude Opus 4.1 (Agentic Bitween) across RSR-Bench (continued)

| Function | Discovered RSR Properties |
|---|---|
| 19. Hyperbolic Cosine | $$2f(x)f(y) - f(x-y) - f(x+y) = 0$$ $$-2f(x)f(y) + f(x-y) + f(x+y) = 0$$ $$-2f(r)f(x) + f(-r+x) + f(r+x) = 0$$ $$-2f(r)f(x) + f(-r+x) + f(r+x) = 0$$ $$f^2(x) + f^2(y) - f(x-y)f(x+y) - 1 = 0$$ $$-f^2(x) - f^2(y) + f(x-y)f(x+y) + 1 = 0$$ $$f(r)f(-r+x) + f(r)f(r+x) - f(2r)f(x) - f(x) = 0$$ $$-f(r)f(2x) - f(r) + f(x)f(-r+x) + f(x)f(r+x) = 0$$ $$f(x)f(z) + f(y)f(x+y+z) - f(x+y)f(y+z) - f(x+z) = 0$$ $$f(x)f(y) + f(z)f(x+y+z) - f(x+y) - f(x+z)f(y+z) = 0$$ $$f(x)f(x+y+z) + f(y)f(z) - f(x+y)f(x+z) - f(y+z) = 0$$ |
| 20. Squared | $$-n^2 f(x) + f(nx) = 0$$ $$2f(x) + 2f(y) - f(x-y) - f(x+y) = 0$$ $$-2f(x) - 2f(y) + f(x-y) + f(x+y) = 0$$ $$-2f(a) - 2f(b) + f(a-b) + f(a+b) = 0$$ $$-2f(r) - 2f(x) + f(-r+x) + f(r+x) = 0$$ $$f(2x) + 4f(y) - 2f(x-y) - 2f(x+y) = 0$$ $$2xy + 2xz + 2yz + f(x) + f(y) + f(z) - f(x+y+z) = 0$$ $$-4f(r) - f(x) + f(-r+x) - f(r+x) + f(2r+x) = 0$$ $$f(x) + f(y) + f(z) - f(x+y) - f(x+z) - f(y+z) + f(x+y+z) = 0$$ $$f(x) + f(y) + f(z) - f(x+y) - f(x+z) - f(y+z) + f(x+y+z) = 0$$ |

*Table 20.* RSR Properties Discovered by Claude Opus 4.1 (Agentic Bitween) across RSR-Bench (continued)

| Function | Discovered RSR Properties |
|---|---|
| 21. Sine | $$f^2(x) - f^2(y) - f(x-y)f(x+y) = 0$$ $$f^2(r) - f^2(x) + f(-r+x)f(r+x) = 0$$ $$f^2(r) + f(x)f(2r+x) - f^2(r+x) = 0$$ $$f^2(a) - f^2(x) + f(-a+x)f(a+x) = 0$$ $$-f^2(x) + f^2(y) + f(x-y)f(x+y) = 0$$ $$f(2x)f(2y) + f^2(x-y) - f^2(x+y) = 0$$ $$f(x)f(2y) - f(y)f(x-y) - f(y)f(x+y) = 0$$ $$f(x)f(x-y) - f(x)f(x+y) + f(2x)f(y) = 0$$ $$f(r)f(-r+x) + f(r)f(r+x) - f(2r)f(x) = 0$$ $$-f(r)f(x) - f(r)f(2r+x) + f(2r)f(r+x) = 0$$ $$f^2(x)f(y) - f^3(y) - f(y)f(x-y)f(x+y) = 0$$ $$-f(x)f(x-y) + f(y)f(2y) + f(x-2y)f(x+y) = 0$$ $$f(r)f(2r) - f(x)f(r+x) + f(-r+x)f(2r+x) = 0$$ $$-f^2(x)f(x+y) + f^2(y)f(x+y) + f(x-y)f^2(x+y) = 0$$ $$-f^2(x)f(x-y) + f^2(y)f(x-y) + f^2(x-y)f(x+y) = 0$$ $$f^2(x)f(2y) - 2f(x)f(y)f(x+y) + f(2x)f^2(y) = 0$$ $$-f^2(x)f(z) + f^2(y)f(z) + f(z)f(x-y)f(x+y) = 0$$ $$f^2(x)f(x-y) - f^2(x)f(x+y) + f(x)f(2x)f(y) = 0$$ $$f^2(x)f(2x) - f(2x)f^2(y) - f(2x)f(x-y)f(x+y) = 0$$ $$f^2(x)f(x+2y) - f^2(y)f(x+2y) - f(x-y)f(x+y)f(x+2y) = 0$$ $$f^2(x)f(x-2y) - f^2(y)f(x-2y) - f(x-2y)f(x-y)f(x+y) = 0$$ $$f(x)f^2(2y) - f(y)f(2y)f(x-y) - f(y)f(2y)f(x+y) = 0$$ $$f(x)f(2x)f(x-y) - f(x)f(2x)f(x+y) + f^2(2x)f(y) = 0$$ |

*Table 20.* RSR Properties Discovered by Claude Opus 4.1 (Agentic Bitween) across RSR-Bench (continued)

| Function | Discovered RSR Properties |
|---|---|
| 22. Hyperbolic Sine | $$f^2(x) - f^2(y) - f(x-y)f(x+y) = 0$$ $$-f^2(x) + f^2(y) + f(x-y)f(x+y) = 0$$ $$f^2(r) - f^2(x) + f(-r+x)f(r+x) = 0$$ $$f^2(r) - f^2(x) + f(-r+x)f(r+x) = 0$$ $$2\sqrt{f^2(y)+1}f(x) - f(x-y) - f(x+y) = 0$$ $$2\sqrt{f^2(x)+1}f(y) + f(x-y) - f(x+y) = 0$$ $$-2\sqrt{f^2(y)+1}f(x) + f(x-y) + f(x+y) = 0$$ $$-2\sqrt{f^2(x)+1}f(y) - f(x-y) + f(x+y) = 0$$ $$-2\sqrt{f^2(x)+1}f(r) + f(r-x) + f(r+x) = 0$$ $$-2\sqrt{f^2(x)+1}f(y) + f(-x+y) + f(x+y) = 0$$ $$-2\sqrt{f^2(y)+1}f(x) - f(-x+y) + f(x+y) = 0$$ $$-2\sqrt{f^2(x)+1}f(r) - f(-r+x) + f(r+x) = 0$$ $$-2\sqrt{f^2(x)+1}f(3r) - f(-3r+x) + f(3r+x) = 0$$ $$-2\sqrt{f^2(x)+1}f(2r) - f(-2r+x) + f(2r+x) = 0$$ $$-2\left(2f^2(r)+1\right)f(x) + f(-2r+x) + f(2r+x) = 0$$ $$f(x)f(2y) - f(y)f(x-y) - f(y)f(x+y) = 0$$ $$f(r)f(-r+x) + f(r)f(r+x) - f(2r)f(x) = 0$$ $$-4\left(f^2(y)+1\right)f^2(x) + (f(x-y)+f(x+y))^2 = 0$$ $$-\left(4f^2(x)+4\right)f^2(y) + (-f(x-y)+f(x+y))^2 = 0$$ $$f^2(r)f(x) - f^3(x) + f(x)f(-r+x)f(r+x) = 0$$ $$-4\left(f^2(x)+1\right)f^2(y) + (-f(x-y)+f(x+y))^2 = 0$$ $$-f^2(r)f(x) + f^3(x) - f(x)f(-r+x)f(r+x) = 0$$ $$f(x)f(x-2y) - f(x)f(x+2y) + 2f(2x)f(2y) + f^2(x-y) - f^2(x+y) = 0$$ |

*Table 20.* RSR Properties Discovered by Claude Opus 4.1 (Agentic Bitween) across RSR-Bench (continued)

| Function | Discovered RSR Properties |
|---|---|
| 23. Cube | $$-n^3 f(x) + f(nx) = 0$$ $$-8f(x + y) + f(2x + 2y) = 0$$ $$-27f(x + y) + f(3x + 3y) = 0$$ $$-y^3 f(x) + f(x)f(y) = 0$$ $$-6xy^2 - 2f(x) + f(x - y) + f(x + y) = 0$$ $$-6x^2 y - 2f(y) - f(x - y) + f(x + y) = 0$$ $$-6ab^2 - 2f(a) + f(a - b) + f(a + b) = 0$$ $$-6a^2 b - 2f(b) - f(a - b) + f(a + b) = 0$$ $$-6r^2 x - 2f(x) + f(-r + x) + f(r + x) = 0$$ $$-6rx^2 - 2f(r) - f(-r + x) + f(r + x) = 0$$ $$-24a^2 b - 2f(b) - f(2a - b) + f(2a + b) = 0$$ $$3xy(x - y) - f(x) + f(y) + f(x - y) = 0$$ $$-3xy(x + y) - f(x) - f(y) + f(x + y) = 0$$ $$-6abc + f(a) + f(b) + f(c) - f(a + b) - f(a + c) - f(b + c)$$ $$+f(a + b + c) = 0$$ |

*Table 20.* RSR Properties Discovered by Claude Opus 4.1 (Agentic Bitween) across RSR-Bench (continued)

| Function | Discovered RSR Properties |
|---|---|
| 24. Logarithm | $$-2f(x) + f(x^2) = 0$$ $$-3f(x) + f(x^3) = 0$$ $$-4f(x) + f(x^4) = 0$$ $$-5f(x) + f(x^5) = 0$$ $$-nf(x) + f(x^n) = 0$$ $$-rf(x) + f(x^r) = 0$$ $$-2f(r) + f(r^2) = 0$$ $$-yf(x) + f(x^y) = 0$$ $$f\left(\frac{1}{x}\right) + f(x) = 0$$ $$3f(x^2) - 2f(x^3) = 0$$ $$-abf(x) + f(x^{ab}) = 0$$ $$6f(\sqrt{x}) - f(x^3) = 0$$ $$9f(\sqrt[3]{x}) - f(x^3) = 0$$ $$f(\sqrt{x}) - \frac{f(x)}{2} = 0$$ $$f(\sqrt[3]{x}) - \frac{f(x)}{3} = 0$$ $$f\left(x^{\frac{1}{n}}\right) - \frac{f(x)}{n} = 0$$ $$-f(r) - f(x) + f(rx) = 0$$ $$-f(x) - f(y) + f(xy) = 0$$ $$-\frac{mf(x)}{n} + f\left(x^{\frac{m}{n}}\right) = 0$$ $$f(r) - f(x) + f\left(\frac{x}{r}\right) = 0$$ $$-2f(r) - f(x) + f(r^2 x) = 0$$ |

*Table 20.* RSR Properties Discovered by Claude Opus 4.1 (Agentic Bitween) across RSR-Bench (continued)

| Function | Discovered RSR Properties |
|---|---|
| 25. Logarithm (cont.) | $$f(x) - f(y) - f\left(\frac{x}{y}\right) = 0$$ $$-2f(y) - f\left(\frac{x}{y}\right) + f(xy) = 0$$ $$-af(x) - bf(y) + f\left(x^a y^b\right) = 0$$ $$-2f(x) - 2f(y) + f\left(x^2 y^2\right) = 0$$ $$-2f(x) - 3f(y) + f\left(x^2 y^3\right) = 0$$ $$-af(x) + bf(y) + f\left(x^a y^{-b}\right) = 0$$ $$f(r) - f\left(\frac{x}{y}\right) + f\left(\frac{x}{ry}\right) = 0$$ $$-2f(x) + 3f(y) + f\left(\frac{x^2}{y^3}\right) = 0$$ $$-\frac{af(x)}{b} + f\left((x^a)^{\frac{1}{b}}\right) = 0$$ $$-nf(x) - nf(y) + f((xy)^n) = 0$$ $$-f(r) - f(x) - f(y) + f(rxy) = 0$$ $$-f(x) - f(y) - f(z) + f(xyz) = 0$$ $$-\frac{f(x)}{2} - \frac{f(y)}{2} + f(\sqrt{xy}) = 0$$ $$f(r) - f(x) - f(y) + f\left(\frac{xy}{r}\right) = 0$$ $$-f(x) - f(y) + f(z) + f\left(\frac{xy}{z}\right) = 0$$ $$-f(x) + f(y) + f(z) + f\left(\frac{x}{yz}\right) = 0$$ $$-f(y) + f(ry) - f\left(\frac{x}{y}\right) + f\left(\frac{x}{ry}\right) = 0$$ $$-f(y) + f(rx) - 2f\left(\frac{x}{y}\right) + f\left(\frac{x}{ry}\right) = 0$$ $$-2f(y) - 2f\left(\frac{x}{y}\right) + f\left(\frac{x}{ry}\right) + f(rxy) = 0$$ |
| 26. Secant | $$f(r + x)\cos(r + x) - 1 = 0$$ $$f(-r + x)\cos(r - x) - 1 = 0$$ $$\left(-f^2(x)\sin^2(r) + 1\right)f(-r + x)f(r + x) - f^2(x) = 0$$ |

*Table 20.* RSR Properties Discovered by Claude Opus 4.1 (Agentic Bitween) across RSR-Bench (continued)

| Function | Discovered RSR Properties |
|---|---|
| 26. Cosecant | $$\left(\cos\left(x-y\right)-\cos\left(x+y\right)\right)f(x)f(y)-2=0$$ $$\left(\sin^2\left(x\right)-\sin^2\left(y\right)\right)f(x-y)f(x+y)-1=0$$ $$\left(\sin\left(x\right)\cos\left(y\right)+\sin\left(y\right)\cos\left(x\right)\right)f(x+y)-1=0$$ $$\left(\sin\left(x\right)\cos\left(y\right)-\sin\left(y\right)\cos\left(x\right)\right)f(x-y)-1=0$$ $$f(x-y)f(x+y)-\frac{1}{\sin^2\left(x\right)\cos^2\left(y\right)-\sin^2\left(y\right)\cos^2\left(x\right)}=0$$ $$f(x+y)\sin\left(x\right)\cos\left(y\right)+f(x+y)\sin\left(y\right)\cos\left(x\right)-f(x+y)\sin\left(x+y\right)=0$$ |
| 28. Sinc Composite | $$f(x+y)\,\mathrm{rsr}_x\left(x,y\right)-\mathrm{rsr}_{sin}\left(x,y\right)=0$$ $$2xf(x)f(2y)-xf(y)f(x-y)-xf(y)f(x+y)+yf(y)f(x-y)$$ $$-yf(y)f(x+y)=0$$ $$xf(x)f(x-y)-xf(x)f(x+y)-yf(x)f(x-y)-yf(x)f(x+y)$$ $$+2yf(2x)f(y)=0$$ |

*Table 20.* RSR Properties Discovered by Claude Opus 4.1 (Agentic Bitween) across RSR-Bench (continued)

| Function | Discovered RSR Properties |
|---|---|
| 29. Modulo | $$-f(x) + f(Ry + x) = 0$$ $$-f(x) + f(Rk + x) = 0$$ $$-f(x) + f(-Rn + x) = 0$$ $$-f(2x) + f(R + 2x) = 0$$ $$-f(R - x) + f(2R - x) = 0$$ $$f(nx) - f(nf(x)) = 0$$ $$-f(af(x)) + f(Rb + ax) = 0$$ $$f(xy) - f(f(x)f(y)) = 0$$ $$f(rx) - f(f(r)f(x)) = 0$$ $$f(x + y) - f(f(x) + f(y)) = 0$$ $$-f(af(x)) + f(x(Rb + a)) = 0$$ $$f(r + x) - f(f(r) + f(x)) = 0$$ $$-f(f(x)f(y)) + f(R + xy) = 0$$ $$-f(f(x)f(y)) + f(Rz + xy) = 0$$ $$-f(f(x)f(y)) + f(-R + xy) = 0$$ $$-f(f(x) + f(y)) + f(R + x + y) = 0$$ $$-f(f(x) + f(y)) + f(-R + x + y) = 0$$ $$f(ax + by) - f(af(x) + bf(y)) = 0$$ $$-f(f(x) + f(y)) + f(-Rz + x + y) = 0$$ $$f(x(R + y)) - f(f(x)f(y)) = 0$$ $$f(xyz) - f(f(x)f(y)f(z)) = 0$$ $$f(y(r + x)) - f(f(y)f(r + x)) = 0$$ $$f(x + y + z) - f(f(x) + f(y) + f(z)) = 0$$ $$f(r + s + x) - f(f(r) + f(s) + f(x)) = 0$$ $$f(r + x + y) - f(f(r) + f(x) + f(y)) = 0$$ $$f((R + x)(R + y)) - f(f(x)f(y)) = 0$$ $$-f(f(x) + f(y) + f(z)) + f(-Rw + x + y + z) = 0$$ $$f(ax + by + cz) - f(af(x) + bf(y) + cf(z)) = 0$$ $$f(wxyz) - f(f(w)f(x)f(y)f(z)) = 0$$ $$f((r + x)(s + y)) - f(f(r + x)f(s + y)) = 0$$ $$f(r + s + t + x) - f(f(r) + f(s) + f(t) + f(x)) = 0$$ |

*Table 20.* RSR Properties Discovered by Claude Opus 4.1 (Agentic Bitween) across RSR-Bench (continued)

| Function | Discovered RSR Properties |
|---|---|
| 30. Modulo Multiplication | $$f(x, y) - f(xy, 1) = 0$$ $$f(x, yz) - f(y, xz) = 0$$ $$f(xy, z) - f(yz, x) = 0$$ $$-f(1, xy) + f(x, y) = 0$$ $$-f(x, ry) + f(rx, y) = 0$$ $$f(rx, sy) - f(sx, ry) = 0$$ $$f(xy, wz) - f(xz, wy) = 0$$ $$-f(x, 1) + f\left(\frac{x}{y}, y\right) = 0$$ $$-f(1, y) + f\left(x, \frac{y}{x}\right) = 0$$ $$-f(1, y) + f\left(\frac{1}{x}, xy\right) = 0$$ $$f\left(x, \frac{1}{y}\right) - f\left(\frac{x}{y}, 1\right) = 0$$ $$f(x, yz) - f(f(x, y), z) = 0$$ $$-f\left(1, \frac{y}{x}\right) + f\left(\frac{1}{x}, y\right) = 0$$ $$-f(x, f(y, z)) + f(xy, z) = 0$$ $$f\left(x, y^2\right) - f(f(x, y), y) = 0$$ $$-f(x, f(x, y)) + f\left(x^2, y\right) = 0$$ $$-f(ab, f(x, y)) + f(ax, by) = 0$$ $$-f(rs, f(x, y)) + f(rx, sy) = 0$$ $$-f\left(x^2, f(y, z)\right) + f(xy, xz) = 0$$ $$-f(x, f(xy, z)) + f\left(x^2 y, z\right) = 0$$ $$f\left(x, y^2 z^2\right) - f\left(f\left(x, y^2\right), z^2\right) = 0$$ $$f(x, f(y, z)) - f(y, f(x, z)) = 0$$ $$f(x, f(y, z)) - f(z, f(x, y)) = 0$$ $$-f\left(x^2, f\left(y^2, z\right)\right) + f\left(x^2 y^2, z\right) = 0$$ $$-f(x, f(y, z)) + f(f(x, y), z) = 0$$ $$f(x, wyz) - f(f(f(x, y), z), w) = 0$$ $$f\left(x, y^3\right) - f(f(f(x, y), y), y) = 0$$ $$f(x, aby) - f(f(f(x, a), b), y) = 0$$ $$-f(x, f(y, f(z, w))) + f(xyz, w) = 0$$ $$-f(x, f(x, f(x, y))) + f\left(x^3, y\right) = 0$$ $$-f(a, f(b, f(x, y))) + f(abx, y) = 0$$ $$-f(z, f(x, f(y, w))) + f(xyz, w) = 0$$ $$f(f(x, y), f(z, w)) - f(f(x, z), f(y, w)) = 0$$ |

*Table 20.* RSR Properties Discovered by Claude Opus 4.1 (Agentic Bitween) across RSR-Bench (continued)

| Function | Discovered RSR Properties |
|---|---|
| 31. Integer Multiplication | $$f(x, y) - f(y, x) = 0$$ $$-cf(x, y) + f(cx, y) = 0$$ $$-cf(x, y) + f(x, cy) = 0$$ $$-f(x, 3y) + f(3x, y) = 0$$ $$-nf(x, y) + f(nx, y) = 0$$ $$-nf(x, y) + f(x, ny) = 0$$ $$-abf(x, y) + f(ax, by) = 0$$ $$-4f^2(x, y) + f^2(x, 2y) = 0$$ $$-4f^2(x, y) + f^2(2x, y) = 0$$ $$-f(x - y, x + y) + f(x + y, x - y) = 0$$ $$-f(r, y) - f(x, y) + f(r + x, y) = 0$$ $$-f(x, s) - f(x, y) + f(x, s + y) = 0$$ $$-f(y, x) - f(y, y) + f(x + y, y) = 0$$ $$f(x, x) - f(y, y) - f(x - y, x + y) = 0$$ $$-f(x, x) + f(y, y) + f(x + y, x - y) = 0$$ $$f(x, y)f(r + x, s + y) - f(x, s + y)f(r + x, y) = 0$$ $$f(x, x + y) - f(y, x) - f(y, y) - f(x - y, x + y) = 0$$ $$f(x, y)f(-r + x, -s + y) - f(x, -s + y)f(-r + x, y) = 0$$ $$-f(r, s) - f(r, y) - f(x, s) - f(x, y) + f(r + x, s + y) = 0$$ $$f(r, s) - f(r, y) + f(x, s) - f(x, y) + f(r + x, -s + y) = 0$$ $$f(r, s) + f(r, y) - f(x, s) - f(x, y) + f(-r + x, s + y) = 0$$ $$-f(r, r) - f(r, y) - f(x, r) - f(x, y) + f(r + x, r + y) = 0$$ $$-f(r, s) + f(r, y) + f(x, s) - f(x, y) + f(-r + x, -s + y) = 0$$ $$-f(r, s) - 2f(r, y) - 2f(x, s) - 4f(x, y) + f(r + 2x, s + 2y) = 0$$ $$-f(x, -s + y)f(-r + x, y)f(r + x, s + y)$$ $$+f(x, s + y)f(-r + x, -s + y)f(r + x, y) = 0$$ |

*Table 20.* RSR Properties Discovered by Claude Opus 4.1 (Agentic Bitween) across RSR-Bench (continued)

| Function | Discovered RSR Properties |
|---|---|
| 32. Hyperbolic Tangent | $$f(r + x) - \frac{f(r) + f(x)}{f(r)f(x) + 1} = 0$$ $$f(-r + x) - \frac{-f(r) + f(x)}{-f(r)f(x) + 1} = 0$$ $$f(2r + 2x) - \frac{f(2r) + f(2x)}{f(2r)f(2x) + 1} = 0$$ $$f(x)f(y)f(x - y) + f(x) - f(y) - f(x - y) = 0$$ $$f(-r + x)f(r + x) - \frac{-f^2(r) + f^2(x)}{-f^2(r)f^2(x) + 1} = 0$$ $$-\frac{2\left(1 - f^2(x)\right)f(r)}{-f^2(r)f^2(x) + 1} - f(-r + x) + f(r + x) = 0$$ $$f(x)f(x - y) + f(x)f(x + y) + f(y)f(x - y) - f(y)f(x + y)$$ $$-2f(x - y)f(x + y) = 0$$ $$f(r)f(-r + x) - f(r)f(r + x) + f(x)f(-r + x) + f(x)f(r + x)$$ $$-2f(-r + x)f(r + x) = 0$$ |
| 33. Sigmoid | $$-\frac{kf(x)}{(k - 1)f(x) + 1} + f(x + \log(k)) = 0$$ $$f(nx) - \frac{f^n(x)}{(1 - f(x))^n + f^n(x)} = 0$$ $$((1 - f(x))(1 - f(y)) + f(x)f(y))f(x + y) - f(x)f(y) = 0$$ $$((1 - f(x))(1 - f(-y)) + f(x)f(-y))f(x - y) - f(x)f(-y) = 0$$ $$f(x + y + z) - \frac{f(z)f(x + y)}{(1 - f(z))(1 - f(x + y)) + f(z)f(x + y)} = 0$$ $$-(1 - f(y))f(x) + ((1 - f(x))f(y) + (1 - f(y))f(x))f(x - y) = 0$$ |

*Table 20.* RSR Properties Discovered by Claude Opus 4.1 (Agentic Bitween) across RSR-Bench (continued)

| Function | Discovered RSR Properties |
|---|---|
| 34. Softmax21 | $$f(0, y) - f(x, x + y) = 0$$ $$f(x, y) - f(x - y, 0) = 0$$ $$f(x, x - y) - f(y, 0) = 0$$ $$f(x, y) - \frac{1}{e^{-x+y} + 1} = 0$$ $$-f(x, 0) + f(x + y, y) = 0$$ $$-f(0, y) + f(x, x + y) = 0$$ $$-f(0, y) + f(x - y, x) = 0$$ $$f(x, 0) + f(y, x + y) - 1 = 0$$ $$f(x, y) - f(r + x - y, r) = 0$$ $$-f(x, y) + f(r + x, r + y) = 0$$ $$f(x, r + y) - f(-r + x, y) = 0$$ $$-f(x, r + y) + f(-r + x, y) = 0$$ $$-f(x, -r + y) + f(r + x, y) = 0$$ $$f(x, x + y) + f(x + y, x) - 1 = 0$$ $$f(y, x + y) + f(x + y, y) - 1 = 0$$ $$f(y, r + x) + f(r + x, y) - 1 = 0$$ $$f(x, r + y) + f(r + y, x) - 1 = 0$$ $$-f(x, y) + f(-r + x, -r + y) = 0$$ $$f(2x, 2y) - f(x - y, -x + y) = 0$$ $$-f(s + x, y) + f(r + s + x, r + y) = 0$$ $$f(r + x, s + y) + f(s + y, r + x) - 1 = 0$$ $$-f^2(x, y) + f(x, y)f(r + x, r + y) = 0$$ $$-\left(1 - f(x, y)\right)f(x, y) + f(x, y)f(y, x) = 0$$ $$f(x, y)f(y, z)f(z, x) - f(x, z)f(y, x)f(z, y) = 0$$ |

*Table 20.* RSR Properties Discovered by Claude Opus 4.1 (Agentic Bitween) across RSR-Bench (continued)

| Function | Discovered RSR Properties |
|---|---|
| 35. Softmax2 2 | $$f(y, x) + f(x - y, 0) - 1 = 0$$ $$f(x, y) - f(x - z, y - z) = 0$$ $$-f(x, y) + f(r + x, r + y) = 0$$ $$f(0, -x + y) + f(y, x) - 1 = 0$$ $$-f(x, y) + f(a + x, a + y) = 0$$ $$f(y, x) + f(r + x, r + y) - 1 = 0$$ $$f(x, r + x) + f(r + x, x) - 1 = 0$$ $$f(y, r + x) + f(r + x, y) - 1 = 0$$ $$f(x, r + y) + f(r + y, x) - 1 = 0$$ $$f(y, x) + f(-r + x, -r + y) - 1 = 0$$ $$f(r + x, s + y) - f(r - s + x, y) = 0$$ $$f(a + x, b + y) - f(a - b + x, y) = 0$$ $$-\frac{a}{a + e^x} + f(x, \log(a)) = 0$$ $$f(-r + x, -s + y) - f(-r + s + x, y) = 0$$ $$-\frac{b}{a + b} + f(\log(a), \log(b)) = 0$$ $$(ae^x + e^y) f(x + \log(a), y) - e^y = 0$$ $$-be^y + (be^y + e^x) f(x, y + \log(b)) = 0$$ $$-f^2(y, x) + 2f(y, x) + f^2(x - y, 0) - 1 = 0$$ $$-\frac{e^x + e^y}{e^y + e^{c+x}} + \frac{f(c + x, y)}{f(x, y)} = 0$$ $$-\frac{be^y}{ae^x + be^y} + f(x + \log(a), y + \log(b)) = 0$$ $$f(x, y)f(y, z)f(z, x) - f(x, z)f(y, x)f(z, y) = 0$$ |
| 36. Logistic | $$-L + f(x) + f(-x + 2x_0) = 0$$ $$-L + f(nr + x) + f(-nr - x + 2x_0) = 0$$ $$f(-r)f^2(r + x) + f(r)f^2(r + x) - f^2(r + x) = 0$$ $$f(-r)f^2(-r + x) + f(r)f^2(-r + x) - f^2(-r + x) = 0$$ $$f(r)f(-r + x) - f(r)f(r + x) + f(x)f(-r + x) + f(x)f(r + x) - f(x)$$ $$-2f(-r + x)f(r + x) + f(r + x) = 0$$ $$f(r)f(-r + x_0) - f(r)f(r + x_0) + f(x_0)f(-r + x_0) + f(x_0)f(r + x_0) - f(x_0)$$ $$-2f(-r + x_0)f(r + x_0) + f(r + x_0) = 0$$ |

*Table 20.* RSR Properties Discovered by Claude Opus 4.1 (Agentic Bitween) across RSR-Bench (continued)

| Function | Discovered RSR Properties |
|---|---|
| 37. Logistic Scaled | $$f(-nr) + f(nr) - 3 = 0$$ $$-L + f(-ar) + f(ar) = 0$$ $$-L + f(x) + f(-x + 2x_0) = 0$$ $$-L + f(x) + f(-x + 2x_0) = 0$$ $$-L + f(-r + x_0) + f(r + x_0) = 0$$ $$-L + f(r + x) + f(-r - x + 2x_0) = 0$$ $$-L + f(-r + x) + f(r - x + 2x_0) = 0$$ $$-L + f(-nr + x_0) + f(nr + x_0) = 0$$ $$-L + f(nr + x) + f(-nr - x + 2x_0) = 0$$ $$-L + f(ar + x) + f(-ar - x + 2x_0) = 0$$ $$Lf(x) - f^2(x) - f(x)f(-x + 2x_0) = 0$$ $$-Lf(x) + f^2(x) + f(x)f(-x + 2x_0) = 0$$ $$(L - f(x))\,f(x) - f(x)f(-x + 2x_0) = 0$$ $$f(r)f(-r + x) - f(r)f(r + x) + f(x)f(-r + x) + f(x)f(r + x) - 3f(x)$$ $$-2f(-r + x)f(r + x) + 3f(r + x) = 0$$ $$f(2r)f(-2r + x) - f(2r)f(2r + x) + f(x)f(-2r + x) + f(x)f(2r + x) - 3f(x)$$ $$-2f(-2r + x)f(2r + x) + 3f(2r + x) = 0$$ |
| 38. Square Loss | $$-\left(1 - \frac{x}{c}\right)^2 + f\left(\frac{x}{c}\right) = 0$$ $$-h - 2x + 2 + \frac{-f(x) + f(h + x)}{h} = 0$$ $$-2h^2 - 2f(x) + f(-h + x) + f(h + x) = 0$$ $$-(1 - c)^2 f(x) + f(c\,(1 - x) + x) = 0$$ $$-2f(x) - 2f(1 - y) + f(x - y) + f(x + y) = 0$$ $$-2f(x) - 2f(1 - r) + f(-r + x) + f(r + x) = 0$$ $$-2f(ax) + f(ax - by) + f(ax + by) - 2f(-by + 1) = 0$$ $$-2x - 2y - (1 - y)\,(2 - 2x) - f(x) - f(y) + f(x + y) + 3 = 0$$ $$\left(\frac{x}{2} - \frac{y}{2}\right)^2 - \frac{f(x)}{2} - \frac{f(y)}{2} + f\left(\frac{x}{2} + \frac{y}{2}\right) = 0$$ |

*Table 20.* RSR Properties Discovered by Claude Opus 4.1 (Agentic Bitween) across RSR-Bench (continued)

| Function | Discovered RSR Properties |
|---|---|
| 40. Savage Loss Basis | $$(g(x-y)+1)^2 f(x-y) - 1 = 0$$ $$(g(r+x)+1)^2 f(r+x) - 1 = 0$$ $$(g(-r+x)+1)^2 f(-r+x) - 1 = 0$$ $$(g(x)g(y)+1)^2 f(x+y) - 1 = 0$$ $$f(x+y) - \frac{1}{(g(x)g(y)+1)^2} = 0$$ $$(g(x)+g(y))^2 f(x-y) - g^2(y) = 0$$ $$(g(x)+g(r+x))^2 f(r) - g^2(x) = 0$$ $$f(x-y) - \frac{g^2(y)}{(g(x)+g(y))^2} = 0$$ |
| 43. Arctangent | $$-\frac{x-y}{xy+1} + \tan(f(x)-f(y)) = 0$$ $$-\frac{x+y}{-xy+1} + \tan(f(x)+f(y)) = 0$$ $$\cos(f(x)+f(y)) - \frac{-xy+1}{\sqrt{(x^2+1)(y^2+1)}} = 0$$ |
| 47. ReLU | $$f(\max(x,y)) - \max(f(x),f(y)) = 0$$ $$f(\min(x,y)) - \min(f(x),f(y)) = 0$$ $$-f(x)f(y) + f(f(x)f(y)) = 0$$ $$-f(\max(x,y)) + \max(f(x),f(y)) = 0$$ $$-f(\min(x,y)) + \min(f(x),f(y)) = 0$$ |
| 54. Square Root | $$r - x + f^2(-r+x) = 0$$ $$-r - x + f^2(r+x) = 0$$ $$-2r - f^2(-r+x) + f^2(r+x) = 0$$ $$-2x + f^2(-r+x) + f^2(r+x) = 0$$ $$r^2 - x^2 + f^2(-r+x)f^2(r+x) = 0$$ $$f^2(r) + f^2(x) - f^2(r+x) = 0$$ $$-2f^2(x) + f^2(-r+x) + f^2(r+x) = 0$$ $$-2f^3(x) + f(x)f^2(-r+x) + f(x)f^2(r+x) = 0$$ $$-2f(r)f^2(x) + f(r)f^2(-r+x) + f(r)f^2(r+x) = 0$$ |

*Table 20.* RSR Properties Discovered by Claude Opus 4.1 (Agentic Bitween) across RSR-Bench (continued)

| Function | Discovered RSR Properties |
|---|---|
| 55. Cube Root | $$-xy + f^3(xy) = 0$$ $$-x - y + f^3(x + y) = 0$$ $$-x + y + f^3(x - y) = 0$$ $$-2f^3(x) + f^3(-r + x) + f^3(r + x) = 0$$ $$-6f^3(y) + 2f^3(2y) - f^3(x - y) + f^3(x + y) = 0$$ $$-6f^3(x) + 2f^3(2x) + f^3(x - y) + f^3(x + y) = 0$$ $$-2f^3(y) + 3f^3(2y) + 2f^3(x - y) - 2f^3(x + y) = 0$$ $$f^3(x) + 2f^3(2x) + f^3(y) + 2f^3(2y) - 5f^3(x + y) = 0$$ $$f^3(x) + 2f^3(2x) - f^3(y) - 2f^3(2y) - 5f^3(x - y) = 0$$ $$-f^3(r) - 2f^3(2r) + 5f^3(3r) + 2f^3(-2r + x) + f^3(-r + x) - f^3(r + x)$$ $$-2f^3(2r + x) = 0$$ $$-f^3(r) + 10f^3(2r) - 3f^3(3r) + 2f^3(-2r + x) + f^3(-r + x) - f^3(r + x)$$ $$-2f^3(2r + x) = 0$$ $$60f^3(r) - 5f^3(2r) - 8f^3(3r) + 5f^3(-2r + x) + 3f^3(-r + x) - 3f^3(r + x)$$ $$-5f^3(2r + x) = 0$$ |
| 57. Floor | $$-f(x)f(y) + f(xy) - f(xy - \lfloor x \rfloor \lfloor y \rfloor) = 0$$ $$-f(x) - f(y) + f(x + y) - f(x + y - \lfloor x \rfloor - \lfloor y \rfloor) = 0$$ $$-f(r) - f(x) + f(r + x) - f(r + x - \lfloor r \rfloor - \lfloor x \rfloor) = 0$$ $$f(r) - f(x) + f(-r + x) - f(-r + x + \lfloor r \rfloor - \lfloor x \rfloor) = 0$$ $$-2f(r) - f(x) + f(2r + x) - f(2r + x - 2 \lfloor r \rfloor - \lfloor x \rfloor) = 0$$ $$-f(s) - f(r + x) + f(r + s + x) - f(r + s + x - \lfloor s \rfloor - \lfloor r + x \rfloor) = 0$$ $$-f(x) - f(y) - f(z) + f(x + y + z) - f(x + y + z - \lfloor x \rfloor - \lfloor y \rfloor - \lfloor z \rfloor) = 0$$ |
| 58. Ceiling | $$f(x + y - f(x + y)) = 0$$ $$-f(x + y) + f(f(x + y)) = 0$$ $$-f(x) - f(y) + f(x + f(y)) = 0$$ $$-f(x) + f(y) + f(x - f(y)) = 0$$ $$-f(x) - f(y) + f(y + f(x)) = 0$$ $$-f(x)f(y) + f(f(x)f(y)) = 0$$ $$-f(x) - f(y) + f(f(x) + f(y)) = 0$$ $$-f(x) + f(y) + f(f(x) - f(y)) = 0$$ $$-f(x)f(y)f(z) + f(f(x)f(y)f(z)) = 0$$ $$-f(x) - f(y) - f(z) + f(f(x) + f(y) + f(z)) = 0$$ |

*Table 20.* RSR Properties Discovered by Claude Opus 4.1 (Agentic Bitween) across RSR-Bench (continued)

| Function | Discovered RSR Properties |
|---|---|
| 59. Fractional Part | $$-f(x) + f(x + \lfloor y \rfloor) = 0$$ $$f(xy) - f(xy - \lfloor x \rfloor \lfloor y \rfloor) = 0$$ $$f(x) + f(y) - f(x + y) - \lfloor f(x) + f(y) \rfloor = 0$$ $$f(x) + f(y) - f(x + y) + \lfloor x \rfloor + \lfloor y \rfloor - \lfloor x + y \rfloor = 0$$ |
| 60. Error Function | $$f(-ax) + f(ax) = 0$$ $$f\left(-\frac{x}{a}\right) + f\left(\frac{x}{a}\right) = 0$$ |
| 61. Gamma | $$-yf(x)f(y) + f(x)f(y + 1) = 0$$ $$-xyf(x)f(y) + f(x + 1)f(y + 1) = 0$$ $$-yf(y)f(x - 1) + f(x - 1)f(y + 1) = 0$$ $$-x(x + 1)f(x)f(y) + f(y)f(x + 2) = 0$$ |
| 62. Exp Sin | $$-f^{2\cos(r)}(x) + f(-r + x)f(r + x) = 0$$ |
| 64. Log Cosine | $$f(x - y) + f(x + y) - \log(|\cos(x - y)|) - \log(|\cos(x + y)|) = 0$$ |
| 65. Square Root of $1 + X^2$ | $$-x^2 y^2 + f^2(xy) - 1 = 0$$ $$-x^2 + y^2 + f^2(x) - f^2(y) = 0$$ $$-4rx - f^2(-r + x) + f^2(r + x) = 0$$ $$2xy - f^2(x) - f^2(y) + f^2(x - y) + 1 = 0$$ $$-2xy - f^2(x) - f^2(y) + f^2(x + y) + 1 = 0$$ $$2r^2 + 2f^2(x) - f^2(-r + x) - f^2(r + x) = 0$$ $$-(x^2 + 1)(y^2 + 1) + f^2(x)f^2(y) = 0$$ $$-2r^2 - 2f^2(x) + f^2(-r + x) + f^2(r + x) = 0$$ $$4x^2 y^2 - (x^2 + y^2 + 1)^2 + f^2(x - y)f^2(x + y) = 0$$ $$4x^2 y^2 - (x^2 + y^2 + 1)^2 + f^2(x - y)f^2(x + y) = 0$$ $$-\sqrt{c^2 + f^2(x) - 1} + f\left(\frac{x}{|c|}\right)|c| = 0$$ $$r^2 + f^2(x) - \frac{f^2(-r + x)}{2} - \frac{f^2(r + x)}{2} = 0$$ $$-2f^2(x) - 2f^2(y) + f^2(x - y) + f^2(x + y) + 2 = 0$$ $$-\sqrt{x^2 - 2xy + y^2 + 1}\sqrt{x^2 + 2xy + y^2 + 1} + f(x - y)f(x + y) = 0$$ $$4x^2 y^2 - (f^2(x) + f^2(y) - 1)^2 + f^2(x - y)f^2(x + y) = 0$$ |

*Table 20.* RSR Properties Discovered by Claude Opus 4.1 (Agentic Bitween) across RSR-Bench (continued)

| Function | Discovered RSR Properties |
|---|---|
| 66. Absolute Value | $$-f(x)\,|c| + f(cx) = 0$$ $$-f(x)f(y) + f(xy) = 0$$ $$f^2(x)f^2(y) - f^2(xy) = 0$$ $$-f^2(r)f^2(x) + f^2(rx) = 0$$ $$-\frac{f(x)}{f(y)} + f\left(\frac{x}{y}\right) = 0$$ $$f((-r+x)(r+x)) - f\left(-r^2 + x^2\right) = 0$$ $$-f^2(x) - f^2(y) + f^2\left(\sqrt{x^2 + y^2}\right) = 0$$ $$-2f^2(r) - 2f^2(x) + f^2(-r+x) + f^2(r+x) = 0$$ $$-14f^2(x) - 2f^2(y) - f^2(x-y) - f^2(x+y) + 2f^2(2x-y) + 2f^2(2x+y) = 0$$ |
| 67. Sign | $$f^3(xy) - f(xy) = 0$$ $$f^5(xy) - f(xy) = 0$$ $$f^3(x+y) - f(x+y) = 0$$ $$f^3(x-y) - f(x-y) = 0$$ |

*Table 20.* RSR Properties Discovered by Claude Opus 4.1 (Agentic Bitween) across RSR-Bench (continued)

| Function | Discovered RSR Properties |
|---|---|
| 68. Gudermannian | $$2\sinh(x)\cosh(r) + \sinh(r-x) - \sinh(r+x) = 0$$ $$2\sinh(r)\cosh(x) - \sinh(r-x) - \sinh(r+x) = 0$$ $$\sinh(r)\cosh(r+x) + \sinh(x) - \sinh(r+x)\cosh(r) = 0$$ $$-\sinh(r)\cosh(r-x) + \sinh(x) + \sinh(r-x)\cosh(r) = 0$$ $$f(r) - \operatorname{atan}\left(\frac{\sinh(r-x) + \sinh(r+x)}{2\cosh(x)}\right) = 0$$ $$f(x) - \operatorname{atan}\left(\frac{-\sinh(r-x) + \sinh(r+x)}{2\cosh(r)}\right) = 0$$ $$f(x+y) - \operatorname{atan}(\sinh(x)\cosh(y) + \sinh(y)\cosh(x)) = 0$$ $$f(r+x) - \operatorname{atan}(\sinh(r)\cosh(x) + \sinh(x)\cosh(r)) = 0$$ $$2\tan(f(x))\cosh(r) - \tan(f(-r+x)) - \tan(f(r+x)) = 0$$ $$f(-r+x) + \operatorname{atan}(\sinh(r)\cosh(x) - \sinh(x)\cosh(r)) = 0$$ $$2\tan(f(x))\cosh(r) - \tan(f(-r+x)) - \tan(f(r+x)) = 0$$ $$2\tan(f(r))\cosh(x) + \tan(f(-r+x)) - \tan(f(r+x)) = 0$$ $$f(x) - \operatorname{atan}\left(\frac{-\sinh(r-x) + \sinh(r+x)}{2\sqrt{\sinh^2(r) + 1}}\right) = 0$$ $$-\frac{\tan(f(-r+x)) + \tan(f(r+x))}{2\cosh(r)} + \tan(f(x)) = 0$$ $$\tan(f(x) - f(y)) - \frac{\sinh(x) - \sinh(y)}{\sinh(x)\sinh(y) + 1} = 0$$ $$\tan(f(x) + f(y)) - \frac{\sinh(x) + \sinh(y)}{-\sinh(x)\sinh(y) + 1} = 0$$ $$2\sqrt{\sinh^2(r) + 1}\tan(f(x)) - \tan(f(-r+x)) - \tan(f(r+x)) = 0$$ $$2\sqrt{\sinh^2(r) + 1}\tan(f(x)) - \tan(f(-r+x)) - \tan(f(r+x)) = 0$$ $$(\sinh(x)\sinh(y) + 1)\tan(f(x) - f(y)) - \sinh(x) + \sinh(y) = 0$$ $$(-\sinh(x)\sinh(y) + 1)\tan(f(x) + f(y)) - \sinh(x) - \sinh(y) = 0$$ $$f(x) - \operatorname{atan}\left(\frac{\tan(f(-r+x)) + \tan(f(r+x))}{2\cosh(r)}\right) = 0$$ $$-\frac{\tan(f(-r+x)) + \tan(f(r+x))}{2\sqrt{\sinh^2(r) + 1}} + \tan(f(x)) = 0$$ $$f(r) - \operatorname{atan}\left(\frac{-\tan(f(-r+x)) + \tan(f(r+x))}{2\cosh(x)}\right) = 0$$ $$f(x+y) - \operatorname{atan}\left(\sqrt{\sinh^2(x) + 1}\sinh(y) + \sqrt{\sinh^2(y) + 1}\sinh(x)\right) = 0$$ $$f(x+y)$$ $$-\operatorname{atan}\left(\sqrt{\tan^2(f(x)) + 1}\tan(f(y)) + \sqrt{\tan^2(f(y)) + 1}\tan(f(x))\right) = 0$$ |

*Table 20.* RSR Properties Discovered by Claude Opus 4.1 (Agentic Bitween) across RSR-Bench (continued)

| Function | Discovered RSR Properties |
|---|---|
| 69. 2 to X | $$-f^n(x) + f(nx) = 0$$ $$f^2(r+x) - f(2r+2x) = 0$$ $$f(r)f(x) - f(r+x) = 0$$ $$-f(x)f(y) + f(x+y) = 0$$ $$-f(x) + f(y)f(x-y) = 0$$ $$-f(r)f(x) + f(r+x) = 0$$ $$f(-r)f(r+x) - f(x) = 0$$ $$f(r)f(-r+x) - f(x) = 0$$ $$f(-r)f(x) - f(-r+x) = 0$$ $$-f(-r)f(x) + f(-r+x) = 0$$ $$-f(2x) + f(x-y)f(x+y) = 0$$ $$-f^2(x)f(y) + f(2x+y) = 0$$ $$-f(x)f^2(y) + f(x+2y) = 0$$ $$f^2\left(\frac{x}{2} + \frac{y}{2}\right) - f(x+y) = 0$$ $$-\frac{f(x)}{f(y)} + f(x-y) = 0$$ $$f(-r+x) - \frac{f(x)}{f(r)} = 0$$ $$-f^2(y)f(x-y) + f(x+y) = 0$$ $$-f(2x) + f(-r+x)f(r+x) = 0$$ $$-f(x) + \frac{f(x+y)}{f(y)} = 0$$ $$-f^2(x) + f(-r+x)f(r+x) = 0$$ $$-f^3(y)f(x-y) + f(x+2y) = 0$$ $$-f(x) + \frac{f(x-y)}{f(-y)} = 0$$ $$-f^3(y)f^2(x-y) + f(2x+y) = 0$$ $$-f(2y) + \frac{f(x+y)}{f(x-y)} = 0$$ $$-f^4(y)f^2(x-y) + f^2(x+y) = 0$$ $$-f(x) + \frac{f(x+y+z)}{f(y+z)} = 0$$ $$f(x)f(z) - f(y)f(x-y+z) = 0$$ $$-f(4y) + \frac{f^2(x+y)}{f^2(x-y)} = 0$$ $$-f(x)f(y)f(z) + f(x+y+z) = 0$$ $$-f(x)f(-y)f(z) + f(x-y+z) = 0$$ $$-f(x)f(y)f(-z) + f(x+y-z) = 0$$ $$-\frac{f(x)f(z)}{f(y)} + f(x-y+z) = 0$$ $$-\frac{f(x)f(y)}{f(z)} + f(x+y-z) = 0$$ $$-f^2(x)f(-y)f(y) + f(x-y)f(x+y) = 0$$ |

*Table 20.* RSR Properties Discovered by Claude Opus 4.1 (Agentic Bitween) across RSR-Bench (continued)

| Function | Discovered RSR Properties |
|---|---|
| 70. 10 to X | $$f(-r)f(r) - 1 = 0$$ $$f^y(x) - f(xy) = 0$$ $$-f^n(x) + f(nx) = 0$$ $$-f^a(x) + f(ax) = 0$$ $$-f^y(x) + f(xy) = 0$$ $$f^{\frac{1}{n}}(x) - f\left(\frac{x}{n}\right) = 0$$ $$-f(x)f(y) + f(x+y) = 0$$ $$f(-r)f(r+x) - f(x) = 0$$ $$f(r)f(-r+x) - f(x) = 0$$ $$-f^{\frac{1}{y}}(x) + f\left(\frac{x}{y}\right) = 0$$ $$-f(x)f(-y) + f(x-y) = 0$$ $$-f(-r)f(r+x) + f(x) = 0$$ $$-f(r)f(-r+x) + f(x) = 0$$ $$-f(2r)f(x) + f(2r+x) = 0$$ $$-f(r)f^2(x) + f(r+2x) = 0$$ $$-f^2(r)f(x) + f(2r+x) = 0$$ $$-f(2y)f(x-y) + f(x+y) = 0$$ $$-f^2(r)f(x) + f(2r+x) = 0$$ $$-\frac{f(x)}{f(y)} + f(x-y) = 0$$ $$-f^2(y)f(x-y) + f(x+y) = 0$$ $$-f^2(x) + f(x-y)f(x+y) = 0$$ $$-f(2x)f(2y) + f(2x+2y) = 0$$ $$-f^2(x) + f(-r+x)f(r+x) = 0$$ $$-f^a(x)f^b(y) + f(ax+by) = 0$$ $$f(-2r+x) - \frac{f(x)}{f^2(r)} = 0$$ $$-\frac{f^2(x)}{f(y)} + f(2x-y) = 0$$ $$-\frac{f^3(x)}{f^2(y)} + f(3x-2y) = 0$$ $$-f(x)f(y)f(z) + f(x+y+z) = 0$$ |

*Table 20.* RSR Properties Discovered by Claude Opus 4.1 (Agentic Bitween) across RSR-Bench (continued)

| Function | Discovered RSR Properties |
|---|---|
| 70.  10 to X (cont.) | $$-f(x)f(y) + f^2\left(\frac{x}{2} + \frac{y}{2}\right) = 0$$ $$-f(x)f(y)f(-z) + f(x + y - z) = 0$$ $$\sqrt{f(x)f(y)} - f\left(\frac{x}{2} + \frac{y}{2}\right) = 0$$ $$-\frac{f(x)}{f(y)f(z)} + f(x - y - z) = 0$$ $$-\frac{f(x)f(y)}{f(z)} + f(x + y - z) = 0$$ $$-\frac{f(x)f(z)}{f(y)} + f(x - y + z) = 0$$ $$-\frac{f(x)f(y)}{f(z)} + f(x + y - z) = 0$$ $$-af(x) + f\left(x + \frac{\log(a)}{\log(10)}\right) = 0$$ $$-f^a(x)f^b(y)f^c(z) + f(ax + by + cz) = 0$$ $$-f^2(x)f(-y)f(y) + f(x - y)f(x + y) = 0$$ $$-\left(f(-r) + f(r)\right)f(x) + f(-r + x) + f(r + x) = 0$$ $$-\left(-f(-r) + f(r)\right)f(x) - f(-r + x) + f(r + x) = 0$$ |
| 71. Pade 1,1 | $$-\frac{x}{2} - \frac{y}{2} + \frac{f(x + y) - 1}{f(x + y) + 1} = 0$$ $$-\frac{x}{2} + \frac{y}{2} + \frac{f(x - y) - 1}{f(x - y) + 1} = 0$$ $$f(x)f(x - y) + f(x)f(x + y) + 2f(x) - 2f(x - y)f(x + y) - f(x - y)$$ $$-f(x + y) = 0$$ |
| 72. Pade 2,2 | $$f(-x - y)f(-x + y)f(x - y)f(x + y) - 1 = 0$$ |

*Table 20.* RSR Properties Discovered by Claude Opus 4.1 (Agentic Bitween) across RSR-Bench (continued)

| Function | Discovered RSR Properties |
|---|---|
| 73. Continued Fraction Golden | $2f(r)f(r-s) + 2f(r)f(r+s) - 6f(r) - 4f(r-s)f(r+s)$ $+3f(r-s) + 3f(r+s) = 0$ $-2f(x)f(-r+x) - 2f(x)f(r+x) + 6f(x) + 4f(-r+x)f(r+x)$ $-3f(-r+x) - 3f(r+x) = 0$ $-2f(r)f(r+x) + 4f(r)f(r+2x) - 3f(r) - 2f(r+x)f(r+2x) + 6f(r+x)$ $-3f(r+2x) = 0$ $2f(rx)f(rx-s) + 2f(rx)f(rx+s) - 6f(rx) - 4f(rx-s)f(rx+s)$ $+3f(rx-s) + 3f(rx+s) = 0$ $2f(x)f(-r-s+x) - 4f(x)f(-r+s+x) - 4f(x)f(r-s+x)$ $+2f(x)f(r+s+x) + 6f(x) - 4f(-r-s+x)f(r+s+x)$ $+3f(-r-s+x) + 8f(-r+s+x)f(r-s+x)$ $-6f(-r+s+x) - 6f(r-s+x) + 3f(r+s+x) = 0$ |
| 74. Continued Fraction Tan | $-nx\left(-n^2x^2 + 15\right) + \left(-6n^2x^2 + 15\right)f(nx) = 0$ $15n^2 f\left(\dfrac{x}{n}\right) - 15nx - 6x^2 f\left(\dfrac{x}{n}\right) + \dfrac{x^3}{n} = 0$ $\left(15n^2 - 6x^2\right)f\left(\dfrac{x}{n}\right) - \dfrac{x\left(15n^2 - x^2\right)}{n} = 0$ |
| 75. Mobius Simple | $-x + f\left(\dfrac{-b+dx}{a-cx}\right) = 0$ $-\dfrac{arx+b}{crx+d} + f(rx) = 0$ $-r(ad-bc) + (cx+d)(-f(x)+f(r+x))(cr+cx+d) = 0$ $-\dfrac{ad-bc}{(-cr+cx+d)(cr+cx+d)} + \dfrac{-f(-r+x)+f(r+x)}{2r} = 0$ $-\dfrac{(r+s)(ad-bc)}{(cx+d)(cr+cs+cx+d)} - f(x) + f(r+s+x) = 0$ $f(x)f(y) - f(x+y) - \dfrac{-ax-ay-b+(cx+cy+d)f(x)f(y)}{cx+cy+d} = 0$ $f(r)f(s) - f(r+s) - \dfrac{-ar-as-b+(cr+cs+d)f(r)f(s)}{cr+cs+d} = 0$ $-\dfrac{(-ar+ax+b)(ar+ax+b)}{(-cr+cx+d)(cr+cx+d)} + f(-r+x)f(r+x) = 0$ $-\dfrac{(x_1-x_2)(x_3-x_4)}{(x_1-x_4)(-x_2+x_3)} + \dfrac{(f(x_1)-f(x_2))(f(x_3)-f(x_4))}{(f(x_1)-f(x_4))(-f(x_2)+f(x_3))} = 0$ $2rf(r)f(-x) - 2rf(-x)f(-r+x) - 2xf(r)f(-x) + 2xf(r)f(-r+x)$ $+9f(r)f(-x)f(-r+x) - f(r)f(-x) - f(r)f(-r+x) - f(-x)f(-r+x) = 0$ |

*Table 20.* RSR Properties Discovered by Claude Opus 4.1 (Agentic Bitween) across RSR-Bench (continued)

| Function | Discovered RSR Properties |
|---|---|
| 76. Mobius Inversion | $$f(x+y) - \frac{1}{x+y} = 0$$ $$f(x-y) - \frac{1}{x-y} = 0$$ $$(x+y)f(x+y) - 1 = 0$$ $$(x-y)f(x-y) - 1 = 0$$ $$f(ax+by) - \frac{1}{ax+by} = 0$$ $$f(cx) - \frac{f(x)}{c} = 0$$ $$-yf(x) + f\left(\frac{x}{y}\right) = 0$$ $$xf(x+y) + yf(x+y) - 1 = 0$$ $$xf(x-y) - yf(x-y) - 1 = 0$$ $$-f(x)f(y) + f(xy) = 0$$ $$f(y)f(x+y) - f(xy+y^2) = 0$$ $$f(y)f(x-y) - f(xy-y^2) = 0$$ $$f(x)f(x+y) - f(x^2+xy) = 0$$ $$f(x-y)f(x+y) - f(x^2-y^2) = 0$$ $$f(xy+z) - \frac{f(z)}{xyf(z)+1} = 0$$ $$f(x)f(y)f(z) - f(xyz) = 0$$ $$-2xf(x^2-y^2) + f(x-y) + f(x+y) = 0$$ $$x^2f(x-y)f(x+y) - y^2f(x-y)f(x+y) - 1 = 0$$ $$f(x)f(y) + f(x)f(x-y) - f(y)f(x-y) = 0$$ $$-f(x)f(y) + f(x)f(x+y) + f(y)f(x+y) = 0$$ $$-f(y)f(x-y) + f(y)f(x+y) + 2f(x-y)f(x+y) = 0$$ $$-(x+y+z)f(xyz) + f(x)f(y) + f(x)f(z) + f(y)f(z) = 0$$ |
| 77. Mobius Cayley | $$f(y)f(z)f(yz) - f(y) - f(z) + f(yz) = 0$$ $$f(x)f(y)f(xy) - f(x) - f(y) + f(xy) = 0$$ $$f(x)f(yz)f(xyz) - f(x) - f(yz) + f(xyz) = 0$$ $$f(x)f(x-y) + f(x)f(x+y) - 2f(x) - 2f(x-y)f(x+y)$$ $$+ f(x-y) + f(x+y) = 0$$ |

*Table 20.* RSR Properties Discovered by Claude Opus 4.1 (Agentic Bitween) across RSR-Bench (continued)

| Function | Discovered RSR Properties |
|---|---|
| 78. Exp $X^2$ | $$-f^{a^2}(x) + f(ax) = 0$$ $$-f^{y^2}(x) + f(xy) = 0$$ $$f(x)f(y) - f\left(\sqrt{x^2 + y^2}\right) = 0$$ $$-f^2(x)f^2(y) + f(x - y)f(x + y) = 0$$ $$-f^2(a)f^2(x) + f(-a + x)f(a + x) = 0$$ $$-f^2(2x)f^2(y) + f(2x - y)f(2x + y) = 0$$ $$-f^2(x)f^2(2y) + f(x - 2y)f(x + 2y) = 0$$ $$f\left(\sqrt{2}x\right)f\left(\sqrt{2}y\right) - f(x - y)f(x + y) = 0$$ $$f(x)f(y)f(z) - f\left(\sqrt{x^2 + y^2 + z^2}\right) = 0$$ $$-f^2(ax)f^2(by) + f(ax - by)f(ax + by) = 0$$ $$-f^2(z)f^2(x + y) + f(x + y - z)f(x + y + z) = 0$$ $$-f^2(x)f^2(y + z) + f(x - y - z)f(x + y + z) = 0$$ $$-f^2(x)f^2(y - z) + f(x - y + z)f(x + y - z) = 0$$ $$-f^2(x)f^4(y)f^2(z) + f(x - y)f(x + y)f(y - z)f(y + z) = 0$$ $$-f^4(x)f^4(y)f^4(z) + f(x - y - z)f(x - y + z)f(x + y - z)f(x + y + z) = 0$$ |
| 79. Exp Co-sine | $$f(r + x)f(r + x + \pi) - 1 = 0$$ $$f(-r + x)f(-r + x + \pi) - 1 = 0$$ $$f\left(r - \frac{\pi}{2}\right)f\left(r + \frac{\pi}{2}\right) - 1 = 0$$ $$f\left(r + x - \frac{\pi}{2}\right)f\left(r + x + \frac{\pi}{2}\right) - 1 = 0$$ $$f\left(-r + x - \frac{\pi}{2}\right)f\left(-r + x + \frac{\pi}{2}\right) - 1 = 0$$ |

*Table 20.* RSR Properties Discovered by Claude Opus 4.1 (Agentic Bitween) across RSR-Bench (continued)

| Function | Discovered RSR Properties |
|---|---|
| 80. Fourth Power | $$-c^4 f(x) + f(cx) = 0$$ $$-f(x)f(y) + f(xy) = 0$$ $$f(x-y)f(x+y) - f\left(x^2 - y^2\right) = 0$$ $$-\frac{f(x)}{f(y)} + f\left(\frac{x}{y}\right) = 0$$ $$f(-a+x)f(a+x) - f\left(-a^2 + x^2\right) = 0$$ $$12x^2 y^2 + 2f(x) + 2f(y) - f(x-y) - f(x+y) = 0$$ $$-12r^2 x^2 - 2f(r) - 2f(x) + f(-r+x) + f(r+x) = 0$$ $$-48x^2 y^2 - 2f(2x) - 2f(y) + f(2x-y) + f(2x+y) = 0$$ $$-24f(r) + 6f(x) + f(-2r+x) - 4f(-r+x) - 4f(r+x) + f(2r+x) = 0$$ $$-20f(x) + f(-3r+x) - 6f(-2r+x) + 15f(-r+x) + 15f(r+x)$$ $$-6f(2r+x) + f(3r+x) = 0$$ $$-24x^2 y^2 - 24x^2 z^2 - 24y^2 z^2 - 4f(x) - 4f(y) - 4f(z) + f(x-y-z)$$ $$+f(x-y+z) + f(x+y-z) + f(x+y+z) = 0$$ |

*Table 21.* RSR Properties Discovered by Claude Opus 4.1 (Agentic Bitween) across the algebraic benchmarks (A01–A17). Notation: DP = $\mathrm{Cl}(2,0)$ product determinant; DPM = matrix-product determinant; DT = $\mathrm{Cl}(2,0)$ determinant; OP = octonion-product norm$^2$; QN = quaternion norm$^2$; QP = quaternion-product norm$^2$.

| Function | Discovered RSR Properties |
|---|---|
| A01. $\det(A)$ (add.) | $$-kf(a,b,c,d) + f(ak,b,ck,d) = 0$$ $$-kf(a,b,c,d) + f(a,bk,c,dk) = 0$$ $$-f(x,y,z,w) + f(x+y,y,w+z,w) = 0$$ $$-f(x,y,z,w) + f(x,x+y,z,w+z) = 0$$ $$-k_1 k_2 f(a,b,c,d) + f(ak_1, bk_1, ck_2, dk_2) = 0$$ $$-kf(1,1,c,d) - f(a,b,c,d) + f(a+k,b+k,c,d) = 0$$ $$-kf(a,b,1,1) - f(a,b,c,d) + f(a,b,c+k,d+k) = 0$$ $$-2f(x,y,z,w) + f(x,y,z,w+z) + f(x,x+y,z,w) = 0$$ $$-2f(x,y,z,w) + f(x,y,w+z,w+z) + f(x+y,x+y,z,w) = 0$$ $$-f(x,y,z,w) - f(x,y,z,w+z) + f(x,y,w+z,w+z)$$ $$+f(x+y,y,z,w) = 0$$ $$-f(x,y,z,w) + f(x,y,z,w+z) + f(x,y,w+z,w)$$ $$-f(x,y,w+z,w+z) = 0$$ $$-f(a,b,c+s,d) + f(a,b,c+s,d+t) + f(a,b+s,c,d)$$ $$-f(a,b+s,c,d+t) = 0$$ $$-f(a,b,c,d) + f(a,b,c,d+r_4) + f(a,b+r_2,c,d)$$ $$-f(a,b+r_2,c,d+r_4) = 0$$ $$-f(a,b,c,d) + f(a,b,c,d+r_4) + f(a,b,c+r_3,d)$$ $$-f(a,b,c+r_3,d+r_4) = 0$$ $$f(a,b,c,d+r_4) - f(a,b+r_2,c,d+r_4) - f(a+r_1,b,c,d)$$ $$+f(a+r_1,b+r_2,c,d) = 0$$ $$f(a,b,c,d+r_4) - f(a,b,c+r_3,d+r_4) - f(a+r_1,b,c,d)$$ $$+f(a+r_1,b,c+r_3,d) = 0$$ $$-f(a_1,b_1,c_1,d_1) - f(a_1,b_2,c_1,d_2)$$ $$-f(a_2,b_1,c_2,d_1) - f(a_2,b_2,c_2,d_2)$$ $$+f(a_1+a_2,b_1+b_2,c_1+c_2,d_1+d_2) = 0$$ |

*Table 21.* RSR Properties Discovered by Claude Opus 4.1 (Agentic Bitween) across the algebraic benchmarks (continued)

| Function | Discovered RSR Properties |
|---|---|
| A02. $\det(A)$ (mult.) | $$-kf(a,b,c,d) + f(ak,bk,c,d) = 0$$ $$-kf(a,b,c,d) + f(a,b,ck,dk) = 0$$ $$-dr - f(a,b,c,d) + f(a+r,b,c,d) = 0$$ $$-ar - f(a,b,c,d) + f(a,b,c,d+r) = 0$$ $$-f(0,r,c,d) - f(a,b,c,d) + f(a,b+r,c,d) = 0$$ $$-f(a,b,c,d) - f(a,b,r,0) + f(a,b,c+r,d) = 0$$ $$-f(a,b,c,d) - f(r_1,r_2,c,d) + f(a+r_1,b+r_2,c,d) = 0$$ $$-f(a,b,c,d) - f(a,b,r_1,r_2) + f(a,b,c+r_1,d+r_2) = 0$$ $$f(a,b,c,r) - f(a,b,c,d+r) - f(r,b,c,d)$$ $$+ f(a+r,b,c,d) = 0$$ $$-f(a,b,c,d) - f(a,b,c,r) + f(a,b,c,d+r)$$ $$-f(a,r,c,d) + f(a,b+r,c,d) = 0$$ $$-f(a,b,c,d) - f(a,b,c,r) + f(a,b,c,d+r)$$ $$-f(a,b,r,d) + f(a,b,c+r,d) = 0$$ $$-d2rf(a1,b1,c1,d1) - \mathrm{DPM}(a1,b1,c1,d1,a2,b2,c2,d2)$$ $$+ \mathrm{DPM}(a1,b1,c1,d1,a2+r,b2,c2,d2) = 0$$ $$3f(a,b,c,d) - f(a,b,c,d+r_4) - f(a,b,c+r_3,d)$$ $$-f(a,b+r_2,c,d) - f(r_1,r_2,r_3,r_4)$$ $$-f(a+r_1,b,c,d) + f(a+r_1,b+r_2,c+r_3,d+r_4) = 0$$ |
| A03. $\mathrm{tr}(A)$ | $$-r - f(a,b,c,d) + f(a+r,b,c,d) = 0$$ $$-s - f(a,b,c,d) + f(a,b,c,d+s) = 0$$ $$-f(a,b,c,d) + f(a+r,b,c,d-r) = 0$$ $$-f(x,b,c,z) + f(x+y,b,c,-y+z) = 0$$ $$-r - s - f(a,b,c,d) + f(a+r,b,c,d+s) = 0$$ $$f(a,b,c,d+s) - f(r,b,c,s) - f(a-r,b,c,d) = 0$$ $$-f(a,b,c,d-s) - f(r,b,c,s) + f(a+r,b,c,d) = 0$$ $$-f(r,b,c,s) - f(x,b,c,y) + f(r+x,b,c,s+y) = 0$$ $$f(x,b,c,y) + f(y,b,c,x) - 2f\left(\frac{x+y}{2},b,c,\frac{x+y}{2}\right) = 0$$ $$2f(a,b,c,d) - f(a,b,c,d-s) - f(r,b,c,s)$$ $$-f(a-r,b,c,d) = 0$$ $$f(a_1,b,c,d_1) + f(a_2,b,c,d_2) - f(a_1+a_2,b,c,d_1+d_2) = 0$$ $$f(a_1,b,c,d_1) - f(a_2,b,c,d_2) - f(a_1-a_2,b,c,d_1-d_2) = 0$$ $$-f(a,b,c,d-s) + f(2a,b,c,2d) - f(r,b,c,s)$$ $$-f(a-r,b,c,d) = 0$$ $$-2f(a_2,b,c,d_2) - f(a_1-a_2,b,c,d_1-d_2)$$ $$+ f(a_1+a_2,b,c,d_1+d_2) = 0$$ $$f(a_1,b,c,d_1) + f(a_2,b,c,d_2) + f(a_3,b,c,d_3)$$ $$-f(a_1+a_2+a_3,b,c,d_1+d_2+d_3) = 0$$ |

*Table 21.* RSR Properties Discovered by Claude Opus 4.1 (Agentic Bitween) across the algebraic benchmarks (continued)

| Function | Discovered RSR Properties |
|---|---|
| A04. $\|q\|^2$ (add.) | $$-2f(a,b,c,d) - 2f(r,r,0,0) + f(a-r,b-r,c,d)$$ $$+f(a+r,b+r,c,d) = 0$$ $$-2f(0,0,t,t) - 2f(a,b,c,d) + f(a,b,c-t,d-t)$$ $$+f(a,b,c+t,d+t) = 0$$ $$-2f(a,b,c,d) - 2f(r,s,t,0) + f(a-r,b-s,c-t,d)$$ $$+f(a+r,b+s,c+t,d) = 0$$ $$-2f(a,b,c,d) - 2f(r,s,t,u) + f(a-r,b-s,c-t,d-u)$$ $$+f(a+r,b+s,c+t,d+u) = 0$$ $$-4f(a,b,c,d) - 4f(r,s,0,0) + f(a-r,b-s,c,d)$$ $$+f(a-r,b+s,c,d) + f(a+r,b-s,c,d) + f(a+r,b+s,c,d) = 0$$ |
| A05. $\|q_1 q_2\|^2$ | $$-k^2 \operatorname{QP}(a1,b1,c1,d1,a2,b2,c2,d2)$$ $$+ \operatorname{QP}(a1k,b1k,c1k,d1k,a2,b2,c2,d2) = 0$$ $$-k^2 \operatorname{QP}(a1,b1,c1,d1,a2,b2,c2,d2)$$ $$+ \operatorname{QP}(a1,b1,c1,d1,a2k,b2k,c2k,d2k) = 0$$ $$-2 \operatorname{QN}(a1,b1,c1,d1) - 2 \operatorname{QN}(a2,b2,c2,d2)$$ $$+ \operatorname{QN}(a1-a2,b1-b2,c1-c2,d1-d2)$$ $$+ \operatorname{QN}(a1+a2,b1+b2,c1+c2,d1+d2) = 0$$ $$-2 \operatorname{QP}(a1,b1,c1,d1,a2,b2,c2,d2) - 2 \operatorname{QP}(a1,b1,c1,d1,x,y,z,w)$$ $$+ \operatorname{QP}(a1,b1,c1,d1,a2-x,b2-y,c2-z,d2-w)$$ $$+ \operatorname{QP}(a1,b1,c1,d1,a2+x,b2+y,c2+z,d2+w) = 0$$ $$-2 \operatorname{QP}(a1,b1,c1,d1,a2,b2,c2,d2) - 2 \operatorname{QP}(x,y,z,w,a2,b2,c2,d2)$$ $$+ \operatorname{QP}(a1-x,b1-y,c1-z,d1-w,a2,b2,c2,d2)$$ $$+ \operatorname{QP}(a1+x,b1+y,c1+z,d1+w,a2,b2,c2,d2) = 0$$ |

*Table 21.* RSR Properties Discovered by Claude Opus 4.1 (Agentic Bitween) across the algebraic benchmarks (continued)

| Function | Discovered RSR Properties |
|---|---|
| A06. $\mathrm{tr}(A^2)$ | $$-k^2 f(a,b,c,d) + f(ak,bk,ck,dk) = 0$$ $$-4ar - f(a-r,b,c,d) + f(a+r,b,c,d) = 0$$ $$-4dr - f(a,b,c,d-r) + f(a,b,c,d+r) = 0$$ $$-4cr - f(a,b-r,c,d) + f(a,b+r,c,d) = 0$$ $$-4br - f(a,b,c-r,d) + f(a,b,c+r,d) = 0$$ $$-4bt - 4cs - f(a,b-s,c-t,d) + f(a,b+s,c+t,d) = 0$$ $$-2f(a,b,c,d) + f(a,b-r,c,d) + f(a,b+r,c,d) = 0$$ $$-2f(a,b,c,d) + f(a,b,c-r,d) + f(a,b,c+r,d) = 0$$ $$-2r\,(b+c+r) - f(a,b,c,d) + f(a,b+r,c+r,d) = 0$$ $$-2ar - 2dt - r^2 - t^2 - f(a,0,0,d) + f(a+r,0,0,d+t) = 0$$ $$-2b^2 - 2f(a,c,d,e) + f(a-b,c,d,e) + f(a+b,c,d,e) = 0$$ $$-4st - 2f(a,b,c,d) + f(a,b-s,c-t,d) + f(a,b+s,c+t,d) = 0$$ $$-2f(a,b,c,d) - 2f(r,0,0,0) + f(a-r,b,c,d)$$ $$+f(a+r,b,c,d) = 0$$ $$-2f(0,0,0,r) - 2f(a,b,c,d) + f(a,b,c,d-r)$$ $$+f(a,b,c,d+r) = 0$$ $$-2ar - 2bt - 2cs - 2du - f(a,b,c,d) - f(r,s,t,u)$$ $$+f(a+r,b+s,c+t,d+u) = 0$$ $$-2f(a,b,c,d) - 2f(r,s,t,u) + f(a-r,b-s,c-t,d-u)$$ $$+f(a+r,b+s,c+t,d+u) = 0$$ |
| A07. $e_2(x,y,z)$ | $$-c^2 f(x,y,z) + f(cx,cy,cz) = 0$$ $$-r\,(y+z) - f(x,y,z) + f(r+x,y,z) = 0$$ $$-r\,(x+z) - f(x,y,z) + f(x,r+y,z) = 0$$ $$-r\,(x+y) - f(x,y,z) + f(x,y,r+z) = 0$$ $$-r^2 - r\,(x+y+2z) - f(x,y,z) + f(r+x,r+y,z) = 0$$ $$-r^2 - r\,(x+2y+z) - f(x,y,z) + f(r+x,y,r+z) = 0$$ $$-r^2 - r\,(2x+y+z) - f(x,y,z) + f(x,r+y,r+z) = 0$$ $$-3r^2 - 2r\,(x+y+z) - f(x,y,z) + f(r+x,r+y,r+z) = 0$$ $$-rs - r\,(y+z) - s\,(x+z) - f(x,y,z)$$ $$+f(r+x,s+y,z) = 0$$ $$-rs - r\,(y+z) - s\,(x+y) - f(x,y,z)$$ $$+f(r+x,y,s+z) = 0$$ $$-rs - r\,(x+z) - s\,(x+y) - f(x,y,z)$$ $$+f(x,r+y,s+z) = 0$$ $$-f(r,r,r) + 2f(x,y,z) - f(x,y,r+z)$$ $$-f(x,r+y,z) - f(r+x,y,z) + f(r+x,r+y,r+z) = 0$$ |

*Table 21.* RSR Properties Discovered by Claude Opus 4.1 (Agentic Bitween) across the algebraic benchmarks (continued)

| Function | Discovered RSR Properties |
|---|---|
| A08. $e_3(x, y, z)$ | $$-2f(r, s, z) - 2f(r, y, t) - 2f(x, s, t)$$ $$-2f(x, y, z) + f(-r + x, -s + y, -t + z) + f(r + x, s + y, t + z) = 0$$ $$-f(r, s, t) - f(r, s, z) - f(r, y, t) - f(r, y, z)$$ $$-f(x, s, t) - f(x, s, z) - f(x, y, t) - f(x, y, z)$$ $$+f(r + x, s + y, t + z) = 0$$ $$f(r, s, t) + f(r, s, z) - f(r, y, t) - f(r, y, z)$$ $$+f(x, s, t) + f(x, s, z) - f(x, y, t) - f(x, y, z)$$ $$+f(r + x, -s + y, t + z) = 0$$ $$f(r, s, t) - f(r, s, z) + f(r, y, t) - f(r, y, z)$$ $$+f(x, s, t) - f(x, s, z) + f(x, y, t) - f(x, y, z)$$ $$+f(r + x, s + y, -t + z) = 0$$ $$f(r, s, t) + f(r, s, z) + f(r, y, t) + f(r, y, z)$$ $$-f(x, s, t) - f(x, s, z) - f(x, y, t) - f(x, y, z)$$ $$+f(-r + x, s + y, t + z) = 0$$ $$f(r, s, t) - f(r, s, z) - f(r, y, t) + f(r, y, z)$$ $$-f(x, s, t) + f(x, s, z) + f(x, y, t) - f(x, y, z)$$ $$+f(-r + x, -s + y, -t + z) = 0$$ |
| A09. $p_2(x, y, z)$ | $$n^2 f(x, y, z) - f(nx, ny, nz) = 0$$ $$-c^2 f(x, y, z) + f(cx, cy, cz) = 0$$ $$-4f(x, y, z) + f(x - y, y - z, -x + z) + f(x + y, y + z, x + z) = 0$$ $$-4(ax + by + cz) - f(-a + x, -b + y, -c + z) + f(a + x, b + y, c + z) = 0$$ $$-5f(x, y, z) + f(x, y + z, y - z) + f(x + y, x - y, z)$$ $$+f(x + z, y, x - z) = 0$$ $$-2f(r_1, r_2, r_3) - 2f(x, y, z) + f(-r_1 + x, -r_2 + y, -r_3 + z)$$ $$+f(r_1 + x, r_2 + y, r_3 + z) = 0$$ |

*Table 21.* RSR Properties Discovered by Claude Opus 4.1 (Agentic Bitween) across the algebraic benchmarks (continued)

| Function | Discovered RSR Properties |
|---|---|
| A10. $\|\mathbf{v}\|^2$ (2D) | $$-c^2 f(x,y) + f(cx, cy) = 0$$ $$2f(0,s) + 2f(x,y) - f(x, -s+y)$$ $$-f(x, s+y) = 0$$ $$-8f(r,0) - 2f(x,y) + f(2r+x, y)$$ $$+f(-2r+x, y) = 0$$ $$-8f(0,s) - 2f(x,y) + f(x, 2s+y)$$ $$+f(x, -2s+y) = 0$$ $$-2f(r,s) - 2f(x,y) + f(-r+x, -s+y)$$ $$+f(r+x, s+y) = 0$$ $$-2f(r,s) - 2f(x,y) + f(-r+x, s+y)$$ $$+f(r+x, -s+y) = 0$$ $$-4f(r,s) - 4f(x,y) + f(-r+x, -s+y)$$ $$+f(-r+x, s+y) + f(r+x, -s+y) + f(r+x, s+y) = 0$$ $$2f(x,y) + f(x, 2s+y) + 3f(-r+x, -s+y)$$ $$-4f(-r+x, s+y) - 4f(r+x, -s+y) + f(r+x, s+y)$$ $$+f(2r+x, y) = 0$$ |
| A11. $\|\mathbf{a} \times \mathbf{b}\|^2$ | $$-c^2 f(x_1, y_1, z_1, x_2, y_2, z_2) + f(cx_1, cy_1, cz_1, x_2, y_2, z_2) = 0$$ $$-c^2 f(x_1, y_1, z_1, x_2, y_2, z_2) + f(x_1, y_1, z_1, cx_2, cy_2, cz_2) = 0$$ $$-(ab)^2 f(x_1, y_1, z_1, x_2, y_2, z_2) + f(ax_1, ay_1, az_1, bx_2, by_2, bz_2) = 0$$ $$-b^2 f(x_1, y_1, z_1, x_2, y_2, z_2) + f(x_1, y_1, z_1, ax_1 + bx_2, ay_1 + by_2, az_1 + bz_2) = 0$$ $$-2f(r_1, r_2, r_3, x_2, y_2, z_2) - 2f(x_1, y_1, z_1, x_2, y_2, z_2)$$ $$+f(-r_1 + x_1, -r_2 + y_1, -r_3 + z_1, x_2, y_2, z_2)$$ $$+f(r_1 + x_1, r_2 + y_1, r_3 + z_1, x_2, y_2, z_2) = 0$$ $$-2f(x_1, y_1, z_1, r_1, r_2, r_3) - 2f(x_1, y_1, z_1, x_2, y_2, z_2)$$ $$+f(x_1, y_1, z_1, -r_1 + x_2, -r_2 + y_2, -r_3 + z_2)$$ $$+f(x_1, y_1, z_1, r_1 + x_2, r_2 + y_2, r_3 + z_2) = 0$$ |

*Table 21.* RSR Properties Discovered by Claude Opus 4.1 (Agentic Bitween) across the algebraic benchmarks (continued)

| Function | Discovered RSR Properties |
|---|---|
| A12. $\|o\|^2$ (8D) | |

$$-2t^2 - 2f(a,b,c,d,p,q,r,s) + f(a-t,b,c,d,p,q,r,s)$$
$$+f(a+t,b,c,d,p,q,r,s) = 0$$
$$-2t^2 - 2f(a,b,c,d,p,q,r,s) + f(a,b-t,c,d,p,q,r,s)$$
$$+f(a,b+t,c,d,p,q,r,s) = 0$$
$$-4t^2 - 2t\,(a+b-c-d) - f(a,b,c,d,p,q,r,s)$$
$$+f(a+t,b+t,c-t,d-t,p,q,r,s) = 0$$
$$-4t^2 - 2t\,(a+c+p+r) - f(a,b,c,d,p,q,r,s)$$
$$+f(a+t,b,c+t,d,p+t,q,r+t,s) = 0$$
$$-4t^2 - 2f(a,b,c,d,p,q,r,s) + f(a-t,b+t,c,d,p,q,r,s)$$
$$+f(a+t,b-t,c,d,p,q,r,s) = 0$$
$$-4t^2 - 2f(a,b,c,d,p,q,r,s) + f(a-t,b,c+t,d,p,q,r,s)$$
$$+f(a+t,b,c-t,d,p,q,r,s) = 0$$
$$-8t^2 - 2f(a,b,c,d,p,q,r,s) + f(a-t,b-t,c-t,d-t,p,q,r,s)$$
$$+f(a+t,b+t,c+t,d+t,p,q,r,s) = 0$$
$$-2f(a,b,c,d,p,q,r,s) - 2f(t,t,0,0,0,0,0,0) + f(a-t,b-t,c,d,p,q,r,s)$$
$$+f(a+t,b+t,c,d,p,q,r,s) = 0$$
$$-8t^2 - 2t\,(a+b+c+d+p+q+r+s) - f(a,b,c,d,p,q,r,s)$$
$$+f(a+t,b+t,c+t,d+t,p+t,q+t,r+t,s+t) = 0$$
$$-8t^2 - 2t\,(a-b+c-d+p-q+r-s)$$
$$-f(a,b,c,d,p,q,r,s)$$
$$+f(a+t,b-t,c+t,d-t,p+t,q-t,r+t,s-t) = 0$$
$$-8t^2 - 2t\,(a+b+c+d-p-q-r-s)$$
$$-f(a,b,c,d,p,q,r,s)$$
$$+f(a+t,b+t,c+t,d+t,p-t,q-t,r-t,s-t) = 0$$
$$-2t_1^2 - 2t_2^2$$
$$-2f(a,b,c,d,p,q,r,s) + f(a-t_1,b-t_2,c,d,p,q,r,s)$$
$$+f(a+t_1,b+t_2,c,d,p,q,r,s) = 0$$
$$-2t_1^2 - 2t_2^2$$
$$-2f(a,b,c,d,p,q,r,s) + f(a-t_1,b,c-t_2,d,p,q,r,s)$$
$$+f(a+t_1,b,c+t_2,d,p,q,r,s) = 0$$

*Table 21.* RSR Properties Discovered by Claude Opus 4.1 (Agentic Bitween) across the algebraic benchmarks (continued)

| Function | Discovered RSR Properties |
|---|---|
| A12. $\|o\|^2$ (8D) (cont.) | $$-f(a, b, c-t, d-t, p, q, r, s)$$ $$-f(a, b, c+t, d+t, p, q, r, s)$$ $$+f(a-t, b-t, c, d, p, q, r, s) + f(a+t, b+t, c, d, p, q, r, s) = 0$$ $$-f(a, b-t, c, d-t, p, q, r, s)$$ $$-f(a, b+t, c, d+t, p, q, r, s)$$ $$+f(a-t, b, c-t, d, p, q, r, s) + f(a+t, b, c+t, d, p, q, r, s) = 0$$ $$-16t^2 - 2f(a, b, c, d, p, q, r, s)$$ $$+f(a-t, b-t, c-t, d-t, p-t, q-t, r-t, s-t)$$ $$+f(a+t, b+t, c+t, d+t, p+t, q+t, r+t, s+t) = 0$$ $$-2\left(ar_1 + br_2 + cr_3 + dr_4 + pr_5 + qr_6 + rr_7 + r_8 s\right)$$ $$-f(a, b, c, d, p, q, r, s) - f(r_1, r_2, r_3, r_4, r_5, r_6, r_7, r_8)$$ $$+f(a+r_1, b+r_2, c+r_3, d+r_4, p+r_5, q+r_6, r+r_7, r_8+s) = 0$$ |
| A13. $\|o_1 o_2\|^2$ | $$-k1^2 k2^2 \, \mathrm{OP}(a1, b1, c1, d1, p1, q1, r1, s1,$$ $$a2, b2, c2, d2, p2, q2, r2, s2)$$ $$+ \mathrm{OP}(a1k1, b1k1, c1k1, d1k1, k1p1, k1q1, k1r1, k1s1,$$ $$a2k2, b2k2, c2k2, d2k2, k2p2, k2q2, k2r2, k2s2) = 0$$ $$-2\, \mathrm{OP}(a1, b1, c1, d1, p1, q1, r1, s1,$$ $$a2, b2, c2, d2, p2, q2, r2, s2)$$ $$-2\, \mathrm{OP}(r1, r2, r3, r4, r5, r6, r7, r8,$$ $$a2, b2, c2, d2, p2, q2, r2, s2)$$ $$+ \mathrm{OP}(a1 - r1, b1 - r2, c1 - r3, d1 - r4, p1 - r5, q1 - r6, r1 - r7, -r8 + s1,$$ $$a2, b2, c2, d2, p2, q2, r2, s2)$$ $$+ \mathrm{OP}(a1 + r1, b1 + r2, c1 + r3, d1 + r4, p1 + r5, q1 + r6, r1 + r7, r8 + s1,$$ $$a2, b2, c2, d2, p2, q2, r2, s2) = 0$$ |

*Table 21.* RSR Properties Discovered by Claude Opus 4.1 (Agentic Bitween) across the algebraic benchmarks (continued)

| Function | Discovered RSR Properties |
|---|---|
| A14. Cl$(3,0)$ conj. | |

$$-c^2 f(s, v_1, v_2, v_3, b_1, b_2, b_3, t) + f(cs, cv_1, cv_2, cv_3, b_1 c, b_2 c, b_3 c, ct) = 0$$

$$-2f(s, v_1, v_2, v_3, b_1, b_2, b_3, t) + f(-r + s, v_1, v_2, v_3, b_1, b_2, b_3, r + t)$$
$$+f(r + s, v_1, v_2, v_3, b_1, b_2, b_3, -r + t) = 0$$

$$2f(0, 0, 0, 0, 0, 0, 0, r) + 2f(s, v_1, v_2, v_3, b_1, b_2, b_3, t)$$
$$-f(s, v_1, v_2, v_3, b_1, b_2, b_3, -r + t) - f(s, v_1, v_2, v_3, b_1, b_2, b_3, r + t) = 0$$

$$2f(0, 0, 0, 0, r, 0, 0, 0) - 2f(s, v_1, v_2, v_3, b_1, b_2, b_3, t)$$
$$+f(s, v_1, v_2, v_3, b_1, b_2, b_3, -r + t) + f(s, v_1, v_2, v_3, b_1, b_2, b_3, r + t) = 0$$

$$2f(0, r, 0, 0, 0, 0, 0, 0) + 2f(s, v_1, v_2, v_3, b_1, b_2, b_3, t)$$
$$-f(s, v_1, v_2, v_3, b_1, b_2, b_3, -r + t) - f(s, v_1, v_2, v_3, b_1, b_2, b_3, r + t) = 0$$

$$2f(r, 0, 0, 0, 0, 0, 0, 0) - 2f(s, v_1, v_2, v_3, b_1, b_2, b_3, t)$$
$$+f(s, v_1, v_2, v_3, b_1, b_2, b_3, -r + t) + f(s, v_1, v_2, v_3, b_1, b_2, b_3, r + t) = 0$$

$$-f(s, v_1, v_2, v_3, b_1, b_2, b_3, -r + t) - f(s, v_1, v_2, v_3, b_1, b_2, b_3, r + t)$$
$$+f(s, -r + v_1, v_2, v_3, b_1, b_2, b_3, t) + f(s, r + v_1, v_2, v_3, b_1, b_2, b_3, t) = 0$$

$$2f(0, 0, 0, r_2, 0, 0, r_3, 0) + 2f(s, v_1, v_2, v_3, b_1, b_2, b_3, t)$$
$$-f(s, v_1, v_2, -r_2 + v_3, b_1, b_2, b_3 - r_3, t)$$
$$-f(s, v_1, v_2, r_2 + v_3, b_1, b_2, b_3 + r_3, t) = 0$$

$$2f(0, 0, r_2, 0, 0, r_3, 0, 0) + 2f(s, v_1, v_2, v_3, b_1, b_2, b_3, t)$$
$$-f(s, v_1, v_2, -r_2 + v_3, b_1, b_2, b_3 - r_3, t)$$
$$-f(s, v_1, v_2, r_2 + v_3, b_1, b_2, b_3 + r_3, t) = 0$$

$$2f(0, 0, 0, 0, r_1, r_2, r_3, 0) + 2f(s, v_1, v_2, v_3, b_1, b_2, b_3, t)$$
$$-f(s, v_1, v_2, v_3, b_1 - r_1, b_2 - r_2, b_3 - r_3, t)$$
$$-f(s, v_1, v_2, v_3, b_1 + r_1, b_2 + r_2, b_3 + r_3, t) = 0$$

$$2f(0, r_1, r_2, r_3, 0, 0, 0, 0) - 2f(s, v_1, v_2, v_3, b_1, b_2, b_3, t)$$
$$+f(s, v_1, v_2, v_3, b_1 - r_1, b_2 - r_2, b_3 - r_3, t)$$
$$+f(s, v_1, v_2, v_3, b_1 + r_1, b_2 + r_2, b_3 + r_3, t) = 0$$

$$-2f(r_1, r_2, 0, 0, r_3, 0, 0, 0) - 2f(s, v_1, v_2, v_3, b_1, b_2, b_3, t)$$
$$+f(-r_1 + s, -r_2 + v_1, v_2, v_3, b_1 - r_3, b_2, b_3, t)$$
$$+f(r_1 + s, r_2 + v_1, v_2, v_3, b_1 + r_3, b_2, b_3, t) = 0$$

$$-f(s, v_1, v_2, -r_2 + v_3, b_1, b_2, b_3 - r_3, t)$$
$$-f(s, v_1, v_2, r_2 + v_3, b_1, b_2, b_3 + r_3, t) + f(s, v_1, -r_2 + v_2, v_3, b_1, b_2 - r_3, b_3, t)$$
$$+f(s, v_1, r_2 + v_2, v_3, b_1, b_2 + r_3, b_3, t) = 0$$

$$-4f(s, v_1, v_2, v_3, b_1, b_2, b_3, t) + f(s, v_1, v_2, v_3, b_1, b_2, b_3, -r + t)$$
$$+f(s, v_1, v_2, v_3, b_1, b_2, b_3, r + t) + f(-r + s, v_1, v_2, v_3, b_1, b_2, b_3, t)$$
$$+f(r + s, v_1, v_2, v_3, b_1, b_2, b_3, t) = 0$$

$$-2f(r_1, r_2, r_3, r_4, r_5, r_6, r_7, r_8) - 2f(s, v_1, v_2, v_3, b_1, b_2, b_3, t)$$
$$+f(-r_1 + s, -r_2 + v_1, -r_3 + v_2, -r_4 + v_3, b_1 - r_5, b_2 - r_6, b_3 - r_7, -r_8 + t)$$
$$+f(r_1 + s, r_2 + v_1, r_3 + v_2, r_4 + v_3, b_1 + r_5, b_2 + r_6, b_3 + r_7, r_8 + t) = 0$$

$$-4f(s, v_1, v_2, v_3, b_1, b_2, b_3, t) + f(s, v_1, v_2, v_3, b_1 - r_1, b_2 - r_2, b_3 - r_3, t)$$
$$+f(s, v_1, v_2, v_3, b_1 + r_1, b_2 + r_2, b_3 + r_3, t)$$
$$+f(s, -r_1 + v_1, -r_2 + v_2, -r_3 + v_3, b_1, b_2, b_3, t)$$
$$+f(s, r_1 + v_1, r_2 + v_2, r_3 + v_3, b_1, b_2, b_3, t) = 0$$

*Table 21.* RSR Properties Discovered by Claude Opus 4.1 (Agentic Bitween) across the algebraic benchmarks (continued)

| Function | Discovered RSR Properties |
|---|---|
| A15. $\mathrm{Cl}(2,0)$ det. | $$-2\,\mathrm{DT}\,(r,a,b,c) - 2\,\mathrm{DT}\,(s,0,0,0)$$ $$+\,\mathrm{DT}\,(-r+s,a,b,c) + \mathrm{DT}\,(r+s,a,b,c) = 0$$ $$-4\,\mathrm{DT}\,(s,a,b,c) + \mathrm{DT}\,(s,a,b,c-r)$$ $$+\,\mathrm{DT}\,(s,a,b,c+r) + \mathrm{DT}\,(s,a,b-r,c)$$ $$+\,\mathrm{DT}\,(s,a,b+r,c) = 0$$ $$\mathrm{DP}\,(-r+s1,a1,b1,c1,s2,a2,b2,c2) + \mathrm{DP}\,(r+s1,a1,b1,c1,s2,a2,b2,c2)$$ $$-2\,\mathrm{DT}\,(r,0,0,0)\,\mathrm{DT}\,(s2,a2,b2,c2)$$ $$-2\,\mathrm{DT}\,(s1,a1,b1,c1)\,\mathrm{DT}\,(s2,a2,b2,c2) = 0$$ $$\mathrm{DP}\,(s1,a1,b1,c1,-r+s2,a2,b2,c2) + \mathrm{DP}\,(s1,a1,b1,c1,r+s2,a2,b2,c2)$$ $$-2\,\mathrm{DT}\,(r,0,0,0)\,\mathrm{DT}\,(s1,a1,b1,c1)$$ $$-2\,\mathrm{DT}\,(s1,a1,b1,c1)\,\mathrm{DT}\,(s2,a2,b2,c2) = 0$$ $$-4\,\mathrm{DP}\,(s1,a1,b1,c1,s2,a2,b2,c2) + \mathrm{DP}\,(s1,a1,b1,c1-r,s2,a2,b2,c2)$$ $$+\,\mathrm{DP}\,(s1,a1,b1,c1+r,s2,a2,b2,c2) + \mathrm{DP}\,(s1,a1-r,b1,c1,s2,a2,b2,c2)$$ $$+\,\mathrm{DP}\,(s1,a1+r,b1,c1,s2,a2,b2,c2) = 0$$ $$-4\,\mathrm{DP}\,(s1,a1,b1,c1,s2,a2,b2,c2) + \mathrm{DP}\,(s1,a1,b1,c1-r,s2,a2,b2,c2)$$ $$+\,\mathrm{DP}\,(s1,a1,b1,c1+r,s2,a2,b2,c2) + \mathrm{DP}\,(s1,a1,b1-r,c1,s2,a2,b2,c2)$$ $$+\,\mathrm{DP}\,(s1,a1,b1+r,c1,s2,a2,b2,c2) = 0$$ |
| A16. $\mathrm{tr}([A,B]^2)$ | $$-k^2 f(a_1,b_1,c_1,d_1,a_2,b_2,c_2,d_2) + f(a_1 k,b_1 k,c_1 k,d_1 k,a_2,b_2,c_2,d_2) = 0$$ $$-k^2 f(a_1,b_1,c_1,d_1,a_2,b_2,c_2,d_2) + f(a_1,b_1,c_1,d_1,a_2 k,b_2 k,c_2 k,d_2 k) = 0$$ $$-k^4 f(a_1,b_1,c_1,d_1,a_2,b_2,c_2,d_2) + f(a_1 k,b_1 k,c_1 k,d_1 k,a_2 k,b_2 k,c_2 k,d_2 k) = 0$$ |

*Table 21.* RSR Properties Discovered by Claude Opus 4.1 (Agentic Bitween) across the algebraic benchmarks (continued)

| Function | Discovered RSR Properties |
|---|---|
| A17. $\mathfrak{sl}(2)$ Killing | $$-f(a,b,ck) + f(a,bk,c) = 0$$ $$-2f(a,b,c) + f(a,b-s,c) + f(a,b+s,c) = 0$$ $$-2f(a,b,c) + f(a,b,c-t) + f(a,b,c+t) = 0$$ $$-f(0,2a,2r) - f(a-r,b,c) + f(a+r,b,c) = 0$$ $$-f(0,b_2,c) - f(a,b_1,c) + f(a,b_1+b_2,c) = 0$$ $$-f(0,b,c_2) - f(a,b,c_1) + f(a,b,c_1+c_2) = 0$$ $$f(a,0,d) - f(a,b,d) - f(a,c,d) + f(a,b+c,d) = 0$$ $$f(a,b,0) - f(a,b,c) - f(a,b,d) + f(a,b,c+d) = 0$$ $$2f(0,b,c) + 2f(a,b,0) - f(a,b,c-t)$$ $$-f(a,b,c+t) = 0$$ $$-2f(a,b,c) - 2f(r,0,0) + f(a-r,b,c)$$ $$+f(a+r,b,c) = 0$$ $$-2f(a,b,0) - 2f(r,b,c) + f(a-r,b,c)$$ $$+f(a+r,b,c) = 0$$ $$-6f(a,b,0) - f(a,b,c-t) - f(a,b,c+t)$$ $$+2f(2a,b,c) = 0$$ $$-f(a,b,c-t) - f(a,b,c+t) + f(a,b-s,c)$$ $$+f(a,b+s,c) = 0$$ $$-2a_1 a_2 + bc - f(a_1,b,c) - f(a_2,b,c)$$ $$+f(a_1+a_2,b,c) = 0$$ $$f(a-r,b-s,c) - f(a-r,b+s,c) - f(a+r,b-s,c)$$ $$+f(a+r,b+s,c) = 0$$ $$2f(0,c,d) - 2f(a,c,d) - 2f(b,c,d)$$ $$+f(a-b,c,d) + f(a+b,c,d) = 0$$ $$-4f(a,b,c) - 2f(r,0,0) - 2f(r,s,0)$$ $$+f(a-r,b-s,c) + f(a-r,b+s,c) + f(a+r,b-s,c)$$ $$+f(a+r,b+s,c) = 0$$ |

