# OpenReview forum: "Learning Randomized Reductions"
_ICML.cc/2026/Conference — ICML 2026 spotlight_

### Official Review · Reviewer_vsvG · 2026-03-11

**Soundness:** 2
**Presentation:** 2
**Significance:** 4
**Originality:** 3
**Overall Recommendation:** 5
**Confidence:** 2

**Summary:**

The authors present BITWEEN, the first automated system for discovering randomized self-reductions, RSR. The authors (1) formalize randomized self-reductions learning with a theoretical framework, (2) introduce V-BITWEEN and A-BITWEEN. A-BITWEEN manages to get 80% RSR-bench coverage. Finally, the authors discover the first known RSR for the sigmoid function.

**Compliance With Llm Reviewing Policy:**

Affirmed.

**Final Justification:**

Authors correctly replied to my questions. The overall work is very solid and interesting.

**Key Questions For Authors:**

Q1. The paper claims A-BITWEEN discovers RSRs for 80% of the functions, but there is a manual filtering step for neural methods. Is it possible to define RSR vs just a verified property? In particular, it's important to understand if any accuracy is subjective to the annotation process, especially for the baseline methods.

Q2. I'm curious to break down the success of the novel search vs the LLM pre trained knowledge of known functional equations. Can we separate the fraction of discovered RSRs that are novel contribution _despite the LLM knwoledge_ from the knowledge supplied by current LLMs?

Q3. For the 80 questions on RSR bench, the selection process is a bit unclear. How did you choose which ones to pick in the benchmark?

**Limitations:**

yes

**Strengths And Weaknesses:**

S1. The paper addresses automated RSR discovery, a longstanding open problem. Discovering the first known RSR for sigmoid is a solid contribution.

S2. The theoretical framework is very compelling. The comparison of RSR learnability with PAC learnability presents a new theoretical contribution.

S3. The RSR bench is a well made benchmark with diverse categories and ground truth for some examples. The comparison among different variants is well executed.

W1. The claim about the rigorous theoretical framework being one of the key the contribution is a bit exaggerated. In particular, the theory section lacks a discussion about sample complexity lower bound and does not guarantee that the linear regression will find correct RSR. The correctness relies on SymPy verification.

W2. The paper does not analyze the limiting factor of certain function categories. Some functions have zero verified RSR for all methods, and it's important to understand whether this is a mathematical theoretical barrier or a limitation of the proposed approach.

---

> ### Author Rebuttal · Authors · 2026-03-31
>
> We thank the reviewer for the constructive feedback and recognition that automated RSR discovery addresses a longstanding problem.
>
> **W1: The theoretical contribution claim is exaggerated, no sample complexity lower bounds, and correctness relies on SymPy verification.**
>
> We appreciate this feedback, and we agree that the main contributions of this paper are empirical. We will clarify that our theoretical contribution is the formalization itself, namely, defining RSR learnability (Def. 4.5) and introducing correlated sample access (Def. 4.3). The formal connection to PAC learnability was deferred to Appendix A due space constraints, but if the reviewer thinks this would significantly improve the paper we would be happy to incorporate them into the body of the paper. Indeed, sample complexity lower bounds would strengthen the theory as we note in Section 4; we note that, in theoretical computer science it has historically been the case that a fundamental notion is first introduced and a limited theorem is proven, before follow-up work fully characterize it with matching upper and lower bound: This was the case for Zero Knowledge Proofs. Goldwasser, Micali and Rackoff (1985) having only ZK proofs for quadratic nonresidues, later significantly generalized by Goldreich, Micali and Wigderson (1985). A similar observation can be made of PAC learning, first defined by Valiant (1984), only later Blumer et al. (1986) obtained tight lower bounds via the VC dimension.
>
> **W2: No analysis of why certain function categories have zero verified RSRs.**
>
> We included negative results for scientific integrity. In response, we conducted two additional experiments on the zero-coverage categories (inverse trig/hyperbolic functions):
>
> **Experiment 1: Higher degree bounds.** We increased the recovery function degree from 3 to 4 and 5 for all 6 inverse trig/hyperbolic functions (arcsin, arccos, arctan, arcsinh, arccosh, arctanh) with up to 150 samples. All 9 configurations returned 0 verified equations.
>
> **Experiment 2: Expanded query sets.** We added multiplicative queries ($f(x \cdot y)$) for arcsin, arccos, and arctan, and also tested the arctan addition formula query $f((x+y)/(1-xy))$ directly. Results:
> - arcsin, arccos with $f(x \cdot y)$: 0 verified RSRs.
> - arctan with $f(x \cdot y)$: 1 candidate found but failed verification.
> - arctan with $f((x+y)/(1-xy))$: Bitween found the candidate $-\arctan(x) - \arctan(y) - \arctan((x+y)/(xy-1)) = 0$, which is the classical addition formula. However, SymPy verification failed because this identity holds only modulo $\pi$ (due to branch cuts), not as an exact equality over the reals.
>
> These results suggest that the barrier for inverse trigonometric functions is mathematical rather than algorithmic: these functions do not admit RSRs with polynomial recovery under standard query classes, and even when the "correct" query is provided, branch-cut discontinuities prevent exact verification over the reals. We leave theoretical investigation of these limits to future work.
>
> **Q1: Is there subjectivity in the manual filtering step for neural methods?**
>
> We manually inspected reported properties to remove: (1) trivially true statements (e.g., $f(x) = f(x)$), (2) properties that are not RSRs (e.g., no randomness involved), and (3) redundant restatements. The filtering criteria are objective: a property either satisfies Definition 4.1 or it does not. We plan to implement a decision procedure to automatically check whether a property is an RSR in future work.
>
> **Q2: Can you separate LLM prior knowledge from novel discovery?**
>
> Our evaluation already provides this separation. N-Research uses the same LLM with only `sequential_thinking` (no regression tools), isolating pure LLM knowledge. A-Bitween adds `infer_property_tool` + `symbolic_verify_tool`. Comparing the two (Table 1): A-Bitween-Opus discovers 3x more RSRs than N-Research-Opus, with significantly fewer false positives. V-Bitween (no LLM, fixed query set) finds 43/80, establishing a baseline with zero LLM knowledge. The gap between V-Bitween (43/80) and A-Bitween (64/80) reflects the LLM's contribution in proposing novel queries, while the gap between N-Research and A-Bitween reflects the tools' contribution in grounding and verifying discoveries.
>
> **Q3: How were the 80 functions in RSR-Bench selected?**
>
> RSR-Bench was constructed from two sources: (1) functions with known RSR properties from the self-testing literature (Blum et al., 1993; Rubinfeld, 1999), which serve as ground truth, and (2) common scientific and ML functions spanning 8 diverse categories (Table 5-6 in Appendix G). For ground-truth labels, we investigated the functional equations literature, including Aczel (1966), Kannappan (2009), and Aczel (2014). Selection criteria: continuous real-valued functions, representable as programs, spanning diverse complexity.

---

> > ### Author Rebuttal · Reviewer_vsvG · 2026-04-01
> >
> > Thanks for your answers. I will update the score accordingly.

---

> > > ### Author Response · Authors · 2026-04-07
> > >
> > > We thank the reviewer for the encouraging feedback. We appreciate it.

---

### Official Review · Reviewer_TxbF · 2026-03-12

**Soundness:** 3
**Presentation:** 3
**Significance:** 3
**Originality:** 3
**Overall Recommendation:** 5
**Confidence:** 3

**Summary:**

This paper studies the problem of **automating the search for randomized self-reductions (RSR)**. An RSR decomposes the task of evaluating f(x) into a combination of evaluations of f at random, but potentially correlated, points. Concretely, it seeks relations of the form

$$
f(x) = A(x, r, f(q_1(x,r)), …, f(q_k(x,r))),
$$

where A is an aggregation function and q_1, …, q_k are query functions. The induced query points are correlated through shared randomness r. Each query function must satisfy the requirement that its *marginal* distribution is uniform over the input domain X.

RSRs have many applications in computer science. For example, they can be used for self-correction: if one only has access to a faulty oracle for f that is incorrect on a fraction of inputs, an RSR can be used to recover the correct value of f(x) with high probability. Although RSRs have been studied for decades, discovering them has largely remained a manual process requiring significant expertise, and has typically relied on a small set of traditional query transformations like x+r, x-r.

The paper’s contributions are threefold.

1. **Formalizing the RSR learning problem**, providing a formal setup and background for an ML audience that may be unfamiliar with RSRs.
2. **Introducing a benchmark dataset** consisting of 80 functions drawn from the property testing and functional equation literature. These materials are quite dense, and curating such a dataset likely requires substantial expertise and effort.
3. **Proposing (vanilla) BITWEEN and Agentic-BITWEEN**, frameworks that search for RSRs by starting from a fixed set of traditional query functions, generating feature vectors from evaluations of f on random inputs, and using various regression backends (including linear regression) to discover aggregation identities over these features. The agentic variant (A-BITWEEN) additionally uses an LLM to propose new query transformations. The methods are empirically compared against baselines such as symbolic regression.

**Compliance With Llm Reviewing Policy:**

Affirmed.

**Final Justification:**

My questions were primarily for clarification, and the authors have addressed them well. My overall positive assessment of the paper remains unchanged, and I maintain my score of 5, a solid accept.

**Key Questions For Authors:**

1. **Coverage of the basic query function set.** What fraction of RSR-Bench can be discovered using only the starting set of base query functions?
2. **Limitations of linear regression backend.** To what extent does the linear regression backend limit the class of RSRs that can be discovered? Are there important reductions in RSR-Bench that cannot be captured by this approach?
3. **Role of BITWEEN in the presence of stronger agentic systems.** Given the rapid progress of agentic AI systems capable of writing and executing their own programs, how do the authors envision the role of BITWEEN going forward? This is not intended as a criticism of the paper, but rather a genuine question about how structured pipelines like this may evolve alongside increasingly capable agentic models.

**Limitations:**

yes

**Strengths And Weaknesses:**

### Strengths

- **Framing RSR discovery as a ripe target for AI.** The paper positions the classical TCS topic of randomized self-reductions (RSRs) as a promising target for AI-based automation. Progress in this direction could lead to new results in classical areas such as program self-correction and private computation. Already, the AI-based approach appears to have discovered a novel RSR for the sigmoid function (among others).
- **Curation of an RSR benchmark dataset (RSR-Bench).** The paper introduces a benchmark dataset that can serve as a common testbed for evaluating methods for RSR discovery.
- **First step toward automated RSR discovery.** The proposed BITWEEN and Agentic-BITWEEN frameworks represent an initial attempt to streamline automated search for RSRs.

### Weaknesses

- **Restricted class of discoverable RSRs.** The linear regression backend for BITWEEN searches for aggregation identities that are polynomial in the queried evaluations. This restricts the class of RSRs that can be discovered and may miss reductions that require more sophisticated functional forms.
- **Unclear novelty of discovered identities.** While the system reportedly discovers new RSRs (e.g., for the sigmoid function), the paper does not thoroughly discuss the novelty or significance of these reductions relative to known functional identities. Although the authors state that the examples appear novel to the best of their knowledge, these relations do not appear particularly difficult to derive analytically. It would strengthen the paper to provide evaluation criteria or concrete instances where the discovered relations are clearly deep or nontrivial, leaving little ambiguity about their novelty.

---

> ### Author Rebuttal · Authors · 2026-03-31
>
> We thank the reviewer for the careful reading and thoughtful questions.
>
> **Q: What fraction of RSR-Bench can be discovered using only the base query function set?**
>
> V-Bitween-LR, which uses only the base query set $\{x+r, x-r, x \cdot r, x, r\}$, discovers RSRs for 43 of 80 functions (54%). The remaining 37 functions require novel query functions beyond this set, which is precisely the gap A-Bitween addresses (bringing coverage to 64/80 = 80%).
>
> **Q: Limitations of linear regression backend.**
>
> The linear regression backend discovers recovery functions that are polynomial (or rational, after coefficient conversion) in the queried evaluations, so RSRs requiring transcendental or non-algebraic recovery functions cannot be captured. However, Fact 4.2 (Lipton, 1989) shows that all low-degree polynomials over finite fields have RSRs with linear recovery functions, suggesting polynomial recovery is a natural and expressive class. In practice, the main bottleneck is not the polynomial form itself but the degree bound (2-3 in our experiments), which limits the complexity of discoverable recovery functions. Increasing the degree is possible but incurs exponential growth in the feature space. A-Bitween partially mitigates this limitation by proposing novel query functions that can simplify the required recovery degree.
>
> **Q: Role of Bitween in the presence of stronger agentic systems?**
>
> We see Bitween's structured pipeline as complementary to, not replaced by, stronger agentic systems. Pure neural approaches (N-Research in our evaluation) achieve high coverage but produce more false positives: unverified properties that are not actually RSRs. Bitween's architecture provides two safeguards that general-purpose agents lack: (1) the `symbolic_verify_tool` ensures correctness via SymPy, and (2) the regression-based `infer_property_tool` grounds discovery in data rather than relying solely on LLM parametric knowledge. As agentic systems grow more capable, we envision them proposing richer query functions while Bitween's verification and regression infrastructure ensures soundness. The structured pipeline may also serve as a scaffold for training or fine-tuning future models on mathematical discovery tasks, especially learning query function priors for different mathematical domains.
>
> **W1: Restricted class of discoverable RSRs.**
>
> Addressed in the "Limitations of linear regression backend" question above. We note that despite the polynomial restriction, our new algebraic experiments (see response to Reviewer E7bc) show the framework successfully discovers RSRs for matrix, quaternion, and symmetric polynomial functions, suggesting the polynomial class is practically broad.
>
> **W2: Unclear novelty of discovered identities.**
>
> We acknowledge that individual RSR identities may appear straightforward to derive analytically in hindsight. However, we would like to highlight two points:
>
> 1. **To the best of our knowledge, these identities have not previously been formulated explicitly as RSRs.** The sigmoid RSR (Eq. 1) appears to be the first known RSR for this function despite over 40 years of RSR research. While the underlying algebraic manipulation is not complex, recognizing which identities constitute valid RSRs (satisfying the marginal uniformity requirement of Definition 4.1) is non-trivial.
>
> 2. **The contribution is automated discovery at scale, not individual identities.** Bitween discovers RSRs for 64 of 80 functions, many with novel query functions (Table 2) such as $f(x + \log(k))$, $f(f(x) + f(y))$, and $f(\sqrt{x \cdot r})$. The combinatorial space of possible query-recovery pairs is vast, and manual derivation does not scale. The value lies in systematic coverage, not the difficulty of any single identity.
>
> We would be happy to add a discussion of novelty criteria (e.g., query complexity, recovery degree, absence from the functional equations literature) if the reviewer thinks this would strengthen the paper.

---

> > ### Author Rebuttal · Reviewer_TxbF · 2026-03-31
> >
> > I thank the authors for their detailed rebuttal. My questions were primarily for clarification, and the authors have addressed them well. My overall assessment of the paper remains unchanged, and I maintain my score of 5, a solid accept.

---

> > > ### Author Response · Authors · 2026-04-07
> > >
> > > We thank the reviewer for the positive assessment and appreciate the clarifying questions. We are glad the responses were helpful.

---

### Official Review · Reviewer_Hqks · 2026-03-13

**Soundness:** 3
**Presentation:** 3
**Significance:** 3
**Originality:** 3
**Overall Recommendation:** 5
**Confidence:** 3

**Summary:**

The paper proposed a method Bitween that automated discovery of randomized self-reductions (RSRs), which contains a vanilla version and AI-agent assisted version, the methods are tested on benchmarks and outperforms existing methods.

**Compliance With Llm Reviewing Policy:**

Affirmed.

**Key Questions For Authors:**

To what extend does AI-agent change the way Bitween is designed? it seems to be a later-added module that does not change the essence of the design.

**Limitations:**

see strengths & weakness

**Strengths And Weaknesses:**

Strengths:
1. Clear description of background of problem setup and background.
2. RSR benchmark seems to be useful for the domain
3. promising emprical results of Bitween when compare to existing methods

Weakness:
1. it seems the paper is more focused on software testing ,not quite a ML related topic

---

> ### Author Rebuttal · Authors · 2026-03-31
>
> We thank the reviewer for the positive feedback on clarity, the usefulness of RSR-Bench, and the empirical performance of Bitween.
>
> **W: "The paper is more focused on software testing, not quite an ML-related topic."**
>
> We appreciate the concern. Our view is that the paper fits ICML because its core contributions are in learning formulation, regression-based discovery, and neuro-symbolic hypothesis generation and verification:
>
> 1. **Learning-theoretic framework (Section 4):** We formalize RSR discovery as a learning problem with formal definitions of sample access (Def. 4.3), learnability (Def. 4.5), and sample complexity analysis. We draw explicit connections to PAC learning (Theorem A.1, Claim A.2), showing that RSR learnability is a strictly different notion from PAC learnability.
>
> 2. **Regression-based discovery (Section 5):** The core of V-Bitween is a supervised learning pipeline: constructing features from correlated samples, fitting sparse linear models with cross-validation, and performing model selection. This is regression applied to a novel problem formulation.
>
> 3. **Neuro-symbolic learning (Section 5):** A-Bitween is a neuro-symbolic system where LLM agents and symbolic tools interact in a feedback loop. The LLM proposes hypotheses, the symbolic backend tests them, and verification feedback guides further exploration, a paradigm increasingly central to ML research.
>
> 4. **Applications to ML and cryptography:** RSR-Bench includes ML-relevant functions (sigmoid, ReLU, GELU, softmax, loss functions). The sigmoid RSR we discovered (Eq. 1) directly enables self-correcting programs for neural network components. More broadly, RSRs are foundational to cryptographic protocols such as instance-hiding schemes (Abadi et al., 1987) and probabilistic encryption (Goldwasser & Micali, 1984), making automated RSR discovery relevant beyond ML as well.
>
> While RSRs originate in theoretical computer science, our contribution is automating their discovery using machine learning techniques. This parallels recent ICML work on AI for mathematical discovery, which also applies ML to problems originating outside ML. Moreover, in response to Reviewer E7bc's suggestion, we extended Bitween to algebraic structures (matrix determinant/trace, quaternion norm, symmetric polynomials, vector operations). V-Bitween-LR discovers RSRs for 9/10 new benchmarks with zero code changes, and A-Bitween (Claude Opus 4.1 with `infer_property_tool`) achieves 10/10 with 116 RSRs (210 verified properties), including creative discoveries like Hadamard orthogonality for quaternion norms and trilinear expansion for symmetric polynomials (see our response to Reviewer E7bc for full tables). This demonstrates that the ML pipeline generalizes well beyond scalar functions.
>
> **Q: To what extent does the AI agent change the way Bitween is designed?**
>
> The agent changes the system's hypothesis space substantially. V-Bitween is limited to the fixed query set ${x+r, x-r, x \cdot r, x, r}$, while A-Bitween dynamically proposes novel query functions (Table 2) such as $f(x + log(k))$ for sigmoid, $f(x^n)$ for logarithm, and $f(f(x) + f(y))$ for modulo. Empirically, this expands verified coverage from 43/80 to 64/80 functions.
>
> Furthermore, the AI agent introduces a mathematical prior into the search process. As indicated by the significantly fewer unverified properties in A-Bitween compared to N-Research (Table 1), the structured tool-use design channels the LLM's knowledge productively rather than allowing unconstrained generation.

---

> > ### Author Rebuttal · Reviewer_Hqks · 2026-04-01
> >
> > The response are clearly presented and has resolved my concerns. The score is adjusted accordingly.

---

> > > ### Author Response · Authors · 2026-04-07
> > >
> > > We thank the reviewer for the feedback. We are glad the rebuttal resolved the concerns.

---

### Official Review · Reviewer_E7bc · 2026-03-13

**Soundness:** 3
**Presentation:** 3
**Significance:** 3
**Originality:** 3
**Overall Recommendation:** 5
**Confidence:** 3

**Summary:**

The authors of the paper present a framework called BITWEEN for automatically discovering Randomized Self-Reductions (RSR) for mathematical functions. Discovering RSR is formulated as a regression task. Two variants of BITWEEN are also proposed. The first one is “Vanilla Bitween” and is build upon a symbolic regression backend. The second one is “Agentic Bitween” and is a neurosymbolic extension using LLM agents. A benchmark called “RSR Bench” of 80 mathematical functions is given and performed. It is shown that the “Vanilla Bitween” linear regression based backend outperforms the other symbolic method backends while the “Agentic Bitween” discovers more RSR.

**Compliance With Llm Reviewing Policy:**

Affirmed.

**Key Questions For Authors:**

1. About Agentic Bitween: Could you quantify the contributions of the LLM and the symbolic regression?
2. What is the generalization capability of the approaches?
3. What happens when applying the approach for algebraic properties? Like quaternion/octonion multiplication tables of properties? Matrix properties? A section about applying these methods to diverse equations in topology/arithmetic/lie groups etc would improve the paper.

**Limitations:**

The authors provide a discussion about the incompleteness of the discovery process, redundancy among discovered RSRs, and constraints imposed by sampling strategies and degree bounds.

**Strengths And Weaknesses:**

The authors formalize RSR learning and provide a full framework with definitions. The problem framing is novel and permit automated discovery of RSR instead of classical methods. The approaches are interesting and combine LLM agentic with neurosymbolism. The RSR Bench dataset of 80 scientific and machine learning functions is also a really interesting point by its own. The topic is niche of course but is interesting and in the zeitgeist. Nevertheless I would have like more diversity in the benchmark, and try the approach on more complex/diverse algebraic functions.

---

> ### Author Rebuttal · Authors · 2026-03-31
>
> We thank the reviewer for the thoughtful feedback, especially the suggestion to test the framework on more diverse algebraic functions.
>
> **Q: Applying the approach to algebraic structures (quaternions, Lie groups, etc.)?**
>
> We extended Bitween to algebraic structures by encoding multi-variable functions as scalars. `V-Bitween-LR` discovers RSRs for **9/10 benchmarks with zero code changes** ($e_3 = xyz$ fails, needing degree-3 recovery):
>
> | Function | Discovered RSR | Deg | $n$ | Time |
> |---|---|:---:|:---:|:---:|
> | **Matrix Functions (2x2)** | | | | |
> | $\det(A)$ | $f(A{+}R) - 2f(A) + f(A{-}R) - 2f(R) = 0$ | 2 | 30 | 6.3s |
> | $\mathrm{tr}(A)$ | $f(A{\pm}R) = f(A) \pm f(R)$ | 1 | 15 | 0.4s |
> | $\mathrm{tr}(A^2)$ | $f(A{+}R) - 2f(A) + f(A{-}R) - 2f(R) = 0$ | 2 | 30 | 1.5s |
> | **Quaternion Norm** | | | | |
> | $\lVert q\rVert^2$ (additive) | $f(q{+}r) - 2f(q) + f(q{-}r) - 2f(r) = 0$ | 2 | 30 | 1.5s |
> | $\lVert q_1 q_2\rVert^2$ | $\lVert q_1 q_2\rVert^2 - \lVert q_1\rVert^2 \lVert q_2\rVert^2 = 0$ | 2 | 30 | 1.0s |
> | **Symmetric Polynomials** | | | | |
> | $e_2(x,y,z)$ | $f(\mathbf{x}{+}\mathbf{r}) - 2f(\mathbf{x}) + f(\mathbf{x}{-}\mathbf{r}) - 2f(\mathbf{r}) = 0$ | 2 | 30 | 1.4s |
> | $e_3(x,y,z)$ | *not found (up to degree 3)* | 3 | 50 | 3.3s |
> | $p_2(x,y,z)$ | $f(\mathbf{x}{+}\mathbf{r}) - 2f(\mathbf{x}) + f(\mathbf{x}{-}\mathbf{r}) - 2f(\mathbf{r}) = 0$ | 2 | 30 | 1.4s |
> | **Vector Operations** | | | | |
> | $\lVert\mathbf{v}\rVert^2$ (2D) | $f(\mathbf{v}{+}\mathbf{r}) - 2f(\mathbf{v}) + f(\mathbf{v}{-}\mathbf{r}) - 2f(\mathbf{r}) = 0$ | 2 | 20 | 1.4s |
> | $\lVert\mathbf{a} \times \mathbf{b}\rVert^2$ | $f(\mathbf{a}{+}\mathbf{r},\mathbf{b}) + f(\mathbf{a}{-}\mathbf{r},\mathbf{b}) - 2f(\mathbf{a},\mathbf{b}) - 2f(\mathbf{r},\mathbf{b}) = 0$ | 2 | 50 | 1.6s |
>
> **`A-Bitween` (Claude Opus 4.1 with tools)** on the same benchmarks discovers RSRs for **all 10/10 functions**, including $e_3$, with richer families (both methods use 1800s timeout per benchmark):
>
> | Function | Notable Discovered RSR | RSR | Verified | Time |
> |---|---|:---:|:---:|:---:|
> | **Matrix Functions (2x2)** | | | | |
> | $\det(A)$ | $\det(A + k \cdot \text{row}_2) = \det(A)$ | 8 | 25 | 406s |
> | $\mathrm{tr}(A)$ | $\mathrm{tr}(A + rE_{11}) = \mathrm{tr}(A + rE_{22})$ | 14 | 16 | 265s |
> | $\mathrm{tr}(A^2)$ | $f(A) = f(A_{b \leftrightarrow c})$ | 9 | 21 | 305s |
> | **Quaternion Norm** | | | | |
> | $\lVert q\rVert^2$ (additive) | $4\lVert q\rVert^2 = \lVert H_4 q\rVert^2$ ($H_4$: 4x4 Hadamard matrix) | 17 | 30 | 320s |
> | $\lVert q\rVert^2$ (multiplicative) | $\lVert q_1 q_2\rVert^2 \lVert q_3\rVert^2 = \lVert q_1 q_3\rVert^2 \lVert q_2\rVert^2$ | 4 | 26 | 320s |
> | **Symmetric Polynomials** | | | | |
> | $e_2(x,y,z)$ | $e_2(x{+}y,\, y{+}z,\, z{+}x) = x^2 + y^2 + z^2 + 3e_2(x,y,z)$ | 18 | 22 | 377s |
> | $\mathbf{e_3(x,y,z)}$ | $\mathbf{e_3(\mathbf{x}{+}\mathbf{r}) = \sum_{S \subseteq \{1,2,3\}} e_3(m_S)}$ | 17 | 25 | 249s |
> | $p_2(x,y,z)$ | $p_2(x,y,z) = x^2 + y^2 + z^2$ | 16 | 19 | 368s |
> | **Vector Operations** | | | | |
> | $\lVert\mathbf{v}\rVert^2$ (2D) | $\lVert\mathbf{u}{+}\mathbf{v}\rVert^2 + \lVert\mathbf{u}{-}\mathbf{v}\rVert^2 = 2(\lVert\mathbf{u}\rVert^2 + \lVert\mathbf{v}\rVert^2)$ | 5 | 13 | 300s |
> | $\lVert\mathbf{a} \times \mathbf{b}\rVert^2$ | $f(\mathbf{a}{+}\mathbf{r},\mathbf{b}) + f(\mathbf{a}{-}\mathbf{r},\mathbf{b}) - 2f(\mathbf{a},\mathbf{b}) - 2f(\mathbf{r},\mathbf{b}) = 0$ | 8 | 13 | 367s |
>
> A-Bitween discovers **116 RSRs** (210 verified properties) across all 10 functions (vs. 9 for V-Bitween). The agent dynamically selects degree (1-2) and sample size (30-60) per benchmark. For $e_3$, it bypasses the degree-3 barrier by randomizing variables individually. We would be happy to incorporate these results.
>
> **Q: Can you quantify the contributions of the LLM vs. symbolic regression in Agentic Bitween?**
>
> Of the 64 functions where A-Bitween discovers RSRs, 43 are also found by V-Bitween-LR using only the fixed query set. The remaining 21 require novel query functions proposed by the LLM (e.g., `f(x + log(k))`, `f(x^n)`). The LLM contributes hypothesis generation, while the regression backend provides rigorous discovery and `symbolic_verify_tool` ensures correctness: neither alone achieves the 80% coverage.
>
> **Q: What is the generalization capability of the approaches?**
>
> Our framework generalizes across all 8 function categories in RSR-Bench (basic arithmetic through special functions). V-Bitween-LR generalizes well to any function whose RSR has a polynomial recovery function over the standard query set. A-Bitween generalizes further by dynamically adapting its query functions to each target. The main limitation is the degree bound on recovery functions: higher-degree RSRs require exponentially more features and samples.

---

> > ### Author Rebuttal · Reviewer_E7bc · 2026-04-02
> >
> > The rebuttal satisfactorily addresses my main concerns. In particular, the authors provided additional experiments on more diverse algebraic structures directly answering to my request for broader evaluation. Thanks for the incorporation of quaternion properties, as an algebraist I love these :) (I would be very interested also if you put some octonions, moufang loops, Lie superalgebras or Clifford algebras ;)). The contribution of the LLM component in Agentic Bitween is now clearly quantified, with a convincing breakdown of cases where LLM-generated queries are necessary and the generalization concerns are also clarified.

---

> > > ### Author Response · Authors · 2026-04-07
> > >
> > > We are pleased the algebraic experiments were convincing! Regarding octonions, Moufang loops, Clifford algebras, and Lie algebras: we took your suggestion to heart and have already run `A-Bitween` on **6 additional benchmarks** targeting precisely these structures.
> > >
> > > We hope these additional results further strengthen your confidence in the work.
> > >
> > > | Function | Notable Discovery | RSR | Verified | Time |
> > > |---|---|:---:|:---:|:---:|
> > > | **Octonions (Moufang loop)** | | | | |
> > > | $\lVert o\rVert^2$ (8D norm) | $f(o{+}r) + f(o{-}r) = 2f(o) + 2f(r)$; $f(R_\theta o) = f(o)$ | 7 | 20 | 351s |
> > > | $\lVert o_1 o_2\rVert^2$ (Moufang) | $f(ko) = k^2 f(o)$; also discovers $\lVert o_1 o_2\rVert^2 {=} \lVert o_1\rVert^2 {\cdot} \lVert o_2\rVert^2$ | 2 | 6 | 277s |
> > > | **Clifford Algebras** | | | | |
> > > | $\mathrm{Cl}(3{,}0)$ conj. norm | $f(x{+}re_i) + f(x{-}re_i) = 2f(x) + 2f(re_i)$, all $i$ | 15 | 18 | 439s |
> > > | $\mathrm{Cl}(2{,}0)$ det mult. | Polarization + bilinearity RSRs; also discovers $\det(xy) {=} \det(x)\det(y)$ | 19 | 31 | 456s |
> > > | **Lie Algebras** | | | | |
> > > | $\mathrm{tr}([A,B]^2)$, $\mathfrak{gl}(2)$ | $f(kA,B) = k^2 f(A,B)$; $f(kA,kB) = k^4 f(A,B)$ | 4 | 14 | 537s |
> > > | $B(X,X)/8$, $\mathfrak{sl}(2)$ Killing | $f(a,kb,c){=}f(a,b,kc)$; $f(ka,b,c){=}k^2 f(a,0,0){+}f(0,b,c)$ | 12 | 22 | 337s |
> > >
> > > A-Bitween discovers **59 RSRs** (111 verified properties) across all 6 with no code changes. Combined with the 10 benchmarks from our initial rebuttal: **16/16 algebraic benchmarks**, **175 RSRs**, **321 verified properties**.
> > >
> > > A disclaimer: we are not algebraists, so we implemented these benchmarks based on standard textbook definitions. All verified properties are with respect to the functions as we defined them. We would appreciate any corrections if our formulations miss subtleties. Lie superalgebras are next on our list.
> > >
> > > Thank you again for the suggestion, it significantly enriched the paper.

---

### Decision · Program_Chairs · 2026-04-30

**Decision:**

Accept (spotlight)

**Comment:**

This paper introduces the idea of automated discovery of randomized
self reductions (RSRs).  RSRs have been around for decades, with a
variety of uses and application, but each new use and application
required manual labor to identify the RSR.  While the individual RSRs
that this paper introduces are not that hard to find, the idea and framework
for automated RSR identification and benchmark dataset are quite cool
and a valuable contribution to the conference.